# Spurious Correlation Learning in Preference Optimization: Mechanisms, Consequences, and Mitigation via Tie Training

Christian Moya [1]  Alex Semendinger [2]  Guang Lin [1 3]  Elliott Thornley [4]

## Abstract

Preference learning methods like Direct Preference Optimization (DPO) are known to induce reliance on spurious correlations, leading to sycophancy and length bias in today's language models and potentially severe goal misgeneralization in future systems. In this work, we provide a unified theoretical analysis of this phenomenon, characterizing the mechanisms of spurious learning, its consequences on deployment, and a provable mitigation strategy. Focusing on log-linear policies, we show that standard preference-learning objectives induce reliance on spurious features at the population level through two channels: mean spurious bias and causal-spurious correlation leakage. We then show that this reliance creates an irreducible vulnerability to distribution shift: more data from the same training distribution fails to reduce the model's dependence on spurious features. To address this, we propose *tie training*, a data augmentation strategy using ties (equal-utility preference pairs) to introduce data-driven regularization. We demonstrate that this approach selectively reduces spurious learning without degrading causal learning. Finally, we validate our theory on log-linear models and provide empirical evidence that both the spurious learning mechanisms and the benefits of tie training persist for neural networks and large language models.

## 1. Introduction

Aligning large language models (LLMs) with human preferences is a key challenge in building safe and useful AI systems. In current alignment pipelines, Reinforcement Learning from Human Feedback (RLHF) learns a reward model from preference data and optimizes a policy with respect to that reward (Ziegler et al., 2019; Ouyang et al., 2022). Direct Preference Optimization (DPO) simplifies this pipeline by directly optimizing the policy on preference pairs (Rafailov et al., 2023). These approaches are used to align widely deployed systems such as ChatGPT (Ouyang et al., 2022) and Claude (Bai et al., 2022). Despite their success, existing theory provides limited insight into how preference optimization behaves under the distributional structure of real-world human feedback.

Understanding this behavior requires examining the structure of preference data itself. Preference optimization methods are trained on datasets of human comparisons (Christiano et al., 2017; Rafailov et al., 2023), where annotators select preferred responses for given a prompt. These datasets encode recurring patterns, reflecting consistent annotator biases and shared superficial characteristics among preferred responses. As a result, feature-level correlations arise between surface attributes and preference labels that are not causally related to response quality.

Surface-level attributes such as length, politeness, formatting, or agreement with the user often correlate with preference labels during training (Sharma et al., 2024; Casper et al., 2023), but may not reflect true response quality. When these correlations shift at deployment, models that rely on them fail to generalize, a behavior we call policy misgeneralization: rather than learning to optimize response quality, the policy learns to optimize a proxy that is non-causally correlated with response quality on the training distribution.

Policy misgeneralization has important safety implications beyond standard failures under distribution shift. Prior work has raised concerns that AI systems may learn objectives that correlate with intended goals during training but pursue misaligned proxy objectives once those correlations break at deployment (Langosco et al., 2022; Shah et al., 2022). In such cases, high training reward can reflect alignment with proxy signals rather than improvements in true task performance, masking failures that emerge under distribution shift (Skalse et al., 2022). While much of this literature focuses on hypothetical capable agents (Ngo et al., 2024;

[1]Department of Mathematics, Purdue University, West Lafayette IN, USA [2]Cambridge Boston Alignment Initiative, Cambridge MA, USA [3]School of Mechanical Engineering, Purdue University, West Lafayette IN, USA [4]Massachusetts Institute of Technology, Cambridge MA, USA. Correspondence to: Christian Moya <cmoyacal@purdue.edu>.

*Proceedings of the 43rd International Conference on Machine Learning*, Seoul, South Korea. PMLR 306, 2026. Copyright 2026 by the author(s).

Bengio et al., 2025), preference optimization in current LLMs provides a concrete setting where this failure mode manifests even without objective misspecification. Understanding the mechanisms by which spurious correlations emerge in these systems is therefore essential for developing robust alignment methods.

Despite these risks, existing work on spurious correlations in preference optimization remains largely empirical. Prior studies report failures such as verbosity (Saito et al., 2023), but describe symptoms rather than identify underlying mechanisms. While supervised learning has developed mathematical frameworks for analyzing spurious correlations through shortcut learning (Geirhos et al., 2020), preference optimization methods such as DPO lack analogous theory. Without such understanding, mitigation strategies remain largely heuristic and lack principled guarantees.

To address this gap, we develop a mathematical framework for spurious correlation learning in preference optimization. We analyze log-linear DPO as a representative and tractable testbed for pairwise preference optimization, and characterize how feature correlations interact with the optimization objective. Our contributions are:

*(i)* We characterize the mechanism of spurious learning by analyzing the population equilibrium of the linearized log-linear DPO objective. We prove that mean spurious bias or causal-spurious correlation in the training distribution induces nonzero spurious parameters (Theorem 4.1). This shows spurious learning arises structurally from the data, not from finite-sample effects or optimization noise.

*(ii)* We analyze deployment consequences when spurious statistics shift between training and deployment. We use the expected preference margin as a population-level deployment proxy to characterize the shift term (Propositions 5.1 and 5.2). To understand finite-sample behavior, we decompose deployment suboptimality into an irreducible shift term driven by spurious parameters and a reducible estimation term that decays as $n \to \infty$ (Theorem 5.3). This shows that scaling training data cannot eliminate shift-induced error.

*(iii)* We propose tie training, a data augmentation strategy that reduces spurious correlation reliance by adding preference pairs with equal utility but differing spurious features. These ties inject curvature along spurious directions, selectively regularizing spurious parameters without affecting causal learning (Theorem 6.2 (i)). We prove that such ties can reduce the irreducible shift error at deployment (Theorem 6.2 (iii)).

We validate our framework through controlled experiments that progressively relax modeling assumptions. Linear models confirm quantitative agreement with theory. Neural networks show the same qualitative mechanisms persist despite hidden representations. In large language models,

*tie training* reduces spurious correlation learning without compromising in-distribution accuracy.

## 2. Related Work

**Spurious correlation learning.** Spurious correlations in supervised learning are a well-established failure mode (Singla & Feizi, 2022). Models trained via empirical risk minimization (ERM) often exploit surface-level features that correlate with labels in the training distribution but lack a causal relationship to the target task, a phenomenon referred to as shortcut learning (Geirhos et al., 2020), simplicity bias (Shah et al., 2020; Morwani et al., 2023), or spurious feature reliance (Arjovsky et al., 2019). When spurious correlations shift at deployment (Zhou et al., 2021), models suffer prediction errors (Sagawa et al., 2020), biased outcomes (Geirhos et al., 2018), and performance degradation (Xiao et al., 2021). Proposed mitigations include data augmentation (Chang et al., 2021; Plumb et al., 2022), reweighting of minority examples (Liu et al., 2021), and modified training dynamics (Izmailov et al., 2022; Kirichenko et al., 2023), though these approaches often require domain knowledge or explicit annotation of spurious features. Theoretically, spurious learning has been analyzed through optimization dynamics, where gradient descent preferentially fits easier features early in training, leading to gradient starvation (Rahaman et al., 2019; Kalimeris et al., 2019; Qiu et al., 2024), as well as through NTK and linearized analyses that characterize implicit biases (Pezeshki et al., 2021; Hermann et al., 2024). Most closely related to our setting, (Bombari & Mondelli, 2025) study high-dimensional linear models under ERM, deriving closed-form solutions that reveal how data covariance induces spurious feature reliance. However, these theories assume pointwise loss landscapes and do not extend to preference optimization, where pairwise comparisons induce fundamentally different learning dynamics.

**Empirical failures in preference optimization.** Prior work on preference optimization has documented spurious correlation learning primarily through empirical observations of reward-hacking behaviors. Studies show that RLHF and DPO models exploit surface-level artifacts, such as verbosity bias (preferring longer responses independently of quality (Singhal et al., 2024; Saito et al., 2023)), sycophancy (agreeing with user beliefs to maximize reward (Sharma et al., 2024)), and formatting bias (over-optimizing for numbered lists or stylistic markers (Zhang et al., 2025)). Existing mitigations target individual symptoms through ad-hoc interventions, including length penalties (Park et al., 2024) for verbosity or synthetic data filtering (Chen et al., 2024). These approaches treat each bias in isolation without addressing the underlying learning dynamics that produce them. In contrast, we provide a population-level analysis

that reveals the structural mechanisms driving these empirically observed failures.

**Theoretical analysis of preference optimization.** Theory for preference optimization has developed along several lines. Work on linear contextual dueling bandits and dueling reinforcement learning establishes regret minimization under realizable reward assumptions (Dudík et al., 2015; Saha et al., 2023). Recent alignment work develops robust or safety-motivated analysis and training procedures, including robust formulations (Xiong et al., 2024; Wu et al., 2025), noise-aware losses (Chowdhury et al., 2024a), privacy-preserving constraints (Chen et al., 2025; Zhou et al., 2025), and divergence-based alignment objectives that explicitly separate preferred and rejected behaviors (Haldar et al., 2025). Complementary analyses in linear or log-linear regimes motivate simplified preference models and analyze learning behavior under idealized assumptions (Zhu et al., 2023; Chowdhury et al., 2024b; Zhou et al., 2025). However, across these lines of work, a critical assumption persists: that the learned feature representation is valid for the target task. These approaches address stochastic or adversarial failures through algorithmic modifications but do not characterize systematic spurious correlation learning.

## 3. Preliminaries

### 3.1. Preference Learning Setup

**Preference dataset.** We consider a preference dataset $\mathcal{D} = \{(x^{(i)}, y_w^{(i)}, y_l^{(i)})\}_{i=1}^N$, where $(y_w^{(i)}, y_l^{(i)})$ denotes the human-preferred and rejected responses to prompt $x^{(i)} \in \mathcal{X}$, following standard pairwise preference supervision (Rafailov et al., 2023; Christiano et al., 2017). We represent each prompt–response pair $(x, y)$ with a feature vector $\phi(x, y) \in \mathbb{R}^d$ and train on the feature difference $\Delta\phi = \phi(x, y_w) - \phi(x, y_l)$.

**Causal and spurious feature decomposition.** We decompose each feature vector as $\phi(x, y) = [\phi_c(x, y); \phi_s(x, y)] \in \mathbb{R}^{d_c + d_s}$. The causal component $\phi_c(x, y) \in \mathbb{R}^{d_c}$ captures true response utility, while the spurious component $\phi_s(x, y) \in \mathbb{R}^{d_s}$ correlates with observed preferences in the training data through statistics that need not persist at deployment. This decomposition induces a corresponding split of feature differences, $\Delta\phi = [\Delta\phi_c; \Delta\phi_s]$. In practice, spurious features arise from data collection biases and domain-specific structure, such as annotators preferring longer or more formal responses at fixed content quality. We call $\phi_s$ spurious not because it cannot influence annotators but because, by design, these correlations should not determine policy behavior at test time. We formalize this distinction through the following invariance assumption.

**Assumption 3.1** (Invariance). Under interventions that modify spurious features while holding causal features fixed, human preferences remain unchanged.

### 3.2. Log-Linear Policy and Data Generation

**Policy model.** We adopt a log-linear policy, a common regime for theoretical analysis that enables tractable characterization of learning dynamics (Zhu et al., 2023; Zhou et al., 2025). The policy takes the form $\pi_\theta(y \mid x) \propto \exp(\theta^\top \phi(x, y))$, where $\theta \in \mathbb{R}^d$ denotes the learnable parameter vector. We decompose the parameter vector as $\theta = [\theta_c; \theta_s] \in \mathbb{R}^{d_c + d_s}$ to match the causal–spurious feature split.

**Data generation.** Our analysis depends only on the feature differences $\Delta\phi$ and their stated moment conditions. We denote by $\theta^\dagger$ the ground-truth parameter encoding true utility. By Assumption 3.1, spurious features do not affect human preferences, so $\theta^\dagger = [\theta_c^\dagger; \mathbf{0}]$.

### 3.3. Direct Preference Optimization

We analyze Direct Preference Optimization (DPO) (Rafailov et al., 2023) as a method for fitting the preference model described above. DPO optimizes the policy relative to a reference policy $\pi_{\theta_{\mathrm{ref}}}$, so we express learning in terms of the deviation $\tilde{\theta} = \theta - \theta_{\mathrm{ref}}$. The DPO loss for a single preference example $(x, y_w, y_l)$ is

$$\ell_{\mathrm{DPO}}(\theta; x, y_w, y_l) = -\log \sigma\left(\beta\, \tilde{\theta}^\top \Delta\phi\right), \qquad (1)$$

where $\beta > 0$ controls the KL regularization strength. Equation (1) shows that, under the log-linear parameterization, DPO reduces to logistic regression on feature differences. While this reduction provides a well-defined population objective, the sigmoid nonlinearity generally prevents closed-form characterization of the population optimum.

**Linearization regime.** To enable analytical progress, we work in a local regime where score differences remain moderate, allowing a first-order Taylor expansion of the sigmoid. Formally:

**Assumption 3.2** (Local regime). With high probability under the data distribution, $|\beta\, \tilde{\theta}^\top \Delta\phi| \ll 1$.

This regime arises near initialization, for bounded feature magnitudes, or when the scaled DPO margin $\beta\tilde{\theta}^\top \Delta\phi$ remains small. Under this linearization, the population optimum admits a closed-form solution that reveals how data structure drives spurious learning.

## 4. Population-Level Spurious Learning

In this section, we analyze the population equilibrium of the DPO objective under a local linearization. We show that spurious correlation learning arises generically: mean spurious bias or causal–spurious correlation in the training data leads to nonzero spurious parameters at the population optimum.

## 4.1. Population Gradient and Early-Training Drift

We study the population objective

$$L(\theta) := -\mathbb{E}\Big[\log \sigma\big(\beta\, \tilde{\theta}^\top \Delta\phi\big)\Big],$$

where the expectation is taken over preference data generated using the ground truth parameters $\theta^\dagger$. A single preference pair induces the gradient $\nabla_\theta \ell(\theta; x, y_w, y_l) = -\beta\, w(\Delta_\theta)\, \Delta\phi$, where $\Delta_\theta := \tilde{\theta}^\top \Delta\phi$ denotes the predicted score difference and $w(\Delta_\theta) := 1 - \sigma(\beta\Delta_\theta) \in (0,1)$. Hard or misclassified pairs receive larger weight $w(\Delta_\theta)$, while confidently satisfied preferences are downweighted. Taking expectations, the population gradient is

$$\nabla_\theta L(\theta) = -\beta\, \mathbb{E}\big[w(\Delta_\theta)\Delta\phi\big].$$

**Early-training drift.** Near the reference policy ($\tilde{\theta} \approx 0$), score differences are small, so $w(\Delta_\theta) \approx \frac{1}{2}$. The gradient simplifies to

$$\nabla_\theta L(\theta) \approx -\frac{\beta}{2}\, \mu, \qquad \mu := \mathbb{E}[\Delta\phi] = \begin{bmatrix} \mu_c \\ \mu_s \end{bmatrix},$$

where $\mu_c := \mathbb{E}[\Delta\phi_c]$ and $\mu_s := \mathbb{E}[\Delta\phi_s]$ are the mean feature differences. Whenever $\mu_s \neq 0$, spurious features are learned from the first gradient step: $\nabla_{\theta_s} L(\theta) \approx -\frac{\beta}{2}\mu_s$. This shows that spurious features with nonzero mean differences are immediately learned, moving the DPO update in spurious directions.

Early-training drift shows that mean spurious bias drives immediate spurious learning. However, this does not guarantee spurious parameters remain nonzero at equilibrium. The gradient dynamics could drive them back to zero. We now characterize the population equilibrium to determine when spurious learning persists.

## 4.2. Linearized Equilibrium

Under Assumption 3.2, we linearize the weighting function $w(\Delta_\theta) \approx \frac{1}{2} - \frac{\beta}{4}\Delta_\theta$ and substitute into the population gradient. This yields (see Appendix A.1 for details)

$$\nabla_\theta L(\theta) \approx -\frac{\beta}{2}\mu + \frac{\beta^2}{4}\Sigma\tilde{\theta}, \quad \Sigma := \mathbb{E}[\Delta\phi\Delta\phi^\top]. \quad (2)$$

At the linearized stationary point $\theta^\star$, the gradient vanishes:

$$\Sigma\,\tilde{\theta}^\star = \frac{2}{\beta}\,\mu, \qquad \tilde{\theta}^\star := \theta^\star - \theta_{\mathrm{ref}}.$$

Partitioning according to the causal-spurious split, we write

$$\Sigma = \begin{bmatrix} \Sigma_{cc} & \Sigma_{cs} \\ \Sigma_{sc} & \Sigma_{ss} \end{bmatrix}, \qquad \mu = \begin{bmatrix} \mu_c \\ \mu_s \end{bmatrix}, \qquad \tilde{\theta}^\star = \begin{bmatrix} \tilde{\theta}_c^\star \\ \tilde{\theta}_s^\star \end{bmatrix},$$

where $\Sigma_{cc} := \mathbb{E}[\Delta\phi_c\Delta\phi_c^\top]$, $\Sigma_{ss} := \mathbb{E}[\Delta\phi_s\Delta\phi_s^\top]$, and $\Sigma_{cs} = \Sigma_{sc}^\top := \mathbb{E}[\Delta\phi_c\Delta\phi_s^\top]$ captures causal-spurious correlation. The equilibrium conditions are

$$\Sigma_{cc}\tilde{\theta}_c^\star + \Sigma_{cs}\tilde{\theta}_s^\star = \frac{2}{\beta}\mu_c, \qquad (3)$$

$$\Sigma_{sc}\tilde{\theta}_c^\star + \Sigma_{ss}\tilde{\theta}_s^\star = \frac{2}{\beta}\mu_s. \qquad (4)$$

**Explicit solution via Schur complement.** The equilibrium conditions (3)–(4) couple causal and spurious parameters through the cross-covariance $\Sigma_{sc}$. To isolate the spurious component, we use the Schur complement method. Assume $\Sigma_{ss}$ and the Schur complement $S_c := \Sigma_{cc} - \Sigma_{cs}\Sigma_{ss}^{-1}\Sigma_{sc}$ are invertible. These conditions hold when $\Sigma \succ 0$. Under these assumptions, the spurious component at equilibrium is

$$\tilde{\theta}_s^\star = \frac{2}{\beta}\Sigma_{ss}^{-1}\Big[\underbrace{(I + \Sigma_{sc}S_c^{-1}\Sigma_{cs}\Sigma_{ss}^{-1})\mu_s}_{\text{mean-bias}} - \underbrace{\Sigma_{sc}S_c^{-1}\mu_c}_{\text{corr.-leakage}}\Big].$$
(5)

The full derivation is provided in Appendix A.2.

## 4.3. Main Mechanism Result

We now state our main result characterizing spurious learning at the population equilibrium.

**Theorem 4.1** (Spurious learning from mean bias and correlation leakage)**.** *Under the linearized population dynamics (Equation (2)), assume $\Sigma = \mathbb{E}[\Delta\phi\Delta\phi^\top] \succ 0$. If $\mu_s \neq 0$ (mean spurious bias) or $\Sigma_{sc} \neq 0$ (causal-spurious correlation), then generically $\tilde{\theta}_s^\star \neq 0$. That is, the population optimum assigns nonzero weight to spurious features.*

Equation (5) reveals two distinct mechanisms:

*(i) Mean spurious bias ($\mu_s \neq 0$):* When spurious features are asymmetrically distributed across preference pairs, they directly contribute to $\tilde{\theta}_s^\star$, even in the absence of correlation ($\Sigma_{sc} = 0$).

*(ii) Correlation leakage ($\Sigma_{sc} \neq 0$):* When spurious features correlate with causal features, weight intended for causal directions leaks into spurious ones. This operates even when spurious features are unbiased ($\mu_s = 0$).

**Deviation from ground truth.** By Assumption 3.1, spurious features should not determine test-time behavior. Theorem 4.1 shows that the learned DPO update nevertheless satisfies $\tilde{\theta}_s^\star \neq 0$ under generic conditions on the training distribution. This creates a systematic bias: the learned policy deviates from the true preference model in directions that do not affect utility. Whether this deviation causes deployment failures depends on how spurious feature statistics differ between training and deployment, which we formalize in Section 5.

# 5. Spurious Learning Deployment Error

Section 4 showed that learned parameters generically satisfy $\tilde{\theta}_s^\star \neq 0$. We now study the consequences for deployment. Spurious learning creates a *potential vulnerability*: learned parameters depend on features that do not affect true utility. Whether this vulnerability translates into deployment error depends on how spurious statistics shift between training and deployment. We show that when spurious statistics shift, the population objective becomes sensitive to learned spurious parameters, with degradation occurring when the shift has the harmful sign. We then decompose deployment suboptimality into a shift component and an estimation component, showing that the former persists regardless of training set size.

## 5.1. Distribution Shift Setup

Let $P$ denote the training distribution and $Q$ the deployment distribution over preference pairs $(x, y_w, y_l)$. These distributions may differ in their feature statistics. We denote:

$$\text{Training:} \quad \mu^{(P)} = \mathbb{E}_P[\Delta\phi], \quad \Sigma^{(P)} = \mathbb{E}_P[\Delta\phi\Delta\phi^\top]$$
$$\text{Deployment:} \quad \mu^{(Q)} = \mathbb{E}_Q[\Delta\phi], \quad \Sigma^{(Q)} = \mathbb{E}_Q[\Delta\phi\Delta\phi^\top]$$

The model learns parameters $\theta_{\text{train}}$ by optimizing on $P$ and deploys with these fixed parameters on $Q$. Differences between $\mu_s^{(P)}$ and $\mu_s^{(Q)}$ can induce deployment error through the learned spurious parameters $\tilde{\theta}_{s,\text{train}}$. We defer the full mathematical characterization of canonical shift scenarios (suppression, adversarial, and rotation) to Appendix E.

## 5.2. Expected Margin as Deployment Metric

To assess how shifts affect performance, we use the expected preference margin, which provides a tractable first-order characterization of model quality under the local regime.

**Margin definition.** The expected margin measures the average score gap between preferred and dispreferred responses: $m_D(\theta) := \mathbb{E}_{(x,y_w,y_l)\sim D}[\beta\,\tilde{\theta}^\top\Delta\phi]$. The margin decomposes into causal and spurious components:

$$m_D(\theta) = \underbrace{\beta\,\tilde{\theta}_c^\top\mu_c^{(D)}}_{=:m_D^{\text{causal}}(\theta)} + \underbrace{\beta\,\tilde{\theta}_s^\top\mu_s^{(D)}}_{=:m_D^{\text{spurious}}(\theta)}. \tag{6}$$

We now show that margin differences provide a first-order approximation to objective differences.

**Proposition 5.1** (First-order margin approximation). *Under the local regime (Assumption 3.2), the DPO objective $\tilde{J}^{(D)}(\theta) := \mathbb{E}_D[\log\sigma(\beta\tilde{\theta}^\top\Delta\phi)]$ satisfies*

$$\tilde{J}^{(Q)}(\theta) - \tilde{J}^{(P)}(\theta) = \frac{1}{2}\big(m_Q(\theta) - m_P(\theta)\big) + O(\beta^2\|\tilde{\theta}\|^2).$$

The proof follows from Taylor expansion of the log-sigmoid; see Appendix C.1.

Proposition 5.1 shows that margin differences drive objective changes. We now isolate the spurious component by considering shifts that preserve causal statistics.

**Proposition 5.2** (Spurious margin drives shift). *When causal statistics are stable ($\mu_c^{(Q)} = \mu_c^{(P)}$), the objective difference at $\theta_{\text{train}}$ is governed by the spurious margin:*

$$\tilde{J}^{(Q)}(\theta_{\text{train}}) - \tilde{J}^{(P)}(\theta_{\text{train}}) \approx \frac{\beta}{2}\tilde{\theta}_{s,\text{train}}^\top(\mu_s^{(Q)} - \mu_s^{(P)}),$$

*up to an $O(\beta^2\|\tilde{\theta}_{\text{train}}\|^2)$ remainder.*

*Proof.* Apply Proposition 5.1 with $\theta = \theta_{\text{train}}$. From (6), the margin difference decomposes as $m_Q(\theta_{\text{train}}) - m_P(\theta_{\text{train}}) = \beta\tilde{\theta}_{c,\text{train}}^\top(\mu_c^{(Q)} - \mu_c^{(P)}) + \beta\tilde{\theta}_{s,\text{train}}^\top(\mu_s^{(Q)} - \mu_s^{(P)})$. When $\mu_c^{(Q)} = \mu_c^{(P)}$, the causal term vanishes. $\square$

Proposition 5.2 identifies when deployment performance becomes sensitive to spurious statistics: when learned spurious parameters $\tilde{\theta}_{s,\text{train}}$ interact with shifts in spurious statistics $\mu_s^{(Q)} - \mu_s^{(P)}$. To first order, degradation occurs when this interaction has the harmful sign. When spurious statistics remain stable ($\mu_s^{(Q)} = \mu_s^{(P)}$), spurious learning is benign despite $\tilde{\theta}_{s,\text{train}} \neq 0$. We now show that this population-level vulnerability is irreducible with respect to sample size by decomposing deployment suboptimality.

## 5.3. Suboptimality Decomposition

We now formalize the irreducibility of deployment degradation by decomposing suboptimality into a shift term and an estimation term. Let $\theta_Q^\star := \arg\max_{\theta\in\Theta_B}\tilde{J}^{(Q)}(\theta)$ denote the deployment-optimal parameters, where $\Theta_B := \{\theta \in \mathbb{R}^d : \|\theta - \theta_{\text{ref}}\|_2 \leq B\}$. Let $\hat{\theta}$ denote the finite-sample estimator trained on $n$ samples from $P$. We define deployment suboptimality as

$$\text{SubOpt}_Q(\hat{\theta}) := \tilde{J}^{(Q)}(\theta_Q^\star) - \tilde{J}^{(Q)}(\hat{\theta}).$$

Inserting the population training optimum $\theta_{\text{train}}$ yields:

$$\text{SubOpt}_Q(\hat{\theta}) = \underbrace{\tilde{J}^{(Q)}(\theta_Q^\star) - \tilde{J}^{(Q)}(\theta_{\text{train}})}_{\text{shift term}}$$
$$+ \underbrace{\tilde{J}^{(Q)}(\theta_{\text{train}}) - \tilde{J}^{(Q)}(\hat{\theta})}_{\text{estimation term}}. \tag{7}$$

**Irreducible shift vs. reducible estimation.** The decomposition separates two sources of suboptimality. When $\hat{\theta}$ is consistent for the population training optimum $\theta_{\text{train}}$, the *estimation term* vanishes as $n \to \infty$. In contrast, the *shift term* is determined by the population optimum $\theta_{\text{train}}$ and persists regardless of sample size:

$$\lim_{n\to\infty}\text{SubOpt}_Q(\hat{\theta}) = \tilde{J}^{(Q)}(\theta_Q^\star) - \tilde{J}^{(Q)}(\theta_{\text{train}}).$$

By Proposition 5.2, when causal statistics are stable, the persistent vulnerability is driven by the interaction between $\tilde{\theta}_{s,\text{train}}$ and $\mu_s^{(Q)} - \mu_s^{(P)}$. The magnitude depends on two factors: the learned spurious parameters $\tilde{\theta}_{s,\text{train}}$ (which Section 4 showed emerge structurally from training) and the spurious shift $\mu_s^{(Q)} - \mu_s^{(P)}$ (which depends on the deployment environment). Collecting more data from $P$ cannot eliminate this vulnerability, it only reduces the vanishing estimation component while leaving $\tilde{\theta}_{s,\text{train}}$ unchanged.

### 5.4. Main Deployment Bound

The margin analysis in Section 5.2 characterized deployment degradation at the population level, assuming access to $\theta_{\text{train}}$. In practice, we learn from finite samples, obtaining an estimator $\hat{\theta}$ that deviates from $\theta_{\text{train}}$. We now bound the estimation term $\tilde{J}^{(Q)}(\theta_{\text{train}}) - \tilde{J}^{(Q)}(\hat{\theta})$ to complete the decomposition in Equation (7), confirming that the irreducible shift dominates as $n \to \infty$.

**Technical assumptions.** We require: (A1) bounded features, $\|\Delta\phi\|_2 \leq 1$ almost surely; (A2) bounded parameters, $\|\tilde{\theta}\|_2 \leq B$ for all $\theta \in \Theta_B$; (A3) local regime (Assumption 3.2); and (A4) geometry transfer, meaning there exists $\kappa_\Pi \geq 1$ such that $\|v\|_{\Sigma^{(Q)}}^2 \leq \kappa_\Pi \|v\|_{H_P}^2$ for all $v \in \mathbb{R}^d$, where $H_P := -\nabla^2 \tilde{J}^{(P)}(\theta_{\text{train}}) + \lambda I$ is the positive definite local curvature of the training objective at $\theta_{\text{train}}$. Assumption (A4) bounds deployment variation by training curvature and holds when $P$ and $Q$ are not too different. See Appendix C for detailed discussion of these conditions.

We define $\hat{\theta}$ as the ridge-regularized MLE:

$$\hat{\theta} := \arg\max_{\theta \in \Theta_B} \left[ \frac{1}{n} \sum_{i=1}^{n} \log \sigma(\beta \tilde{\theta}^\top \Delta\phi^{(i)}) - \frac{\lambda}{2} \|\tilde{\theta}\|_2^2 \right],$$

where $\lambda > 0$ is the regularization parameter.

**Theorem 5.3** (Deployment sub-optimality bound). *Under assumptions (A1)–(A4), with probability at least $1 - \delta$:*

$$\text{SubOpt}_Q(\hat{\theta}) \leq \underbrace{\tilde{J}^{(Q)}(\theta_Q^\star) - \tilde{J}^{(Q)}(\theta_{\text{train}})}_{\text{shift term}} + \underbrace{\kappa_{\text{est}} \Gamma_n}_{\text{estimation term}},$$

(8)

*where $\kappa_{\text{est}} := G_Q + \kappa_\Pi \Gamma_n / 2$, $G_Q := \|\nabla \tilde{J}^{(Q)}(\theta_{\text{train}})\|_{H_P^{-1}}$, and $\Gamma_n := 2\beta\sqrt{2(d + \log(1/\delta))/n} + B\sqrt{\lambda}$.*

*Proof sketch.* The decomposition (7) holds by definition. For the estimation term, we expand $\tilde{J}^{(Q)}$ around $\theta_{\text{train}}$. The linear term is bounded by $G_Q \|\hat{\theta} - \theta_{\text{train}}\|_{H_P}$, while the quadratic term is controlled by Assumption (A4). Concentration gives $\|\hat{\theta} - \theta_{\text{train}}\|_{H_P} \leq \Gamma_n$. Full details are in Appendix C.8. □

**Interpretation.** Theorem 5.3 reveals the structure of deployment suboptimality. The estimation term decreases with more training data and vanishes in the large-sample limit (up to the $O(B\sqrt{\lambda})$ regularized bias). The shift term persists: even with infinite training data, deployment error remains bounded by the gap between $\theta_{\text{train}}$ and $\theta_Q^\star$.

By Proposition 5.2, when causal statistics are stable, learned spurious parameters make the deployment objective sensitive to shifted spurious statistics. This decomposition clarifies two sources of vulnerability: Section 4 showed that $\tilde{\theta}_{s,\text{train}} \neq 0$ arises structurally from the training distribution, while the deployment shift $\mu_s^{(Q)} - \mu_s^{(P)}$ depends on the environment. Spurious learning creates *latent vulnerability*; harm materializes only when spurious statistics shift. Crucially, scaling training data cannot eliminate this vulnerability, as it leaves $\tilde{\theta}_{s,\text{train}}$ unchanged.

## 6. Tie Training Reduces Spurious Reliance

Section 5 showed that deployment error contains an irreducible shift term driven by learned spurious parameters $\tilde{\theta}_{s,\text{train}}$. We now describe a simple data-level intervention that directly targets this mechanism. We introduce *tie training*, a data augmentation strategy that reduces spurious reliance by adding curvature selectively in spurious directions. The approach constructs preference pairs with equal utility but differing spurious features, assigns labels randomly, and mixes these ties with standard preference data during training.

### 6.1. Tie Construction and Training

**Definition 6.1** (Tie pair). A tie pair is a tuple $(x, y_A, y_B)$ with equal utility: $u^\dagger(x, y_A) = u^\dagger(x, y_B)$.

We construct ties so that causal features match while spurious features differ: $\Delta\phi_c = 0$ and $\Delta\phi_s = \delta_s \neq 0$. For each tie, we assign the winner-loser label uniformly at random. Thus, ties enter the standard DPO loss as hard-labeled pairs and require no change to the objective. The random orientation gives $\mathbb{E}_{\text{tie}}[\Delta\phi] = 0$ and

$$\Sigma^{\text{tie}} := \mathbb{E}_{\text{tie}}[\Delta\phi\Delta\phi^\top] = \begin{bmatrix} 0 & 0 \\ 0 & \Sigma_{ss}^{\text{tie}} \end{bmatrix}.$$

This covariance structure ensures ties add curvature only in spurious directions.

**Mixed training.** We train on a mixture $P_{\text{mix}} := \alpha P + (1 - \alpha)P_{\text{tie}}$ with $\alpha \in (0, 1)$. Under the local regime (Assumption 3.2), the linearized equilibrium satisfies

$$\tilde{\theta}_{\text{mix}}^\star = \frac{2\alpha}{\beta}\left(\Sigma^{\text{mix}}\right)^{-1}\mu^{(P)}, \quad \Sigma^{\text{mix}} := \alpha\Sigma^{(P)} + (1-\alpha)\Sigma^{\text{tie}}.$$

(9)

Since $\Sigma^{\text{tie}}$ has support only on spurious coordinates, mixed training increases the spurious curvature while scaling the raw causal–spurious second-moment by $\alpha$. This shrinks

the spurious component relative to strict-only training. Full derivation is in Appendix D.4.

## 6.2. Main Result

We now formalize the effect of tie training on spurious reliance and deployment performance.

**Theorem 6.2** (Tie training reduces spurious reliance and deployment shift). *Under the conditions of Theorem 5.3, the tie construction above, and a regularity condition made explicit in Appendix D.5:*

*(i) Spurious shrinkage. Let $\theta^\star$ denote the strict-only population optimizer and $\theta^\star_{\mathrm{mix}}$ the mixed-training optimizer. Then*

$$\|\tilde{\theta}^\star_{s,\mathrm{mix}}\|_{\Sigma^{\mathrm{mix}}_{ss}} \ \leq \ \|\tilde{\theta}^\star_s\|_{\Sigma^{(P)}_{ss}},$$

*where $\|v\|_A := \sqrt{v^\top A v}$. The inequality is strict when tie training adds curvature along active spurious directions.*

*(ii) Shift reduction. If $\mu^{(Q)}_c = \mu^{(P)}_c$, the first-order deployment shift is bounded by the strict-only worst-case envelope: $|\tilde{\theta}^{\star\top}_{s,\mathrm{mix}}\delta\mu_s| \leq \|\tilde{\theta}^\star_s\|_{\Sigma^{(P)}_{ss}} \cdot \|\delta\mu_s\|_{(\Sigma^{(P)}_{ss})^{-1}}$, where $\delta\mu_s := \mu^{(Q)}_s - \mu^{(P)}_s$. In the scalar case ($d_s = 1$), this strengthens to $|\tilde{\theta}^\star_{s,\mathrm{mix}}\delta\mu_s| \leq |\tilde{\theta}^\star_s \delta\mu_s|$.*

*(iii) Finite-sample bound. Let $\hat{\theta}^{\mathrm{mix}}$ be trained on $n$ samples from $P_{\mathrm{mix}}$. With probability at least $1 - \delta$,*

$$\mathrm{SubOpt}_Q(\hat{\theta}^{\mathrm{mix}}) \leq \underbrace{\tilde{J}^{(Q)}(\theta^\star_Q) - \tilde{J}^{(Q)}(\theta^\star_{\mathrm{mix}})}_{\text{reduced shift}} + \underbrace{\kappa^{\mathrm{mix}}_{\mathrm{est}}\Gamma_{n,\mathrm{mix}}}_{\text{estimation}},$$

*where $\kappa^{\mathrm{mix}}_{\mathrm{est}} := G_{Q,\mathrm{mix}} + \kappa_\Pi\Gamma_{n,\mathrm{mix}}/2$ and $G_{Q,\mathrm{mix}} := \|\nabla\tilde{J}^{(Q)}(\theta^\star_{\mathrm{mix}})\|_{H^{-1}_{P_{\mathrm{mix}}}}.$*

The proof combines the equilibrium characterization (Equation (9)) with the deployment bound framework; see Appendix D.3.

**Corollary 6.3** (Quantitative reduction under isotropic ties). *If $\Sigma^{\mathrm{tie}}_{ss} = \sigma^2 I_{d_s}$ and $\Sigma^{(P)}_{ss} = \lambda_0 I_{d_s}$, then*

$$\frac{\|\tilde{\theta}^\star_{s,\mathrm{mix}}\|_2}{\|\tilde{\theta}^\star_s\|_2} \leq \frac{\alpha\lambda_0}{\alpha\lambda_0 + (1-\alpha)\sigma^2}.$$

**Interpretation.** Theorem 6.2 shows that tie training directly targets the irreducible shift by shrinking spurious parameters through selective regularization. Part (i) establishes spurious weight reduction at the population level. Part (ii) bounds the deployment shift contribution, with pointwise reduction in the scalar case. Part (iii) provides finite-sample guarantees. Corollary 6.3 gives an explicit reduction as a function of $\alpha$.

## 6.3. Practical Considerations

We discuss three practical aspects of tie training next. **Soft ties:** Exact utility equality is not required; near-ties with $|u^\dagger(x, y_A) - u^\dagger(x, y_B)| \leq \varepsilon$ yield similar effects. **Random labeling:** Using both $(y_A \succ y_B)$ and $(y_B \succ y_A)$ with equal probability ensures $\mathbb{E}_{\mathrm{tie}}[\Delta\phi] = 0$; single-direction labeling injects bias. **Tie selection:** Systematic tie construction at scale remains an open problem; our experiments (Section 7) use manual construction or simple heuristics based on known spurious features.

# 7. Experiments

We validate three theoretical predictions: (i) preference optimization learns spurious correlations (Theorem 4.1), (ii) deployment error under spurious shift is irreducible (Theorem 5.3), and (iii) tie training reduces this error (Corollary 6.3). We evaluate these claims across three regimes of increasing realism: linear models, neural networks, and large language models. Code is available at https://github.com/cmoyacal/tie-training.

## 7.1. Linear Models: Quantitative Validation

**Setup.** We generate Gaussian features $\phi = [\phi_c; \phi_s]$ with $d_c = 5$ causal and $d_s = 5$ spurious dimensions. Preference labels are generated by a Bradley–Terry logistic teacher. Spurious features correlate with preferences under training distribution $P$ and shift at deployment $Q$. Full details are in Appendix F.1.

**Results.** Figure 1 validates all three theoretical predictions. Panel (a) shows that learned spurious parameters $\|\hat{\theta}_s\|$ match the closed-form prediction of spurious parameters (Theorem 4.1). When the local regime assumption weakens and our quantitative predictions are no longer valid, we find empirically that including second-order curvature terms restores agreement (bottom panel). Panel (b-top) plots deployment suboptimality versus sample size $n$. The estimation error decays as $n \to \infty$, while the shift term plateaus, confirming irreducibility (Theorem 5.3). Panel (b-bottom) reports the tie-induced reduction ratio $\|\theta^{\mathrm{mix}}_s\|/\|\theta^{\mathrm{strict}}_s\|$ across mixing fractions $\alpha$. Empirical curves match the theoretical reduction factor (Corollary 6.3) for $\beta = 0.3$ and ratio $\sigma^2/\lambda_0 = 5$. These results confirm the population equilibrium characterization (Section 4), the irreducibility decomposition (Section 5), and the tie training guarantees (Section 6).

## 7.2. Neural Networks: Qualitative Persistence

Neural networks hide the causal–spurious decomposition inside nonlinear representations, violating linear assumptions. We test whether the same mechanisms persist qualitatively.

**Setup.** We sample latent causal and spurious features, apply a random nonlinear mixing $\phi = g([\phi_c; \phi_s])$, and train an MLP scorer with DPO. The causal–spurious decomposition is hidden from the model. Details are in Appendix F.2.

**(a)** **(b)**

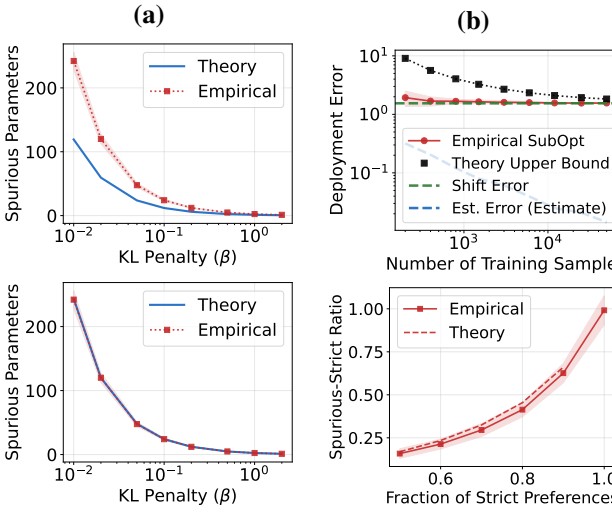

*Figure 1.* Linear theory validation. **(a)** Learned spurious parameters norm against theoretical prediction (Theorem 4.1) (top); second-order corrections restore agreement when local regime is violated (bottom). **(b)** Deployment suboptimality decomposition: estimation error decays as $n \to \infty$, while shift error persists, demonstrating irreducibility. As predicted by Theorem 5.3, empirical deployment error $\text{SubOpt}_Q$ remains bounded as the number of training samples increases (top); tie reduction ratio satisfies the theoretical formula (Corollary 6.3) across the fraction of strict preferences $\alpha$ and the spurious variance ratio $\sigma^2/\lambda_0 = 5$ (bottom).

**Proxy metrics.** Since $\|\theta_s\|$ is not observable, we measure spurious reliance using (i) the *spurious gap*, $\text{Acc}_{\text{aligned}} - \text{Acc}_{\text{misaligned}}$, where the two terms denote pairwise preference accuracy when spurious features align with or oppose true utility, respectively, and (ii) *adversarial accuracy*, the pairwise preference accuracy under reversed spurious correlation.

**Results.** Strict training learns spurious reliance, yielding spurious gap $\approx 0.7$ and adversarial accuracy $\approx 0.25$. Tie training ($\alpha = 0.75$) reduces the spurious gap to $\approx 0.45$ and improves adversarial accuracy to $\approx 0.7$, while preserving in-distribution accuracy. Figure 2 shows that spurious gap decreases monotonically with tie mixing fraction $\alpha$, and tie training breaks the adversarial accuracy plateau observed under strict training as sample size increases. Consistent with Theorems 5.3 and 6.2, tie training has limited visible effect at small $n$, where estimation error dominates, but its advantage emerges as estimation error decays and shift error remains. Although the theory does not apply exactly, the qualitative behavior matches the linear predictions.

### 7.3. LLMs: Synthetic Hotel Preferences

**Dataset.** We construct a controlled hotel preference benchmark. Causal attributes affecting utility include price, distance, and rating. Spurious attributes (not causally affecting utility) include building age, renovation year, chain tier,

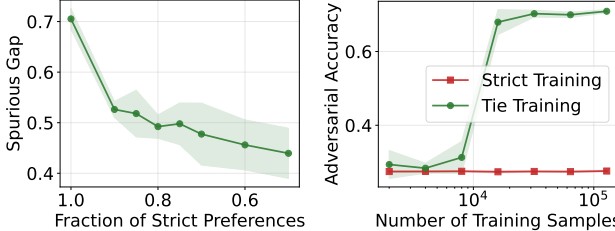

*Figure 2.* Neural network validation. **Left:** Spurious gap (accuracy on aligned minus misaligned spurious conditions) decreases monotonically with tie mixing fraction $\alpha$. **Right:** Strict training exhibits a persistent adversarial accuracy plateau despite increasing data; tie training (with $\alpha = 0.75$) breaks this plateau, improving robustness from $\approx 0.25$ to $\approx 0.7$.

*Table 1.* Informative tie training (Tie-I) improves robustness under spurious shift without sacrificing in-distribution accuracy.

| Method | In-Distr. ($P$) | Sup. ($Q_{\text{sup}}$) | Adv. ($Q_{\text{adv}}$) |
|--------|-----------------|-------------------------|--------------------------|
| Strict | 92.25% | 74.00% | 64.20% |
| Tie-I  | 92.40% | 82.85% | 86.70% |

lobby size, and employee count. These correlate with utility during training but do not affect true quality. Full details are in Appendix F.3.

**Tie construction.** Informative ties satisfy three properties: (i) near-equal utility ($|u_A - u_B| < \tau$), (ii) strong spurious contrast (maximal $|\Delta\phi_s|$), and (iii) random labels. As an ablation, in Appendix F.3, we also evaluate non-informative ties that use monotonic spurious assignment correlated with utility, providing weak regularization signal.

**Results.** Standard DPO achieves 92% in-distribution accuracy but degrades to 74% (suppressed correlation) and 64% (adversarial correlation) under spurious shift. Informative tie training maintains 92% in-distribution while improving to 83% (suppressed) and 87% (adversarial). Table 1 reports the results.

## 8. Discussion

**On our assumptions.** Our closed-form results on spurious reliance (Section 4), irreducible deployment error (Section 5), and tie training (Section 6) hold in the local regime (Assumption 3.2) for log-linear policies. Within this regime, the analysis identifies mean spurious bias and causal–spurious leakage as the mechanisms behind spurious learning, links them to irreducible deployment error, and motivates tie training as a targeted intervention.

Linear analysis is standard in the shortcut- and spurious-learning literature for isolating shift-induced failure modes. For example, (Nagarajan et al., 2021) use linear models to characterize spurious-feature failure under distribution shift

in classification. Here, we study the corresponding question for pairwise preference optimization. To make the regime boundary explicit, we report the diagnostic $|\beta\tilde{\theta}^\top\Delta\phi|$ in our log-linear experiments (Appendix F.1), confirming the parameter ranges where the local approximation holds.

Beyond these assumptions, the same qualitative pattern appears in neural network and LLM experiments outside the local regime: spurious reliance emerges under correlated training data, additional in-distribution data fails to remove the resulting deployment error, and tie training reduces it. Thus, the theory remains useful as a diagnostic guide for designing the intervention.

Finally, the finite-sample deployment result also invokes boundedness and a transfer condition (Section 5.4). These assumptions control finite-sample behavior under shift but do not change the qualitative conclusion. The theory does not require the Bradley–Terry (Bradley & Terry, 1952) annotator model; it depends only on feature-difference statistics.

**On the controlled experimental design.** Our experiments are deliberately controlled: they separate causal and spurious attributes, vary spurious statistics across training and deployment, and keep the underlying preference structure fixed. This control is necessary to directly test the paper's central claims and is generally unavailable in real preference data, where causal and spurious factors are entangled. The method itself does not require explicit identification of spurious factors. The construction is used only to make the mechanism empirically testable. The spurious correlations we study are also reported in deployed alignment systems (Casper et al., 2023), so the isolated mechanisms remain practically relevant despite the synthetic benchmark. Broader evaluation on natural preference data remains an important next step.

One might expect the controlled construction to require idealized ties. However, the ties in our LLM and nonlinear experiments are already approximate: near-ties can preserve a weak causal margin while injecting larger variation along spurious directions, showing that exact ties are not essential for the mechanism. The local log-linear framework also extends to this regime by replacing ideal tie statistics with approximate ones. Appendix F.1 gives this characterization and a log-linear near-tie sweep.

**Tie construction and the role of $\alpha$.** A limitation of the method is that the theory does not prescribe how to construct ties. However, tie training is not restricted to hand-crafted ties. The mechanism suggests three practical regimes. (i) *Known spurious attribute:* when the spurious factor is identified, as in our LLM experiments, ties can be constructed by direct perturbation. (ii) *Candidate axes:* when the spurious direction is unknown but plausible candidates can be listed (e.g., verbosity, style, formatting), ties are constructed

by content-preserving perturbations across these axes. (iii) *Fully agnostic:* cross-prompt constructions form tie-like pairs that can perturb many spurious directions at once.

This data-level view connects to RRM (Liu et al., 2025), which addresses spurious correlations in reward learning through a causal framework and feature-agnostic augmentation. RRM targets reward-model training with soft labels for neutral pairs, whereas we use symmetric hard-labeled pairs in the unmodified DPO loss as a direct consequence of the mechanism we analyze. We view RRM's augmentation as introducing preference-neutral signal at the data level, complementary to ours.

The parameter $\alpha$ controls the strict–tie mixture: decreasing $\alpha$ increases tie mass, and Corollary 6.3 establishes monotone spurious shrinkage as tie mass grows. In practice, a moderate tie budget can induce spurious-direction regularization. We used an unoptimized $\alpha = 0.3$ in the LLM experiment, with a full $\alpha$-sweep reported in Appendix F.3. Tie effectiveness also depends on which pairs are added. Suppression scales with the alignment between ties and $\Sigma_{ss}$, so under a fixed budget, selecting informative pairs can yield greater suppression than uniformly increasing tie mass. Appendix F.1 further examines the $\alpha$-dependence of causal components. Two problems remain open: automatic tie construction and scalable allocation of a fixed tie budget toward high-impact directions.

## 9. Conclusion

This work analyzes spurious correlation learning in preference optimization and establishes three main results. First, under standard preference objectives, spurious correlations can be learned at the population level through mean spurious bias and causal–spurious correlation leakage (Theorem 4.1). Second, when spurious statistics shift between training and deployment, such reliance can induce deployment degradation that persists even with unlimited training data drawn from the same distribution (Theorem 5.3). Third, tie training, a simple data augmentation strategy using equal-utility preference pairs, can reduce spurious reliance by selectively regularizing spurious directions (Theorem 6.2).

Together, these results suggest that robustness to spurious correlations in preference optimization depends not only on model capacity or data scale, but also on the structure of supervision. Tie training provides one concrete instance of this approach within standard preference objectives. Our analysis focuses on a local regime and assumes access to informative ties, enabling tractable characterization; extending these results beyond this regime and developing scalable tie construction methods remain important directions for future work. Further discussion appears in Appendix H.

## Impact statement

This paper presents work whose goal is to advance the field of machine learning. There are many potential societal consequences of our work, none which we feel must be specifically highlighted here.

## Acknowledgments

We thank the anonymous reviewers for their helpful comments. We would also like to thank the Future Impact Group for their help in initiating this project. GL would like to thank the support of National Science Foundation (DMS-2533878, DMS-2053746, DMS-2134209, ECCS-2328241, CBET-2347401 and OAC-2311848), and U.S. Department of Energy (DOE) Office of Science Advanced Scientific Computing Research program DE-SC0023161, the SciDAC LEADS Institute, and DOE–Fusion Energy Science, under grant number: DE-SC0024583.

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

# A. Full Proof of Spurious Learning Mechanism (Theorem 4.1)

This section provides the full proof of Theorem 4.1 by progressively isolating the mechanisms that drive spurious learning in direct preference optimization for log-linear policies. We first restate the population objective and derive its exact gradient, then linearize the dynamics under the local regime assumption. We next decompose the resulting covariance structure into causal and spurious blocks and solve the corresponding equilibrium in closed form. Finally, we interpret the solution to separate and quantify the mean-bias and correlation-leakage contributions, completing the proof.

Our analysis relies on the local linearization assumption introduced in Section 3 (Assumption 3.2), which we restate below for completeness.

**Assumption A.1** (Local regime). We assume that the scaled DPO margin satisfies

$$|\beta\,\tilde{\theta}^\top \Delta\phi| \ll 1.$$

with high probability under the data distribution.

This regime arises in several settings. First, during early training the policy remains close to the reference policy. Second, the margin is small when competing responses receive similar scores under the current policy. Finally, the regime naturally arises in regions of near-indifference. In these settings, the DPO objective admits a first-order linear approximation, enabling analytical characterization of the training dynamics.

## A.1. Population Objective and Gradient

We study the population objective:

$$L(\theta) := \mathbb{E}_{(x,y_w,y_l)\sim\mathcal{D}}[\ell(\theta; x, y_w, y_l)] = -\mathbb{E}\left[\log\sigma\big(\beta\,\tilde{\theta}^\top\Delta\phi\big)\right],$$

where $\tilde{\theta} := \theta - \theta_{\text{ref}}$ and $\Delta\phi = \phi(x, y_w) - \phi(x, y_l)$ is the feature difference vector.

**Full DPO pairwise gradient.** It follows from the chain rule and $\frac{d}{dz}[-\log\sigma(z)] = \sigma(z) - 1$. In particular,

$$\frac{d}{dz}\ell(z) = \frac{d}{dz}\big(-\log\sigma(z)\big) = -\frac{1}{\sigma(z)}\cdot\sigma(z)\big(1-\sigma(z)\big) = -\big(1-\sigma(z)\big) = \sigma(z) - 1.$$

Defining the adaptive weight $w(\Delta_\theta) := 1 - \sigma(\beta\Delta_\theta) \in (0,1)$, where $\Delta_\theta := \tilde{\theta}^\top\Delta\phi$, and applying the chain rule with $z = \beta\tilde{\theta}^\top\Delta\phi$, the gradient is:

$$\nabla_\theta\ell(\theta; x, y_w, y_l) = (\sigma(\beta\Delta_\theta) - 1)\cdot\beta\Delta\phi = -\beta\cdot w(\Delta_\theta)\cdot\Delta\phi$$

**Linearized population gradient.** Taking the expectation of the per-example gradient,

$$\nabla_\theta L(\theta) = \mathbb{E}[\nabla_\theta\ell] = -\beta\,\mathbb{E}[w(\Delta_\theta)\Delta\phi].$$

We decompose the expectation of this product via the vector–scalar covariance identity,

$$\mathbb{E}[w(\Delta_\theta)\Delta\phi] = \mathbb{E}[w(\Delta_\theta)]\mathbb{E}[\Delta\phi] + \text{Cov}\big(\Delta\phi, w(\Delta_\theta)\big),$$

and linearize each term in turn (Steps 1–2) before recombining (Step 3).

Under the local regime (Assumption A.1), we derive the linearized population gradient next. To this end, let us first define the feature second moment matrix:

$$\Sigma := \mathbb{E}[\Delta\phi\Delta\phi^\top] \in \mathbb{R}^{d\times d}$$

This is related to the covariance matrix $C = \text{Cov}(\Delta\phi)$ by $\Sigma = C + \mu\mu^\top$ where $\mu = \mathbb{E}[\Delta\phi]$. We use second moments (rather than centered covariances) because they lead to simpler final formulas.

**Step 1: We linearize the expected weight:**

$$\mathbb{E}[w(\Delta_\theta)] \approx \mathbb{E}\left[\frac{1}{2} - \frac{\beta}{4}\tilde{\theta}^\top \Delta\phi\right] = \frac{1}{2} - \frac{\beta}{4}\tilde{\theta}^\top \mu$$

This approximation follows from the first-order Taylor expansion of the sigmoid around zero: $\sigma(z) \approx \frac{1}{2} + \frac{z}{4}$, which is valid under Assumption A.1.

**Step 2: We compute the covariance term as follows:**

$$\mathrm{Cov}(\Delta\phi, w(\Delta_\theta)) = \mathrm{Cov}\left(\Delta\phi, \frac{1}{2} - \frac{\beta}{4}\tilde{\theta}^\top \Delta\phi\right) = -\frac{\beta}{4}\mathrm{Cov}(\Delta\phi, \tilde{\theta}^\top \Delta\phi)$$

Using the identity for covariance between a vector and a linear form yields:

$$\mathrm{Cov}(\Delta\phi, \tilde{\theta}^\top \Delta\phi) = \mathbb{E}[\Delta\phi \cdot (\tilde{\theta}^\top \Delta\phi)] - \mathbb{E}[\Delta\phi]\mathbb{E}[\tilde{\theta}^\top \Delta\phi]$$
$$= \mathbb{E}[\Delta\phi\Delta\phi^\top]\tilde{\theta} - \mu(\mu^\top \tilde{\theta}) = \Sigma\tilde{\theta} - \mu\mu^\top \tilde{\theta}$$

Therefore:

$$\mathrm{Cov}(\Delta\phi, w(\Delta_\theta)) = -\frac{\beta}{4}(\Sigma\tilde{\theta} - \mu\mu^\top \tilde{\theta})$$

**Step 3: We combine the terms:**

$$\nabla_\theta L(\theta) = -\beta \cdot \mathbb{E}[w(\Delta_\theta)] \cdot \mu - \beta \cdot \mathrm{Cov}(\Delta\phi, w(\Delta_\theta))$$
$$= -\beta\left(\frac{1}{2} - \frac{\beta}{4}\tilde{\theta}^\top \mu\right)\mu - \beta\left(-\frac{\beta}{4}(\Sigma\tilde{\theta} - \mu\mu^\top \tilde{\theta})\right)$$
$$= -\frac{\beta}{2}\mu + \frac{\beta^2}{4}(\tilde{\theta}^\top \mu)\mu + \frac{\beta^2}{4}\Sigma\tilde{\theta} - \frac{\beta^2}{4}\mu\mu^\top \tilde{\theta}$$
$$= -\frac{\beta}{2}\mu + \frac{\beta^2}{4}(\mu^\top \tilde{\theta})\mu + \frac{\beta^2}{4}\Sigma\tilde{\theta} - \frac{\beta^2}{4}\mu(\mu^\top \tilde{\theta})$$

The terms $\frac{\beta^2}{4}(\mu^\top \tilde{\theta})\mu$ and $-\frac{\beta^2}{4}\mu(\mu^\top \tilde{\theta})$ cancel exactly (this cancellation is why using $\Sigma$ (rather than $C$) gives a clean final formula). Thus, the linearized population gradient takes the form

$$\nabla_\theta L(\theta) \approx -\frac{\beta}{2}\mu + \frac{\beta^2}{4}\Sigma\tilde{\theta}, \tag{10}$$

accurate up to third-order terms in the scaled margin.

## A.2. Solving the Equilibrium

We now derive the explicit equilibrium expressions for the causal and spurious components of the linearized population optimum stated in Section 4.

**Assumption A.2** (Spurious-Causal Feature Decomposition). We decompose the feature map into causal and spurious components, i.e.,

$$\phi(x, y) = (\phi_c(x, y), \phi_s(x, y)),$$

where $\phi_c(x, y) \in \mathbb{R}^{d_c}$ contains *causal features* and $\phi_s(x, y) \in \mathbb{R}^{d_s}$ contains *spurious features*. This induces the same decomposition for parameters and pairwise differences: $\tilde{\theta} = (\tilde{\theta}_c, \tilde{\theta}_s)$, $\Delta\phi = (\Delta\phi_c, \Delta\phi_s)$.

**Setup.** Using the linearized population gradient (10), the equilibrium condition $\nabla_\theta L(\theta^\star) = \mathbf{0}$ yields

$$\Sigma\tilde{\theta}^\star = \frac{2}{\beta}\mu,$$

where $\tilde{\theta}^\star = \theta^\star - \theta_{\text{ref}}$. Writing $\tilde{\theta}^\star = [\tilde{\theta}_c^{\star\top}, \tilde{\theta}_s^{\star\top}]^\top$, $\mu = [\mu_c^\top, \mu_s^\top]^\top$, and

$$\Sigma = \begin{bmatrix} \Sigma_{cc} & \Sigma_{cs} \\ \Sigma_{sc} & \Sigma_{ss} \end{bmatrix},$$

the equilibrium equations decompose as

$$\Sigma_{cc}\tilde{\theta}_c^\star + \Sigma_{cs}\tilde{\theta}_s^\star = \frac{2}{\beta}\mu_c, \tag{11}$$

$$\Sigma_{sc}\tilde{\theta}_c^\star + \Sigma_{ss}\tilde{\theta}_s^\star = \frac{2}{\beta}\mu_s. \tag{12}$$

**Solving for the spurious component.** Assume $\Sigma_{ss}$ is invertible. From (12),

$$\tilde{\theta}_s^\star = \Sigma_{ss}^{-1}\left(\frac{2}{\beta}\mu_s - \Sigma_{sc}\tilde{\theta}_c^\star\right).$$

Substituting into (11) gives

$$\left(\Sigma_{cc} - \Sigma_{cs}\Sigma_{ss}^{-1}\Sigma_{sc}\right)\tilde{\theta}_c^\star = \frac{2}{\beta}\left(\mu_c - \Sigma_{cs}\Sigma_{ss}^{-1}\mu_s\right).$$

**Schur complement.** Define the Schur complement:

$$S_c := \Sigma_{cc} - \Sigma_{cs}\Sigma_{ss}^{-1}\Sigma_{sc}.$$

Assuming $S_c$ is invertible, the causal component satisfies

$$\tilde{\theta}_c^\star = \frac{2}{\beta}S_c^{-1}\left(\mu_c - \Sigma_{cs}\Sigma_{ss}^{-1}\mu_s\right).$$

**Explicit spurious equilibrium.** Substituting the expression for $\tilde{\theta}_c^\star$ back yields

$$\begin{aligned}
\tilde{\theta}_s^\star &= \Sigma_{ss}^{-1}\left(\frac{2}{\beta}\mu_s - \Sigma_{sc}\tilde{\theta}_c^\star\right) \\
&= \frac{2}{\beta}\Sigma_{ss}^{-1}\left[\mu_s - \Sigma_{sc}S_c^{-1}\mu_c + \Sigma_{sc}S_c^{-1}\Sigma_{cs}\Sigma_{ss}^{-1}\mu_s\right] \\
&= \frac{2}{\beta}\Sigma_{ss}^{-1}\left[\left(I + \Sigma_{sc}S_c^{-1}\Sigma_{cs}\Sigma_{ss}^{-1}\right)\mu_s - \Sigma_{sc}S_c^{-1}\mu_c\right].
\end{aligned}$$

This completes the derivation of the explicit spurious equilibrium reported in Section 4.

## A.3. Interpretation of the Equilibrium Solution

We now interpret the equilibrium solution derived above and state our main result characterizing spurious learning at the population level. The following theorem formalizes when and why the population optimizer assigns nonzero weight to spurious features under the linearized dynamics.

**Theorem A.3** (Spurious learning from mean bias and correlation leakage). *Under the linearized population dynamics (Equation (10)), assume $\Sigma = \mathbb{E}[\Delta\phi\Delta\phi^\top] \succ 0$ (equivalently, $\Sigma_{ss} \succ 0$ and $S_c := \Sigma_{cc} - \Sigma_{cs}\Sigma_{ss}^{-1}\Sigma_{sc} \succ 0$). If either $\mu_s \neq 0$ (mean spurious bias) or $\Sigma_{sc} \neq 0$ (causal–spurious leakage), then generically[1] the population optimum satisfies $\tilde{\theta}_s^\star \neq 0$.*

*Proof.* The linearized population gradient takes the form (see Appendix A.1 for details)

$$g(\tilde{\theta}) = -\frac{\beta}{2}\mu + \frac{\beta^2}{4}\Sigma\tilde{\theta},$$

---

[1] Outside a Lebesgue measure-zero set of problem instances.

so the equilibrium satisfies $g(\tilde{\theta}^\star) = 0$, i.e.

$$\Sigma\tilde{\theta}^\star = \frac{2}{\beta}\mu.$$

Using the spurious-causal feature decomposition (Assumption A.2), we can write the equilibrium condition in block form as

$$\Sigma_{cc}\tilde{\theta}_c^\star + \Sigma_{cs}\tilde{\theta}_s^\star = \frac{2}{\beta}\mu_c, \tag{13}$$

$$\Sigma_{sc}\tilde{\theta}_c^\star + \Sigma_{ss}\tilde{\theta}_s^\star = \frac{2}{\beta}\mu_s. \tag{14}$$

Assume $\Sigma_{ss} \succ 0$ and $S_c \succ 0$ so the solution is unique. Solving by Schur complement (see Appendix A.2 for details) yields,

$$\tilde{\theta}_c^\star = \frac{2}{\beta}S_c^{-1}\big(\mu_c - \Sigma_{cs}\Sigma_{ss}^{-1}\mu_s\big),$$

Substituting into (14) yields

$$\tilde{\theta}_s^\star = \frac{2}{\beta}\Sigma_{ss}^{-1}\Big[\mu_s - \Sigma_{sc}S_c^{-1}\mu_c + \Sigma_{sc}S_c^{-1}\Sigma_{cs}\Sigma_{ss}^{-1}\mu_s\Big]$$

$$= \underbrace{\frac{2}{\beta}\Sigma_{ss}^{-1}\Big(I + \Sigma_{sc}S_c^{-1}\Sigma_{cs}\Sigma_{ss}^{-1}\Big)\mu_s}_{\text{(i) mean-bias term}} + \underbrace{\Big(-\frac{2}{\beta}\Sigma_{ss}^{-1}\Sigma_{sc}S_c^{-1}\mu_c\Big)}_{\text{(ii) correlation-leakage term}}. \tag{15}$$

This proves the claimed decomposition into a component driven by $\mu_s$ and a component driven by $\Sigma_{sc}$. It remains to show: if either $\mu_s \neq 0$ or $\Sigma_{sc} \neq 0$, then generically $\tilde{\theta}_s^\star \neq 0$.

**Case 1:** $\mu_s \neq 0$. The mean-bias term in (15) equals $A\mu_s$ with $A := \frac{2}{\beta}\Sigma_{ss}^{-1}\big(I + \Sigma_{sc}S_c^{-1}\Sigma_{cs}\Sigma_{ss}^{-1}\big)$. The matrix $A$ is invertible: indeed, $I + \Sigma_{sc}S_c^{-1}\Sigma_{cs}\Sigma_{ss}^{-1}$ is similar to $I + \Sigma_{ss}^{-1/2}\Sigma_{sc}S_c^{-1}\Sigma_{cs}\Sigma_{ss}^{-1/2}$, whose second term is positive semidefinite. Hence $A\mu_s \neq 0$ when $\mu_s \neq 0$. Since $\tilde{\theta}_s^\star$ also contains the leakage term, $\tilde{\theta}_s^\star = 0$ can occur only through exact cancellation, $A\mu_s = \frac{2}{\beta}\Sigma_{ss}^{-1}\Sigma_{sc}S_c^{-1}\mu_c$. This equality defines a proper affine constraint on the problem instance. Thus, generically $\tilde{\theta}_s^\star \neq 0$.

**Case 2:** $\mu_s = 0$ **but** $\Sigma_{sc} \neq 0$. Then $\tilde{\theta}_c^\star = \frac{2}{\beta}S_c^{-1}\mu_c$. Substituting into (14) gives

$$\tilde{\theta}_s^\star = -\Sigma_{ss}^{-1}\Sigma_{sc}\tilde{\theta}_c^\star = -\frac{2}{\beta}\Sigma_{ss}^{-1}\Sigma_{sc}S_c^{-1}\mu_c.$$

If $\Sigma_{sc} \neq 0$, then for generic $\mu_c$ this expression is nonzero (failing only when $\mu_c$ lies in the nullspace of $\Sigma_{sc}S_c^{-1}$, again a measure-zero condition). Thus, generically $\tilde{\theta}_s^\star \neq 0$.

Combining the two cases establishes the theorem. $\qquad\square$

# B. Deployment Error and Margin-Based Analysis

This appendix analyzes how spurious correlations learned during training induce vulnerabilities at deployment. Let $P$ denote the training preference distribution over triples $(x, y_w, y_l)$, and let $Q$ denote the deployment preference distribution over the same space. We write

$$\theta_{\text{train}} \in \arg \max_{\theta \in \Theta_B} \tilde{J}^{(P)}(\theta), \qquad \tilde{\theta}_{\text{train}} := \theta_{\text{train}} - \theta_{\text{ref}},$$

for the population optimizer of the log-linear DPO surrogate under $P$, i.e., $\tilde{J}^{(P)}(\theta) := \mathbb{E}_P[\log \sigma(\beta \, \tilde{\theta}^\top \Delta \phi)]$. The learned parameters are fixed after training; deployment changes the distribution from $P$ to $Q$, not the parameters.

Our analysis operates directly on the log-linear DPO surrogate and the induced margins. In the local regime, the surrogate admits a first-order expansion in terms of the expected preference margin, which provides a transparent characterization of deployment degradation under distribution shift.

# C. Assumptions for Deployment Analysis

We start by providing detailed discussion of the assumptions used in the deployment suboptimality analysis.

**Assumption C.1** (Bounded feature differences). We assume that feature differences are uniformly bounded:

$$\|\Delta \phi(x, y_w, y_l)\|_2 \leq 1 \quad \text{for all preference pairs } (x, y_w, y_l).$$

This is a normalization convention rather than a substantive restriction. Specifically, if the feature map $\phi(x, y)$ satisfies

$$R := \sup_{x, y_w, y_l} \|\phi(x, y_w) - \phi(x, y_l)\|_2 < \infty,$$

we rescale $\tilde{\phi} = \phi / R$ and reparametrize $\tilde{\theta} = R\theta$, preserving all margins: $\tilde{\theta}^\top \tilde{\phi} = \theta^\top \phi$. In practice, the assumption holds whenever features are normalized (e.g., embeddings projected onto the unit sphere), bounded by construction (e.g., indicator features, clipped activations), or supported on a compact set.

**Assumption C.2** (Bounded parameters). The deviation from the reference parameters is bounded:

$$\|\tilde{\theta}\|_2 \leq B \quad \text{for all } \theta \in \Theta_B, \qquad \tilde{\theta} := \theta - \theta_{\text{ref}}.$$

This assumption restricts optimization to a compact parameter set and ensures that preference score differences $\beta \, \tilde{\theta}^\top \Delta \phi$ remain bounded. The bound $B$ may be enforced explicitly (e.g., via constrained optimization) or implicitly (e.g., via early stopping).

**Joint implication.** Under Assumptions C.1 and C.2, preference score differences satisfy

$$|\beta \, \tilde{\theta}^\top \Delta \phi| \leq \beta \, \|\tilde{\theta}\|_2 \, \|\Delta \phi\|_2 \leq \beta B.$$

Thus the local regime (Assumption C.3) holds uniformly over $\Theta_B$ whenever $\beta B \leq \epsilon$.

**Assumption C.3** (Local regime). Preference score differences remain small:

$$|\beta \, \tilde{\theta}^\top \Delta \phi| \leq \epsilon \ll 1 \quad \text{with high probability under } P \text{ and } Q.$$

**Role in the analysis.** The local regime assumption is used in two places.

1. It justifies the first-order expansion of the surrogate in terms of the margin (Proposition C.5).

2. It ensures that the pairwise Fisher curvature remains close to a constant multiple of the unweighted second moment, enabling the geometry transfer (Assumption C.4).

The local regime is also naturally satisfied for tie and near-tie pairs, since these pairs have small margins by construction.

**Quantitative scale.** The linearization $\sigma(z) \approx \frac{1}{2} + \frac{z}{4}$ has absolute error below $0.02$ for $|z| \leq 1$ and below $0.12$ for $|z| \leq 2$. The approximation is accurate when $|\beta\, \tilde{\theta}^\top \Delta\phi| \lesssim 1$.

**Beyond the local regime.** When $\beta B \gg 1$, the weight function $w(\Delta_\theta) = \sigma(z)(1 - \sigma(z))$ varies strongly with the margin and saturates toward zero for large $|z|$. In that case, second-order expansions around $\theta_{\mathrm{ref}}$ no longer provide uniform control over the objective, and first-order margin approximations remain qualitatively informative but the quantitative bounds derived below require modification.

**Deployment curvature.** Let $\|v\|_A^2 := v^\top A v$ for any positive semi-definite matrix $A$. Define the (pairwise) Fisher information under deployment $Q$, evaluated at $\theta_{\mathrm{train}}$:

$$\bar{\Sigma}^{(Q)} := \mathbb{E}_{(x,y_w,y_l) \sim Q}\left[\beta^2\, w(\Delta_\theta)\, \Delta\phi\Delta\phi^\top\right], \qquad w(\Delta_\theta) := \sigma(z)(1 - \sigma(z)), \;\; z := \beta\, \tilde{\theta}_{\mathrm{train}}^\top \Delta\phi.$$

Define analogously $\bar{\Sigma}^{(P)}$ by replacing $Q$ with $P$. Let $H_P := \bar{\Sigma}^{(P)} + \lambda I$ for regularization parameter $\lambda > 0$.

Informally, in the local regime, $w(z) = \frac{1}{4} + O(z^2)$, so

$$\bar{\Sigma}^{(Q)} = \frac{\beta^2}{4}\,\Sigma^{(Q)} + O\!\left(\beta^2\, \mathbb{E}_Q[z^2\, \Delta\phi\, \Delta\phi^\top]\right), \qquad \Sigma^{(Q)} := \mathbb{E}_Q[\Delta\phi\, \Delta\phi^\top].$$

Thus, locally, the deployment Fisher is controlled by the same second-moment structure that appears in the population gradient analysis (Appendix A.1).

**Assumption C.4** (Geometry transfer). There exists $\kappa_\Pi \geq 1$ such that for all $v \in \mathbb{R}^d$,

$$\|v\|_{\frac{\beta^2}{4}\Sigma^{(Q)}}^2 \leq \kappa_\Pi \|v\|_{H_P}^2.$$

This assumption controls how much the local curvature of the loss can change between training and deployment. Without it, the estimation error measured under $P$ would not transfer to $Q$: the Hessian under $Q$ could be arbitrarily larger, making the quadratic bound on the estimation term vacuous.

**Sufficient conditions for geometry transfer.** Assumption C.4 holds with an explicit constant $\kappa_\Pi$ under any of the following conditions.

**Condition 1 (Direct curvature domination).** If there exists $c_\Pi \geq 1$ such that

$$\tfrac{\beta^2}{4}\Sigma^{(Q)} \preceq c_\Pi\, H_P,$$

then Assumption C.4 holds with $\kappa_\Pi \leq c_\Pi$.

**Condition 2 (Bounded density ratio).** If $Q \ll P$ and $\frac{dQ}{dP} \leq \rho_{\max}$ almost surely, then by change of measure:

$$\bar{\Sigma}^{(Q)} = \mathbb{E}_P\left[\frac{dQ}{dP}\, \beta^2\, w(z)\, \Delta\phi\, \Delta\phi^\top\right] \preceq \rho_{\max}\, \bar{\Sigma}^{(P)} \preceq \rho_{\max}\, H_P,$$

which bounds the worst-case reweighting of preference pairs under $Q$. To obtain Assumption C.4 in the form above, one applies the same argument to the raw second moment, $\frac{\beta^2}{4}\Sigma^{(Q)} \preceq \rho_{\max}\frac{\beta^2}{4}\Sigma^{(P)}$, together with a curvature lower bound $\frac{\beta^2}{4}\Sigma^{(P)} \preceq c'\, H_P$ as supplied by Condition 3. The resulting constant is $\kappa_\Pi \leq c'\rho_{\max}$.

**Condition 3 (Covariance transfer under local regime).** Assume the local regime holds uniformly, so that $|\beta\, \tilde{\theta}_{\mathrm{train}}^\top \Delta\phi| \leq \beta B$ and therefore $\beta^2\gamma_{\beta,B}\, \Delta\phi\Delta\phi^\top \preceq \beta^2 w(\Delta_\theta)\, \Delta\phi\Delta\phi^\top \preceq \frac{\beta^2}{4}\, \Delta\phi\Delta\phi^\top$, where $\gamma_{\beta,B} := \sigma(\beta B)(1 - \sigma(\beta B))$. If the feature covariances satisfy

$$\Sigma^{(Q)} := \mathbb{E}_Q[\Delta\phi\Delta\phi^\top] \preceq c\left(\Sigma^{(P)} + \lambda I\right)$$

for some $c \geq 1$, then

$$\frac{\beta^2}{4}\,\Sigma^{(Q)} \preceq \frac{c}{4}\,\beta^2\!\left(\Sigma^{(P)} + \lambda I\right).$$

Moreover, since $w(z) \geq \gamma_{\beta,B}$ uniformly on $\Theta_B$,

$$\bar{\Sigma}^{(P)} \succeq \beta^2 \gamma_{\beta,B} \Sigma^{(P)}.$$

If additionally $\beta^2 \gamma_{\beta,B} \leq 1$ (which holds in particular whenever $\beta \leq 1$, since then $\beta^2 \gamma_{\beta,B} \leq 1/4$), then

$$H_P = \bar{\Sigma}^{(P)} + \lambda I \succeq \beta^2 \gamma_{\beta,B} \Sigma^{(P)} + \lambda I \succeq \gamma_{\beta,B} \beta^2 (\Sigma^{(P)} + \lambda I),$$

where the last step uses $\beta^2 \gamma_{\beta,B} \leq 1$. Combining,

$$\frac{\beta^2}{4} \Sigma^{(Q)} \preceq \frac{c}{4\gamma_{\beta,B}} H_P,$$

so, provided $\beta^2 \gamma_{\beta,B} \leq 1$, Assumption C.4 holds with $\kappa_\Pi \leq c/(4\gamma_{\beta,B})$.

## C.1. Deployment Proxy: Expected Margins

We now characterize deployment degradation through the expected preference margin.

**Margin definition.** The expected margin under distribution $D$ measures the average score gap between preferred and dispreferred responses: $m_D(\theta) := \mathbb{E}_{(x,y_w,y_l)\sim D}[\beta \tilde{\theta}^\top \Delta\phi]$. The margin decomposes into causal and spurious components:

$$m_D(\theta) = \underbrace{\beta \tilde{\theta}_c^\top \mu_c^{(D)}}_{=:m_D^{\mathrm{causal}}(\theta)} + \underbrace{\beta \tilde{\theta}_s^\top \mu_s^{(D)}}_{=:m_D^{\mathrm{spurious}}(\theta)}. \tag{16}$$

where $\mu_c^{(D)} := \mathbb{E}_D[\Delta\phi_c]$ and $\mu_s^{(D)} := \mathbb{E}_D[\Delta\phi_s]$.

Under the training distribution $P$, the spurious term may contribute positively to the learned classifier. Under deployment $Q$, this term may shrink, flip sign, or become pure noise, depending on the shift scenario (Section E).

**Margin approximation of objective differences.** We now restate Proposition 5.1 and provide the margin approximation proof.

**Proposition C.5** (First-order margin approximation). *Under the local regime (Assumption C.3), the pairwise surrogate objective $\tilde{J}^{(D)}(\theta) := \mathbb{E}_D[\log \sigma(\beta\tilde{\theta}^\top \Delta\phi)]$ satisfies*

$$\tilde{J}^{(Q)}(\theta) - \tilde{J}^{(P)}(\theta) = \frac{1}{2}\big(m_Q(\theta) - m_P(\theta)\big) + O(\beta^2 \|\tilde{\theta}\|^2).$$

*Proof.* Let $z := \beta \tilde{\theta}^\top \Delta\phi$. Under Assumption C.3, we have $|z| \leq \epsilon \ll 1$ with high probability. Since $\log \sigma$ is $C^2$, a Taylor expansion at $0$ gives

$$\log \sigma(z) = \log \sigma(0) + (\log \sigma)'(0)\, z + O(z^2) = \log \frac{1}{2} + \frac{1}{2} z + O(z^2),$$

because $(\log \sigma)'(z) = 1 - \sigma(z)$ and hence $(\log \sigma)'(0) = 1 - \sigma(0) = 1/2$.

Taking expectations under $D$ yields

$$\tilde{J}^{(D)}(\theta) = \mathbb{E}_D[\log \sigma(z)] = \log \frac{1}{2} + \frac{1}{2}\mathbb{E}_D[z] + O(\mathbb{E}_D[z^2])$$

$$= \log \frac{1}{2} + \frac{1}{2} m_D(\theta) + O(\mathbb{E}_D[(\beta \tilde{\theta}^\top \Delta\phi)^2]).$$

Under Assumption C.1, $\|\Delta\phi\|_2 \leq 1$, so

$$\mathbb{E}_D[(\beta \tilde{\theta}^\top \Delta\phi)^2] \leq \beta^2 \|\tilde{\theta}\|_2^2 \, \mathbb{E}_D[\|\Delta\phi\|_2^2] \leq \beta^2 \|\tilde{\theta}\|_2^2,$$

and therefore

$$\tilde{J}^{(D)}(\theta) = \log \frac{1}{2} + \frac{1}{2}m_D(\theta) + O(\beta^2 \|\tilde{\theta}\|_2^2).$$

Applying this with $D = P$ and $D = Q$ and subtracting cancels the constant $\log(1/2)$, giving

$$\tilde{J}^{(Q)}(\theta) - \tilde{J}^{(P)}(\theta) = \frac{1}{2}\big(m_Q(\theta) - m_P(\theta)\big) + O(\beta^2 \|\tilde{\theta}\|_2^2),$$

as claimed. $\qquad\square$

We now show that spurious margin drives shift under stable causal statistics.

**Proposition C.6** (Spurious Margin Drives Shift). *When causal statistics are stable ($\mu_c^{(Q)} = \mu_c^{(P)}$), the objective difference at $\theta_{\text{train}}$ is determined by the spurious margin:*

$$\tilde{J}^{(Q)}(\theta_{\text{train}}) - \tilde{J}^{(P)}(\theta_{\text{train}}) = \frac{\beta}{2}\tilde{\theta}_{s,\text{train}}^\top(\mu_s^{(Q)} - \mu_s^{(P)}) + O(\beta^2 \|\tilde{\theta}_{\text{train}}\|^2).$$

*Proof.* Apply Proposition C.5 with $\theta = \theta_{\text{train}}$. By (16):

$$m_Q(\theta_{\text{train}}) - m_P(\theta_{\text{train}}) = \beta\,\tilde{\theta}_{c,\text{train}}^\top(\mu_c^{(Q)} - \mu_c^{(P)}) + \beta\,\tilde{\theta}_{s,\text{train}}^\top(\mu_s^{(Q)} - \mu_s^{(P)}).$$

When $\mu_c^{(Q)} = \mu_c^{(P)}$, the causal term vanishes:

$$m_Q(\theta_{\text{train}}) - m_P(\theta_{\text{train}}) = \beta\,\tilde{\theta}_{s,\text{train}}^\top(\mu_s^{(Q)} - \mu_s^{(P)}).$$

Substituting into Proposition C.5 yields the result. $\qquad\square$

**Interpretation.** Deployment degradation is driven by the spurious margin because the training optimizer $\tilde{\theta}_{s,\text{train}}$ was optimized to exploit the spurious correlation under $P$. When the deployment distribution $Q$ breaks this correlation, the same parameter becomes a source of vulnerability. The magnitude of degradation is controlled by $\|\tilde{\theta}_{s,\text{train}}\|$, the same quantity that tie training reduces (see Appendix D).

## C.2. Deployment Suboptimality Definition

Let

$$\theta_Q^\star := \arg\max_{\theta \in \Theta_B} \tilde{J}^{(Q)}(\theta)$$

denote the deployment-optimal parameters, where $\Theta_B := \{\theta \in \mathbb{R}^d : \|\theta - \theta_{\text{ref}}\|_2 \leq B\}$. Let $\hat{\theta}$ denote the empirical estimator trained on $n$ samples from $P$. Define the deployment suboptimality:

$$\text{SubOpt}_Q(\hat{\theta}) := \tilde{J}^{(Q)}(\theta_Q^\star) - \tilde{J}^{(Q)}(\hat{\theta}).$$

**Decomposition.** Inserting the population training optimizer $\theta_{\text{train}}$:

$$\text{SubOpt}_Q(\hat{\theta}) = \underbrace{\tilde{J}^{(Q)}(\theta_Q^\star) - \tilde{J}^{(Q)}(\theta_{\text{train}})}_{\text{shift term (distributional, irreducible)}} + \underbrace{\tilde{J}^{(Q)}(\theta_{\text{train}}) - \tilde{J}^{(Q)}(\hat{\theta})}_{\text{estimation term (statistical, reducible)}}. \qquad (17)$$

**Irreducible and reducible components.** The shift term depends on $\theta_{\text{train}}$ and does not depend on sample size. As $n \to \infty$, $\hat{\theta} \to \theta_{\text{train}}$ and the estimation term vanishes (as demonstrated in Appendix C.3):

$$\lim_{n \to \infty} \text{SubOpt}_Q(\hat{\theta}) = \tilde{J}^{(Q)}(\theta_Q^\star) - \tilde{J}^{(Q)}(\theta_{\text{train}}).$$

Proposition C.6 shows that, when causal statistics are stable, the deployment objective evaluated at $\theta_{\text{train}}$ is first-order sensitive to $\tilde{\theta}_{s,\text{train}}^\top(\mu_s^{(Q)} - \mu_s^{(P)})$. Thus, when the $Q$-optimal parameter removes or reverses this spurious reliance, the persistent shift term is driven by the same learned spurious component. The estimation term vanishes as $\hat{\theta} \to \theta_{\text{train}}$, so additional samples from $P$ do not remove this structural bottleneck. The fundamental bottleneck is spurious component $\tilde{\theta}_{s,\text{train}}$, which arises structurally from the training distribution (Section 4).

## C.3. Proof Deployment Suboptimality Bound

This appendix provides the proof of the deployment suboptimality bound (Theorem 5.3). For simplicity, we state the following high-probability concentration bound as an assumption. It follows from the self-normalized MLE analysis of (Zhu et al., 2023), specialized to our regularized estimator and bounded-feature setting.

**Assumption C.7** (Training-estimator concentration (Zhu et al., 2023)). With probability at least $1 - \delta$,

$$\|\hat{\theta} - \theta_{\text{train}}\|_{H_P} \leq \Gamma_n,$$

where

$$\Gamma_n = 2\beta\sqrt{\frac{2(d + \log(1/\delta))}{n}} + B\sqrt{\lambda}.$$

This is a standard self-normalized concentration bound for regularized generalized linear estimators (Zhu et al., 2023); in our setting, $H_P = \bar{\Sigma}^{(P)} + \lambda I$ plays the role of the regularized Fisher matrix. The first term captures statistical error and vanishes as $n \to \infty$; the second reflects regularization bias and vanishes as $\lambda \to 0$.

**Theorem C.8** (Deployment sub-optimality bound). *Under Assumptions C.1, C.2, C.3, C.4, and C.7, with probability at least $1 - \delta$:*

$$\text{SubOpt}_Q(\hat{\theta}) \leq \underbrace{\tilde{J}^{(Q)}(\theta_Q^\star) - \tilde{J}^{(Q)}(\theta_{\text{train}})}_{\text{shift term}} + \underbrace{G_Q\Gamma_n + \frac{\kappa_\Pi}{2}\Gamma_n^2}_{\text{estimation term}}, \tag{18}$$

*where $G_Q := \|\nabla\tilde{J}^{(Q)}(\theta_{\text{train}})\|_{H_P^{-1}}$.*

The estimation term decomposes into a first-order contribution controlled by the deployment gradient norm $G_Q$ and a second-order contribution controlled by the geometry transfer constant $\kappa_\Pi$. The gradient norm measures how far $\hat{\theta}$ is from being optimal under $Q$: when $P = Q$, this quantity is small near $\theta_{\text{train}}$; under distribution shift, it captures the first-order cost of deploying a $P$-optimal estimator under $Q$.

*Proof.* Let $\theta_{\text{train}}$ denote the (population) optimizer of the training surrogate under $P$ over $\Theta_B$, and let $\theta_Q^\star$ denote the (population) optimizer under $Q$. We organize the proof into six steps.

**Step 1 (Decomposition).** From (17):

$$\text{SubOpt}_Q(\hat{\theta}) = \underbrace{\tilde{J}^{(Q)}(\theta_Q^\star) - \tilde{J}^{(Q)}(\theta_{\text{train}})}_{\text{shift term}} + \underbrace{\tilde{J}^{(Q)}(\theta_{\text{train}}) - \tilde{J}^{(Q)}(\hat{\theta})}_{\text{estimation term}}. \tag{19}$$

It remains to bound the estimation term $\tilde{J}^{(Q)}(\theta_{\text{train}}) - \tilde{J}^{(Q)}(\hat{\theta})$.

**Step 2 (Smoothness upper bound).** For each $\theta \in \Theta_B$, the negative Hessian of $\tilde{J}^{(Q)}$ is

$$-\nabla^2\tilde{J}^{(Q)}(\theta) = \mathbb{E}_Q\left[\beta^2\,\sigma(z_\theta)(1 - \sigma(z_\theta))\,\Delta\phi\Delta\phi^\top\right], \qquad z_\theta := \beta(\theta - \theta_{\text{ref}})^\top\Delta\phi.$$

Since $\sigma(z)(1 - \sigma(z)) \leq 1/4$ for all $z$,

$$-\nabla^2\tilde{J}^{(Q)}(\theta) \preceq \frac{\beta^2}{4}\Sigma^{(Q)} \quad \text{for all } \theta \in \Theta_B.$$

Hence, by the standard smoothness inequality for concave functions, applied around $\theta_{\text{train}}$,

$$\tilde{J}^{(Q)}(\hat{\theta}) \geq \tilde{J}^{(Q)}(\theta_{\text{train}}) + \nabla\tilde{J}^{(Q)}(\theta_{\text{train}})^\top(\hat{\theta} - \theta_{\text{train}}) - \frac{1}{2}\|\hat{\theta} - \theta_{\text{train}}\|_{\frac{\beta^2}{4}\Sigma^{(Q)}}^2.$$

Rearranging gives

$$\tilde{J}^{(Q)}(\theta_{\text{train}}) - \tilde{J}^{(Q)}(\hat{\theta}) \leq \nabla\tilde{J}^{(Q)}(\theta_{\text{train}})^\top(\theta_{\text{train}} - \hat{\theta}) + \frac{1}{2}\|\hat{\theta} - \theta_{\text{train}}\|_{\frac{\beta^2}{4}\Sigma^{(Q)}}^2. \tag{20}$$

**Step 3 (Bound the linear term).**    By Cauchy–Schwarz in the Euclidean inner product:

$$\nabla \tilde{J}^{(Q)}(\theta_{\text{train}})^{\top}(\theta_{\text{train}} - \hat{\theta}) \leq \underbrace{\|\nabla \tilde{J}^{(Q)}(\theta_{\text{train}})\|_{H_P^{-1}}}_{=:G_Q} \cdot \|\hat{\theta} - \theta_{\text{train}}\|_{H_P}.$$

This uses the identity $g^{\top} v = g^{\top} H_P^{-1/2} H_P^{1/2} v \leq \|H_P^{-1/2} g\| \|H_P^{1/2} v\| = \|g\|_{H_P^{-1}} \|v\|_{H_P}$, which holds for any positive definite $H_P$ without requiring geometry transfer.

**Step 4 (Geometry transfer for the quadratic term).**    By Assumption C.4,

$$\|\hat{\theta} - \theta_{\text{train}}\|^2_{\frac{\beta^2}{4}\Sigma(Q)} \leq \kappa_\Pi \|\hat{\theta} - \theta_{\text{train}}\|^2_{H_P}.$$

**Step 5 (Concentration).**    By Assumption C.7, with probability at least $1 - \delta$:

$$\|\hat{\theta} - \theta_{\text{train}}\|_{H_P} \leq \Gamma_n.$$

**Step 6 (Combine).**    Substituting Steps 3–5 into (20):

$$\tilde{J}^{(Q)}(\theta_{\text{train}}) - \tilde{J}^{(Q)}(\hat{\theta}) \leq G_Q \, \Gamma_n + \frac{\kappa_\Pi}{2} \Gamma_n^2.$$

Inserting into the decomposition from Step 1 yields the stated bound.    $\square$

**Interpretation.**    The bound reveals three distinct sources of deployment suboptimality:

1. **Shift term:** The irreducible gap between the $P$-optimal and $Q$-optimal parameters, driven by $\tilde{\theta}_{s,\text{train}}^{\top}(\mu_s^{(Q)} - \mu_s^{(P)})$ when causal statistics are stable (Proposition C.6).

2. **Gradient term:** The first-order cost of deploying a $P$-trained estimator under $Q$. This quantity $G_Q$ measures how far the population training optimizer $\theta_{\text{train}}$ is from stationarity under $Q$. It is small when $P \approx Q$ near $\theta_{\text{train}}$ and grows with distribution shift.

3. **Quadratic term:** The second-order estimation penalty, controlled by the geometry transfer constant $\kappa_\Pi$ and the statistical rate $\Gamma_n^2 = O(d/n + B^2\lambda)$.

Tie training (Appendix D) reduces the shift term by suppressing $\|\tilde{\theta}_{s,\text{train}}^{\star}\|$ and simultaneously reduces the gradient term by making the estimator less sensitive to spurious distributional changes.

# D. Tie Training Theory

Theorem A.3 identifies two drivers of spurious learning: spurious mean bias $\mu_s = \mathbb{E}[\Delta\phi_s]$ and causal–spurious leakage $\Sigma_{sc} = \mathbb{E}[\Delta\phi_c\Delta\phi_s^\top]$. Both are properties of the training distribution, so the natural remedy is a data-level intervention. Pairs with matched causal features ($\Delta\phi_c = 0$), varying spurious features ($\Delta\phi_s \neq 0$), and zero spurious mean bias ($\mathbb{E}_{\text{tie}}[\Delta\phi_s] = 0$) dilute the effective contribution of both drivers under mixture training. This section formalizes this construction as *tie training*, derives the resulting mixed-training equilibrium, proves that tie training reduces the spurious parameter magnitude (Theorem D.3), with the reduction being unconditional in the scalar spurious case ($d_s = 1$), and derives a quantitative reduction bound (Corollary D.4) that sharpens the main theorem.

## D.1. Tie Construction Details

We formalize the tie data-generating process and labeling scheme. We start by defining a tie pair:

**Definition D.1** (Tie pair). A tie pair is a tuple $(x, y_A, y_B)$ with equal utility, $u^\dagger(x, y_A) = u^\dagger(x, y_B)$.

**Feature structure.** We construct tie pairs so that causal features match while spurious features differ: $\Delta\phi_c := \phi_c(x, y_A) - \phi_c(x, y_B) = 0$ and $\Delta\phi_s := \phi_s(x, y_A) - \phi_s(x, y_B) = \delta_s \neq 0$. Thus, ties vary only in spurious coordinates.

**Random labeling.** For each tie pair, we assign the winner-loser label uniformly at random. With probability $1/2$, we label $y_A \succ y_B$ and set $\Delta\phi = [0; \delta_s]$; with probability $1/2$, we label $y_B \succ y_A$ and set $\Delta\phi = [0; -\delta_s]$. The zero-mean property $\mathbb{E}_{\text{tie}}[\Delta\phi] = 0$ follows from the symmetric random labeling, not from any assumption on the distribution of $\delta_s$. Then

$$\Sigma^{\text{tie}} := \mathbb{E}_{\text{tie}}[\Delta\phi\Delta\phi^\top] = \begin{bmatrix} 0 & 0 \\ 0 & \Sigma_{ss}^{\text{tie}} \end{bmatrix}, \qquad \Sigma_{ss}^{\text{tie}} \succeq 0.$$

**Example.** For hotel recommendations: $y_A$ = "Excellent service, prime location" (concise); $y_B$ = "Excellent service, prime location, with detailed amenities list" (verbose). Both have equal utility but differ in length, a common spurious feature.

## D.2. Expected Gradient from Ties

We now derive the expected gradient contribution of tie examples. Under Assumption A.1, the linearized *weight function* is:

$$w(\Delta_\theta) = 1 - \sigma(\beta\tilde{\theta}^\top\Delta\phi) \approx \frac{1}{2} - \frac{\beta}{4}\tilde{\theta}^\top\Delta\phi.$$

**Expected tie gradient.** Then, for a single tie example, the DPO gradient is

$$\nabla_\theta\ell_{\text{tie}} = -\beta\big(1 - \sigma(\beta\tilde{\theta}^\top\Delta\phi)\big)\Delta\phi.$$

Taking expectation over random tie labeling and using $\mathbb{E}_{\text{tie}}[\Delta\phi] = 0$ gives

$$\begin{aligned} g_{\text{tie}}(\tilde{\theta}) &= -\beta\,\mathbb{E}_{\text{tie}}\Big[\big(1 - \sigma(\beta\tilde{\theta}^\top\Delta\phi)\big)\Delta\phi\Big] \\ &= -\beta\,\text{Cov}_{\text{tie}}\Big(\Delta\phi,\, 1 - \sigma(\beta\tilde{\theta}^\top\Delta\phi)\Big). \end{aligned}$$

The second equality uses $\mathbb{E}_{\text{tie}}[\Delta\phi] = 0$, so $\mathbb{E}[w \cdot \Delta\phi] = \text{Cov}(\Delta\phi, w)$. Under the local regime (Assumption A.1), this gives

$$g_{\text{tie}}(\tilde{\theta}) \approx \frac{\beta^2}{4}\Sigma^{\text{tie}}\,\tilde{\theta}.$$

## D.3. Linearized Tie Dynamics

We now characterize the structural effect of ties on the population gradient.

**Tie covariance.** Let us define the tie covariance matrix as

$$\Sigma^{\text{tie}} := \mathbb{E}_{\text{tie}}[\Delta\phi\Delta\phi^\top] \in \mathbb{R}^{d\times d}.$$

By the tie construction (Appendix D.1), its block structure is

$$\Sigma^{\text{tie}} = \begin{bmatrix} 0 & 0 \\ 0 & \Sigma_{ss}^{\text{tie}} \end{bmatrix}, \qquad \Sigma_{ss}^{\text{tie}} \succeq 0.$$

We assume $\Sigma_{ss}^{\text{tie}} \succeq 0$. When $\Sigma_{ss}^{\text{tie}} \succ 0$, tie training provides curvature in all spurious directions. In general, $\Sigma_{ss}^{\text{tie}}$ may be low-rank, in which case the additional suppression from ties acts only along the subspace spanned by the tie perturbations, while the remaining spurious directions are regularized by the strict preference data.

**Interpretation.** To first order, the tie gradient $g_{\text{tie}}(\tilde\theta) \approx \frac{\beta^2}{4}\Sigma^{\text{tie}}\tilde\theta$ acts as a data-dependent quadratic regularizer on the spurious parameters while leaving causal parameters unaffected. The regularization strength is controlled by $\Sigma_{ss}^{\text{tie}}$, which is a design choice determined by the tie distribution.

### D.4. Mixed Training Equilibrium

We now solve for the equilibrium parameters under mixed strict–tie training. We start by defining the data model.

**Mixed data model.** We consider training on a mixture of strict preference pairs from distribution $P$ and tie pairs from $P_{\text{tie}}$. Let $\alpha \in (0,1)$ denote the fraction of strict preferences, and define the mixed distribution

$$P_{\text{mix}} := \alpha P + (1-\alpha)P_{\text{tie}}.$$

All expectations below are taken with respect to $P_{\text{mix}}$ unless stated otherwise.

**Population gradient under mixed training.** Under the above data model and local regime (Assumption A.1), the population gradient decomposes linearly:

$$\nabla_\theta L_{\text{mix}}(\tilde\theta) = \alpha\, g_P(\tilde\theta) + (1-\alpha)\, g_{\text{tie}}(\tilde\theta).$$

Using the local expansions

$$g_P(\tilde\theta) = -\frac{\beta}{2}\mu^{(P)} + \frac{\beta^2}{4}\Sigma^{(P)}\tilde\theta, \qquad g_{\text{tie}}(\tilde\theta) = \frac{\beta^2}{4}\Sigma^{\text{tie}}\tilde\theta,$$

we obtain

$$\nabla_\theta L_{\text{mix}}(\tilde\theta) = -\alpha\frac{\beta}{2}\mu^{(P)} + \frac{\beta^2}{4}\Sigma^{\text{mix}}\tilde\theta,$$

where the effective covariance is

$$\Sigma^{\text{mix}} := \alpha\Sigma^{(P)} + (1-\alpha)\Sigma^{\text{tie}}.$$

Since $\Sigma^{(P)} \succ 0$ and $\Sigma^{\text{tie}} \succeq 0$, the effective covariance satisfies $\Sigma^{\text{mix}} \succ 0$ for all $\alpha \in (0,1]$.

**Structure of the effective covariance.** Writing the block decomposition of the covariance

$$\Sigma^{\text{mix}} = \begin{bmatrix} \Sigma_{cc}^{\text{mix}} & \Sigma_{cs}^{\text{mix}} \\ \Sigma_{sc}^{\text{mix}} & \Sigma_{ss}^{\text{mix}} \end{bmatrix},$$

and using the tie construction ($\Sigma_{cc}^{\text{tie}} = \Sigma_{cs}^{\text{tie}} = 0$), we obtain

$$\Sigma_{cc}^{\text{mix}} = \alpha\,\Sigma_{cc}^{(P)}, \qquad \Sigma_{sc}^{\text{mix}} = \alpha\,\Sigma_{sc}^{(P)}, \qquad \Sigma_{ss}^{\text{mix}} = \alpha\,\Sigma_{ss}^{(P)} + (1-\alpha)\,\Sigma_{ss}^{\text{tie}}.$$

Thus tie training scales the raw causal–spurious second-moment blocks by $\alpha$ and adds positive semidefinite mass to the spurious block.

**Population equilibrium.** At the population equilibrium, we have $\nabla_\theta L_{\mathrm{mix}}(\tilde{\theta}^\star) = 0$, yielding

$$\alpha \frac{\beta}{2} \mu^{(P)} = \frac{\beta^2}{4} \Sigma^{\mathrm{mix}} \tilde{\theta}^\star_{\mathrm{mix}},$$

and therefore

$$\tilde{\theta}^\star_{\mathrm{mix}} = \frac{2\alpha}{\beta} (\Sigma^{\mathrm{mix}})^{-1} \mu^{(P)}. \tag{21}$$

**Spurious block.** Applying the same Schur complement algebra as in Appendix A.2 to the mixed system $\Sigma^{\mathrm{mix}} \tilde{\theta}^\star_{\mathrm{mix}} = \frac{2\alpha}{\beta} \mu^{(P)}$, the spurious component satisfies

$$\tilde{\theta}^\star_{s,\mathrm{mix}} = \frac{2\alpha}{\beta} (\Sigma^{\mathrm{mix}}_{ss})^{-1} \Big[ \big( I + \Sigma^{\mathrm{mix}}_{sc} (S^{\mathrm{mix}}_c)^{-1} \Sigma^{\mathrm{mix}}_{cs} (\Sigma^{\mathrm{mix}}_{ss})^{-1} \big) \mu^{(P)}_s - \Sigma^{\mathrm{mix}}_{sc} (S^{\mathrm{mix}}_c)^{-1} \mu^{(P)}_c \Big],$$

where $S^{\mathrm{mix}}_c := \Sigma^{\mathrm{mix}}_{cc} - \Sigma^{\mathrm{mix}}_{cs} (\Sigma^{\mathrm{mix}}_{ss})^{-1} \Sigma^{\mathrm{mix}}_{sc}$.

**Effect on the causal block.** Tie training does not introduce direct causal bias, since the tie construction contributes neither causal mean nor causal–spurious cross-moments. The effect on the causal block is indirect through the Schur complement.

### D.5. Proof of Theorem 6.2

Equipped with the mixed-training equilibrium (21) derived in Appendix D.4, we now provide the full proof of Theorem 6.2. To this end, we use the following regularity assumption on the interaction between the tie covariance and the spurious driving term.

**Assumption D.2** (Spurious driving term stability)**.** The spurious driving term satisfies

$$b_s(\Sigma^{\mathrm{mix}}, \mu^{(P)})^\top (\Sigma^{(P)}_{ss})^{-1} b_s(\Sigma^{\mathrm{mix}}, \mu^{(P)}) \ \leq \ b_s(\Sigma^{(P)}, \mu^{(P)})^\top (\Sigma^{(P)}_{ss})^{-1} b_s(\Sigma^{(P)}, \mu^{(P)}),$$

where $b_s(\Sigma, \mu) := (I + \Sigma_{sc} S_c^{-1} \Sigma_{cs} \Sigma_{ss}^{-1}) \mu_s - \Sigma_{sc} S_c^{-1} \mu_c$ and $S_c := \Sigma_{cc} - \Sigma_{cs} \Sigma_{ss}^{-1} \Sigma_{sc}$.

This regularity condition requires that the contraction of the raw causal–spurious block $\Sigma^{\mathrm{mix}}_{sc} = \alpha \Sigma^{(P)}_{sc}$ is not offset by amplification through the Schur complement and spurious-block inverses. In the scalar spurious case ($d_s = 1$), the theorem below gives the stronger pointwise shift reduction whenever the corresponding scalar monotonicity condition $|b^{\mathrm{mix}}_s| \leq |b^{(P)}_s|$ holds. This condition is immediate in pure mean-bias settings with $\Sigma^{(P)}_{cs} = 0$, where $b^{\mathrm{mix}}_s = b^{(P)}_s = \mu_s$; in pure-leakage and mixed regimes it is the scalar instance of Assumption D.2. For general $d_s > 1$, Assumption D.2 rules out adversarial tie designs that increase the effective spurious driving term despite adding spurious curvature.

**Theorem D.3** (Tie training reduces spurious reliance and deployment shift)**.** *Assume the conditions of Theorem 5.3, the tie construction in Appendix D.1, and Assumption D.2. Then:*

*(i)* **Spurious shrinkage (population level).** *Let $\theta^\star$ denote the strict-only population optimizer under $P$, and $\theta^\star_{\mathrm{mix}}$ the mixed-training optimizer under $P_{\mathrm{mix}} = \alpha P + (1 - \alpha) P_{\mathrm{tie}}$. Then*

$$\|\tilde{\theta}^\star_{s,\mathrm{mix}}\|_{\Sigma^{\mathrm{mix}}_{ss}} \ \leq \ \|\tilde{\theta}^\star_s\|_{\Sigma^{(P)}_{ss}},$$

*where $\Sigma^{\mathrm{mix}}_{ss} = \alpha \Sigma^{(P)}_{ss} + (1 - \alpha) \Sigma^{\mathrm{tie}}_{ss}$ and $\|v\|_A := \sqrt{v^\top A v}$ denotes the weighted norm. The $\Sigma_{ss}$-norm measures the contribution of spurious parameters to the variance of implicit reward margins; the theorem thus guarantees that tie training reduces the reward-variance attributable to spurious features. The inequality is strict whenever tie curvature acts on the mixed spurious driving direction and the driving-term stability bound is not tight.*

*(ii)* **Shift reduction under stable causal statistics.** *Suppose $\mu^{(Q)}_c = \mu^{(P)}_c$ and let $\delta \mu_s := \mu^{(Q)}_s - \mu^{(P)}_s$.*

*Scalar case ($d_s = 1$). If the scalar driving term is monotone under mixing, i.e.,*

$$|b^{\mathrm{mix}}_s| \leq |b^{(P)}_s|,$$

*then the first-order deployment shift satisfies*

$$\left| \tilde{\theta}^{\star}_{s,\text{mix}} \delta\mu_s \right| \;\leq\; \left| \tilde{\theta}^{\star}_{s} \delta\mu_s \right|,$$

*up to the same local-regime remainder $O(\beta^2 \|\tilde{\theta}\|^2)$.*

*General case ($d_s \geq 1$). The first-order spurious shift contribution satisfies*

$$\left| \tilde{\theta}^{\star\top}_{s,\text{mix}} \delta\mu_s \right| \;\leq\; \|\tilde{\theta}^{\star}_s\|_{\Sigma^{(P)}_{ss}} \cdot \|\delta\mu_s\|_{(\Sigma^{(P)}_{ss})^{-1}},$$

*and the right-hand side is controlled by the same-matrix norm shrinkage established in the proof below.*

*(iii) Deployment bound. Let $\hat{\theta}^{\text{mix}}$ be the ridge-regularized estimator trained on $P_{\text{mix}}$ with $n$ samples. Then, with probability at least $1 - \delta$,*

$$\text{SubOpt}_Q(\hat{\theta}^{\text{mix}}) \leq \underbrace{\tilde{J}^{(Q)}(\theta^{\star}_Q) - \tilde{J}^{(Q)}(\theta^{\star}_{\text{mix}})}_{\textit{reduced shift}} + \underbrace{G_{Q,\text{mix}}\Gamma_{n,\text{mix}} + \frac{\kappa_\Pi}{2}\Gamma^2_{n,\text{mix}}}_{\textit{estimation}}.$$

*Proof.* Let $P_{\text{mix}} := \alpha P + (1 - \alpha)P_{\text{tie}}$ be the mixed training distribution, and let $\theta^{\star}_{\text{mix}}$ denote the corresponding population optimizer (equilibrium) in the local regime. Let $\theta^{\star}$ denote the strict-only population optimizer under $P$.

We use the mixed linearized equilibrium derived in Appendix D.4:

$$\tilde{\theta}^{\star}_{\text{mix}} = \frac{2\alpha}{\beta}\left(\Sigma^{\text{mix}}\right)^{-1}\mu^{(P)}, \qquad \Sigma^{\text{mix}} := \alpha\Sigma^{(P)} + (1 - \alpha)\Sigma^{\text{tie}}, \tag{22}$$

and the strict-only equilibrium

$$\tilde{\theta}^{\star} = \frac{2}{\beta}\left(\Sigma^{(P)}\right)^{-1}\mu^{(P)}. \tag{23}$$

From the Schur complement decomposition (Appendix D.4), the spurious components satisfy

$$\tilde{\theta}^{\star}_{s,\text{mix}} = \frac{2\alpha}{\beta}\left(\Sigma^{\text{mix}}_{ss}\right)^{-1} b^{\text{mix}}_s, \qquad \tilde{\theta}^{\star}_s = \frac{2}{\beta}\left(\Sigma^{(P)}_{ss}\right)^{-1} b^{(P)}_s,$$

where $b^{\text{mix}}_s := b_s(\Sigma^{\text{mix}}, \mu^{(P)})$ and $b^{(P)}_s := b_s(\Sigma^{(P)}, \mu^{(P)})$.

**Proof of Part (i): spurious weight reduction.**

**Step 1: PSD inverse bound.** Since $\Sigma^{\text{mix}}_{ss} = \alpha\,\Sigma^{(P)}_{ss} + (1-\alpha)\,\Sigma^{\text{tie}}_{ss} \succeq \alpha\,\Sigma^{(P)}_{ss}$, we have $(\Sigma^{\text{mix}}_{ss})^{-1} \preceq \frac{1}{\alpha}(\Sigma^{(P)}_{ss})^{-1}$. Therefore

$$
\begin{aligned}
\|\tilde{\theta}^{\star}_{s,\text{mix}}\|^2_{\Sigma^{\text{mix}}_{ss}} &= \left(\frac{2\alpha}{\beta}\right)^2 (b^{\text{mix}}_s)^\top (\Sigma^{\text{mix}}_{ss})^{-1} b^{\text{mix}}_s \\
&\leq \left(\frac{2\alpha}{\beta}\right)^2 \cdot \frac{1}{\alpha}\,(b^{\text{mix}}_s)^\top (\Sigma^{(P)}_{ss})^{-1} b^{\text{mix}}_s \\
&= \alpha\left(\frac{2}{\beta}\right)^2 (b^{\text{mix}}_s)^\top (\Sigma^{(P)}_{ss})^{-1} b^{\text{mix}}_s.
\end{aligned}
$$

**Step 2: Driving term stability.** By Assumption D.2,

$$(b^{\text{mix}}_s)^\top (\Sigma^{(P)}_{ss})^{-1} b^{\text{mix}}_s \;\leq\; (b^{(P)}_s)^\top (\Sigma^{(P)}_{ss})^{-1} b^{(P)}_s.$$

**Combining.** Using $\alpha \leq 1$:

$$\|\tilde{\theta}^{\star}_{s,\text{mix}}\|^2_{\Sigma^{\text{mix}}_{ss}} \;\leq\; \alpha\left(\frac{2}{\beta}\right)^2 (b^{(P)}_s)^\top (\Sigma^{(P)}_{ss})^{-1} b^{(P)}_s \;\leq\; \|\tilde{\theta}^{\star}_s\|^2_{\Sigma^{(P)}_{ss}}.$$

**Strictness.** If $\Sigma_{ss}^{\text{tie}}$ adds positive curvature along a nonzero component of $b_s^{\text{mix}}$, the PSD bound in Step 1 is strict along that direction, yielding a strict inequality.

**Proof of Part (ii):**

**Intermediate result: same-matrix norm reduction.** We first establish that

$$\|\tilde{\theta}_{s,\text{mix}}^{\star}\|_{\Sigma_{ss}^{(P)}} \;\leq\; \|\tilde{\theta}_s^{\star}\|_{\Sigma_{ss}^{(P)}}. \tag{24}$$

Let $A := \Sigma_{ss}^{\text{mix}}$ and $B := \Sigma_{ss}^{(P)}$. From the equilibrium expression:

$$\|\tilde{\theta}_{s,\text{mix}}^{\star}\|_B^2 = \left(\frac{2\alpha}{\beta}\right)^2 (b_s^{\text{mix}})^\top A^{-1} B\, A^{-1}\, b_s^{\text{mix}}.$$

Since $A = \alpha B + (1-\alpha)\Sigma_{ss}^{\text{tie}} \succeq \alpha B$, we have $B \preceq \frac{1}{\alpha}A$. Applying congruence by $A^{-1/2}$:

$$A^{-1/2}\, B\, A^{-1/2} \preceq \frac{1}{\alpha}\, I,$$

so

$$A^{-1}\, B\, A^{-1} = A^{-1/2}\Big(A^{-1/2}\, B\, A^{-1/2}\Big)A^{-1/2} \preceq \frac{1}{\alpha} A^{-1} \preceq \frac{1}{\alpha^2} B^{-1},$$

where the last step uses $A^{-1} \preceq \frac{1}{\alpha}B^{-1}$ from the PSD inverse bound in Part (i). Substituting:

$$\|\tilde{\theta}_{s,\text{mix}}^{\star}\|_B^2 \leq \left(\frac{2\alpha}{\beta}\right)^2 \cdot \frac{1}{\alpha^2}\, (b_s^{\text{mix}})^\top B^{-1} b_s^{\text{mix}} = \left(\frac{2}{\beta}\right)^2 (b_s^{\text{mix}})^\top B^{-1} b_s^{\text{mix}}.$$

By Assumption D.2:

$$(b_s^{\text{mix}})^\top B^{-1} b_s^{\text{mix}} \leq (b_s^{(P)})^\top B^{-1} b_s^{(P)},$$

so

$$\|\tilde{\theta}_{s,\text{mix}}^{\star}\|_{\Sigma_{ss}^{(P)}}^2 \leq \left(\frac{2}{\beta}\right)^2 (b_s^{(P)})^\top (\Sigma_{ss}^{(P)})^{-1} b_s^{(P)} = \|\tilde{\theta}_s^{\star}\|_{\Sigma_{ss}^{(P)}}^2,$$

establishing (24).

**Scalar case ($d_s = 1$).** When $d_s = 1$, all spurious quantities are scalar. From the equilibrium expressions:

$$|\tilde{\theta}_{s,\text{mix}}^{\star}| = \frac{2\alpha}{\beta}\frac{|b_s^{\text{mix}}|}{\Sigma_{ss}^{\text{mix}}}, \qquad |\tilde{\theta}_s^{\star}| = \frac{2}{\beta}\frac{|b_s^{(P)}|}{\Sigma_{ss}^{(P)}}.$$

Since $\Sigma_{ss}^{\text{mix}} = \alpha\,\Sigma_{ss}^{(P)} + (1-\alpha)\,\Sigma_{ss}^{\text{tie}} \geq \alpha\,\Sigma_{ss}^{(P)}$:

$$|\tilde{\theta}_{s,\text{mix}}^{\star}| \leq \frac{2\alpha}{\beta}\frac{|b_s^{\text{mix}}|}{\alpha\,\Sigma_{ss}^{(P)}} = \frac{2}{\beta}\frac{|b_s^{\text{mix}}|}{\Sigma_{ss}^{(P)}}.$$

Under the condition $|b_s^{\text{mix}}| \leq |b_s^{(P)}|$:

$$|\tilde{\theta}_{s,\text{mix}}^{\star}| \leq \frac{2}{\beta}\frac{|b_s^{(P)}|}{\Sigma_{ss}^{(P)}} = |\tilde{\theta}_s^{\star}|.$$

The shift inequality follows immediately:

$$|\tilde{\theta}_{s,\text{mix}}^{\star}\,\delta\mu_s| = |\tilde{\theta}_{s,\text{mix}}^{\star}|\,|\delta\mu_s| \leq |\tilde{\theta}_s^{\star}|\,|\delta\mu_s| = |\tilde{\theta}_s^{\star}\,\delta\mu_s|.$$

**General case** ($d_s \geq 1$).  For general $d_s$, Cauchy–Schwarz in the $\Sigma_{ss}^{(P)}$-inner product gives

$$|\tilde{\theta}_{s,\text{mix}}^{\star\top}\delta\mu_s| \leq \|\tilde{\theta}_{s,\text{mix}}^\star\|_{\Sigma_{ss}^{(P)}} \cdot \|\delta\mu_s\|_{(\Sigma_{ss}^{(P)})^{-1}}.$$

By the same-matrix norm reduction (24):

$$|\tilde{\theta}_{s,\text{mix}}^{\star\top}\delta\mu_s| \leq \|\tilde{\theta}_s^\star\|_{\Sigma_{ss}^{(P)}} \cdot \|\delta\mu_s\|_{(\Sigma_{ss}^{(P)})^{-1}}.$$

This bounds the mixed-training shift by the strict-only worst-case envelope: the right-hand side equals the maximum of $|\tilde{\theta}_s^{\star\top}v|$ over all deployment shifts $v$ with $\|v\|_{(\Sigma_{ss}^{(P)})^{-1}} = \|\delta\mu_s\|_{(\Sigma_{ss}^{(P)})^{-1}}$, so tie training guarantees the mixed model stays within this envelope.

**Proof of Part (iii): deployment bound.**  Apply Theorem C.8 with the training distribution replaced by $P_{\text{mix}}$ and the corresponding population optimizer $\theta_{\text{mix}}^\star$ and ridge-MLE $\hat{\theta}^{\text{mix}}$. The same proof yields, with probability $1 - \delta$,

$$\text{SubOpt}_Q(\hat{\theta}^{\text{mix}}) \leq \underbrace{\tilde{J}^{(Q)}(\theta_Q^\star) - \tilde{J}^{(Q)}(\theta_{\text{mix}}^\star)}_{\text{shift}(P_{\text{mix}})} + \underbrace{G_{Q,\text{mix}}\Gamma_{n,\text{mix}} + \frac{\kappa_\Pi}{2}\Gamma_{n,\text{mix}}^2}_{\text{estimation}(P_{\text{mix}})},$$

where $\Gamma_{n,\text{mix}}$ is the same self-normalized radius but computed with the mixed-training curvature $H_{P_{\text{mix}}}$ (and the same $\beta, \lambda, B$). This proves Part (iii). □

## D.6. Quantitative Reduction Bound

We conclude this appendix by providing the full proof of Corollary 6.3. For completeness, we restate the corollary below and then provide the proof.

**Corollary D.4** (Quantitative reduction under isotropic ties). *Under the conditions of Theorem D.3, if ties are isotropic, i.e., $\Sigma_{ss}^{\text{tie}} = \sigma^2 I_{d_s}$, and $\Sigma_{ss}^{(P)} = \lambda_0 I_{d_s}$, then*

$$\frac{\|\tilde{\theta}_{s,\text{mix}}^\star\|_2}{\|\tilde{\theta}_s^\star\|_2} \leq \frac{\alpha\lambda_0}{\alpha\lambda_0 + (1-\alpha)\sigma^2}.$$

*Proof.* Under isotropy,

$$\Sigma_{ss}^{(P)} = \lambda_0 I, \qquad \Sigma_{ss}^{\text{tie}} = \sigma^2 I,$$

so

$$\Sigma_{ss}^{\text{mix}} = \alpha\lambda_0 I + (1-\alpha)\sigma^2 I = (\alpha\lambda_0 + (1-\alpha)\sigma^2)I.$$

Hence

$$(\Sigma_{ss}^{\text{mix}})^{-1} = \frac{1}{\alpha\lambda_0 + (1-\alpha)\sigma^2}I.$$

From the equilibrium expressions,

$$\tilde{\theta}_{s,\text{mix}}^\star = \frac{2\alpha}{\beta}\frac{1}{\alpha\lambda_0 + (1-\alpha)\sigma^2}b_s^{\text{mix}}, \qquad \tilde{\theta}_s^\star = \frac{2}{\beta}\frac{1}{\lambda_0}b_s^{(P)}.$$

Therefore

$$\frac{\|\tilde{\theta}_{s,\text{mix}}^\star\|_2}{\|\tilde{\theta}_s^\star\|_2} = \frac{\alpha\lambda_0}{\alpha\lambda_0 + (1-\alpha)\sigma^2} \cdot \frac{\|b_s^{\text{mix}}\|_2}{\|b_s^{(P)}\|_2}.$$

Under Assumption D.2, which reduces to Euclidean contraction in the isotropic case,

$$\|b_s^{\text{mix}}\|_2 \leq \|b_s^{(P)}\|_2,$$

yielding the claimed bound. □

# E. Distribution Shift Scenarios

This appendix provides detailed analysis of the three shift scenarios from Section 5. Throughout, we assume $\tilde{\theta}_{s,\text{train}} \neq 0$ and stable causal statistics ($\mu_c^{(Q)} \approx \mu_c^{(P)}$).

## E.1. Suppression ($\mu_s^{(Q)} = 0$)

Spurious correlation absent at deployment while $\mu_s^{(P)} \neq 0$.

**Margin behavior.** $m_Q^{\text{spur}} = \beta \tilde{\theta}_{s,\text{train}}^\top \cdot 0 = 0$. Systematic spurious bias vanishes.

**Variance-induced accuracy degradation.** Zero mean does not imply robustness. The spurious variance

$$\beta^2 \tilde{\theta}_{s,\text{train}}^\top \Sigma_{ss}^{(Q)} \tilde{\theta}_{s,\text{train}} > 0$$

adds noise to predictions.

**Examples.**

- **Hotels:** Training has longer reviews for quality hotels; test set balanced $\Rightarrow$ length adds noise.
- **Code:** Training has verbose correct solutions; test balanced $\Rightarrow$ concise correct solutions underrated.
- **Safety:** Training has hedged safe responses; test balanced $\Rightarrow$ direct safe responses underrated.

## E.2. Adversarial Reversal ($\mu_s^{(Q)} = -\mu_s^{(P)}$)

Spurious correlation flips sign.

**Margin behavior.**

$$m_Q^{\text{spur}} = -m_P^{\text{spur}} \quad \Rightarrow \quad \Delta m^{\text{spur}} = -2m_P^{\text{spur}}.$$

If spurious features helped at training, they hurt equally at deployment.

**Worst-case analysis.** For constrained $\|\mu_s^{(Q)}\| \leq R$:

$$\min_{\|\mu_s^{(Q)}\| \leq R} m_Q^{\text{spur}} = -\beta \|\tilde{\theta}_{s,\text{train}}\| R,$$

attained when $\mu_s^{(Q)} \propto -\tilde{\theta}_{s,\text{train}}$ (adversarial direction).

**Examples.**

- **Hotels:** US prefers long reviews; international region has long complaint reviews $\Rightarrow$ low-quality preferred.
- **Code:** Expert code is verbose; beginner incorrect code is also verbose $\Rightarrow$ incorrect preferred.
- **Safety:** Safe responses hedge; adversarial unsafe responses hedge more $\Rightarrow$ unsafe rated as safe.

Spurious correlation changes direction (relevant when $d_s > 1$).

**Alignment analysis.** The spurious margin depends on alignment:

$$m_Q^{\text{spur}} = \beta \|\tilde{\theta}_{s,\text{train}}\| \cdot \|\mu_s^{(Q)}\| \cdot \cos\alpha,$$

where $\psi = \angle(\tilde{\theta}_{s,\text{train}}, \mu_s^{(Q)})$.

| $\cos\psi$ | Effect |
|---|---|
| $+1$ | Aligned: no degradation |
| $0$ | Orthogonal: spurious noise only |
| $-1$ | Opposite: maximum degradation (reversal) |

**Examples.**

- **Hotels:** Length correlates positively in US, negatively in Europe; star rating correlates oppositely $\Rightarrow$ partial misalignment.

- **Code:** Comment density and line count have different correlation patterns across languages $\Rightarrow$ rotation.

- **Safety:** Formality and hedging evolve differently over time $\Rightarrow$ temporal rotation.

# F. Experimental Details and Additional Results

This appendix provides a comprehensive empirical validation of the theoretical results presented in the main paper, progressing from settings that exactly match the theory to increasingly realistic and expressive model classes. We begin with linear preference models, where the assumptions of the theory hold and we obtain precise quantitative agreement with predicted bounds and scaling laws. We then move to nonlinear neural networks, where the causal–spurious decomposition is hidden inside nonlinear representations, and finally to large language models trained with DPO on synthetic preference data. While exact quantitative predictions no longer apply beyond the linear regime, we demonstrate that the core qualitative mechanisms identified by the theory, spurious correlation learning, irreducible deployment error under distribution shift, and mitigation via tie training, persist across all stages. Together, these experiments show that the linear analysis captures essential dynamics of preference learning systems even when deployed with rich nonlinear models. Across all experiments, shaded regions denote $\pm 1$ standard deviation. The number of seeds varies by setting, with many runs in the linear and deployment-error experiments and fewer runs for the costly LLM experiments. We report the exact number in the corresponding hyperparameter table for each setting. Code is available at https://github.com/cmoyacal/tie-training.

## F.1. Linear Models (Theoretical Ground Truth)

### F.1.1. DATASET CONSTRUCTION

**Feature decomposition.** We construct a synthetic preference dataset with an explicit causal–spurious feature decomposition. Each example is represented by a Gaussian feature vector $\phi = [\phi_c; \phi_s]$, where $\phi_c$ denotes causal features and $\phi_s$ denotes spurious features. Causal features determine true preference utility, while spurious features do not affect utility. Under the training distribution $P$, spurious features are correlated with causal features. At deployment, this correlation changes under a shifted distribution $Q$.

**Strict preference pairs.** Strict preference pairs are generated by

$$P(y = 1 \mid \Delta\phi) = \sigma(\beta\,\theta^{\dagger\top}\Delta\phi),$$

where $\theta^\dagger$ decomposes into causal and spurious components. In the experiments, we use $\beta = 1.0$. The preferences depend on the full feature difference, inducing spurious correlations when $\phi_s$ is correlated with $\phi_c$. Spurious features are drawn as

$$\phi_s \sim \mathcal{N}(0, \lambda_0 I),$$

where $\lambda_0$ controls the variance scale of spurious features in strict data. This construction ensures that spurious cues are predictive during training despite being non-causal.

**Tie examples.** Tie examples are defined by a small causal margin

$$|\theta_c^{\dagger\top}\Delta\phi_c| \le \tau,$$

which produces pairs with weak causal signal. Labels for ties are randomized as $y \sim \text{Bernoulli}(0.5)$, removing any causal dependence. Spurious features for ties are drawn from

$$\phi_s \sim \mathcal{N}(0, \sigma^2 I),$$

where $\sigma^2$ controls the variance injected by tie data. This construction decorrelates spurious features from labels while amplifying spurious variance, creating targeted negative evidence against spurious reliance.

### F.1.2. MODEL AND OBJECTIVE

**Model class.** We train a log-linear preference model

$$r_\theta(\phi) = \theta^\top \phi,$$

which matches the assumptions of the theoretical analysis. The parameter vector $\theta$ decomposes into causal and spurious components. This setting allows direct measurement of spurious reliance through parameter norms. As a result, population predictions can be tested exactly.

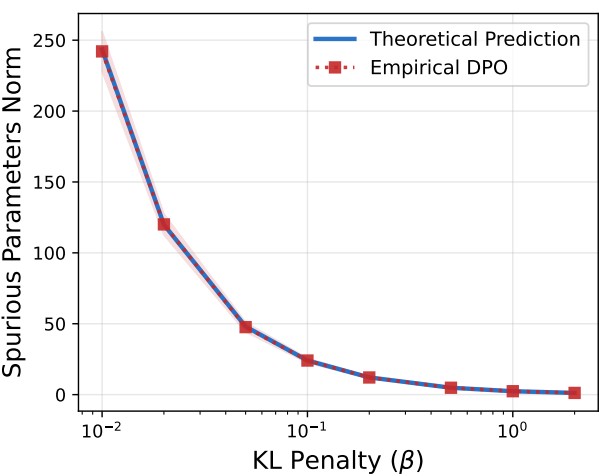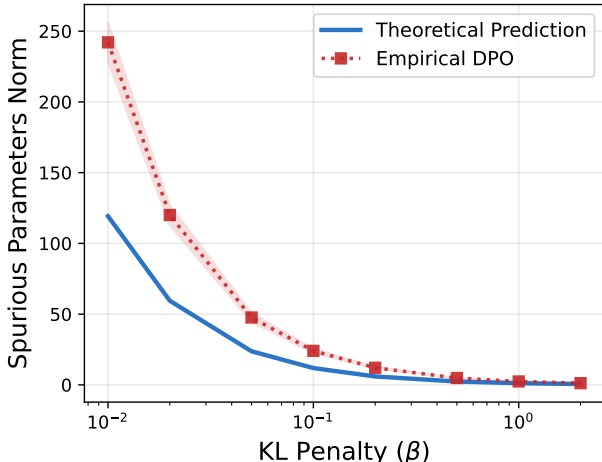

*Figure 3.* Population scaling of spurious parameters in DPO. We compare empirical spurious parameter norms with the population prediction from **Theorem 4.1**. *Left:* Including curvature yields accurate predictions across $\beta$. *Right:* Ignoring curvature systematically underestimates spurious reliance, leading to large relative error even with infinite data. This confirms that curvature helps correct population scaling when the local regime assumption (Assumption A.1) is not valid.

**Training objective (DPO-style).** Training minimizes a DPO-style pairwise logistic loss

$$\min_\theta \ \mathbb{E}\Big[\log\Big(1 + e^{-\beta y \theta^\top \Delta\phi}\Big)\Big].$$

The parameter $\beta$ controls the sharpness of preference separation. We vary the fraction of strict versus tie examples using $\alpha$. This setup exactly matches the assumptions of Theorems 4.1, 5.3, and 6.2. The same parameterization also subsumes RLHF-style reward learning, which corresponds to this objective with $\beta = 1$ and zero reference. We show in Appendix G that our mechanisms and shift vulnerabilities therefore apply to standard reward modeling as well.

### F.1.3. METRICS

**Spurious reliance.** We measure spurious reliance using the norm $\|\theta_s\|$. This quantity directly quantifies how much the learned model relies on spurious features. Because the model is linear, this metric has a clear population interpretation. It provides a precise test of theoretical predictions.

**Reduction ratio.** We compare empirical and theoretical reductions in spurious reliance under tie training. Empirically, we compute

$$r_{\text{emp}} = \frac{\|\theta_{s,\text{mix}}\|}{\|\theta_s\|}.$$

The theory predicts

$$r_{\text{th}} = \frac{\alpha\lambda_0}{\alpha\lambda_0 + (1-\alpha)\sigma^2},$$

which depends on the strict fraction $\alpha$ and the spurious variance ratio $\sigma^2/\lambda_0$.

### F.1.4. RESULTS

**Spurious correlation learning** Strict-only training learns nonzero spurious parameters. Figure 3 compares empirical spurious parameter norms to the population prediction from Theorem 4.1. Empirically, we found that correcting for curvature leads to the correct scaling across $\beta$.

*Remark* F.1 (Curvature-corrected linear prediction). The local linear approximation replaces the logit curvature of the logistic loss by its value at the origin, $\sigma'(0) = 1/4$, giving $\theta_{\text{lin}} = \frac{2}{\beta}\Sigma^{-1}\mu$ with $\mu = \mathbb{E}[\Delta\phi]$ and $\Sigma = \mathbb{E}[\Delta\phi\Delta\phi^\top]$. This is accurate when $|\beta\theta^\top\Delta\phi|$ is small. Away from this regime, the loss no longer has effective logit curvature $1/4$: logistic saturation reduces the average curvature to $\zeta = \mathbb{E}[\sigma(\beta\theta_{\text{DPO}}^\top\Delta\phi)(1 - \sigma(\beta\theta_{\text{DPO}}^\top\Delta\phi))]$. We therefore use the curvature-corrected

*Table 2.* Key hyperparameters for the log-linear mechanism experiments. Remaining settings follow the defaults in our released code.

| Hyperparameter | Value |
| --- | --- |
| *Data-generating process* | |
| Causal feature dim $d_c$ | 5 |
| Spurious feature dim $d_s$ | 5 |
| Training samples $n$ | 10,000 |
| KL strength $\beta$ | $\{10^{-2},\ 10^{-1},\ 10^{0}\}$ |
| *Training (SGD ablation)* | |
| Optimizer | SGD on DPO loss |
| Learning rate $\eta$ | $5 \times 10^{-3}$ |
| Iterations | 15,000 |
| KL strength $\beta$ | 0.5 |
| DGP levels | NO_BT, BT_CAUSAL, BT_FULL |
| Seeds | 30 |

approximation $\theta_{\mathrm{corr}} = \frac{1/4}{\zeta}\theta_{\mathrm{lin}}$. When logits remain near zero, $\zeta \approx 1/4$ and the correction is negligible. In a more saturated regime, $\zeta < 1/4$, so the same mean bias $\mu$ produces a larger displacement; the factor $(1/4)/\zeta$ corrects this curvature mismatch while preserving the direction predicted by the linear local theory.

**SGD ablation: monitoring the local regime.** We perform a stochastic-gradient ablation that exposes the dynamics underlying the equilibrium prediction. We consider three data-generating levels of increasing complexity. In NO_BT, the feature differences are drawn from a Gaussian with nonzero mean and arbitrary covariance. In BT_CAUSAL, features are Gaussian with a correlation $\rho$ between causal and spurious blocks, and the Bradley–Terry annotator depends only on the causal block ($\theta_s^\dagger = 0$). In BT_FULL, features are independent isotropic Gaussian and the Bradley–Terry annotator depends on both blocks ($\theta_s^\dagger \neq 0$). For each level we run SGD on the DPO loss with learning rate $\eta = 5 \times 10^{-3}$ for 15,000 iterations, repeated over 30 seeds with a single fixed ground truth. We fix $\beta = 0.5$ and use $d_c = d_s = 5$ with $n = 10,000$ training samples per run. To diagnose whether the linearized analysis applies along the trajectory, at every SGD step we record the worst-case magnitude $\max_i |\beta\, \theta_t^\top \Delta\phi_i|$ on a fixed 1,000-point subsample: this is the local-regime indicator, since the first-order Taylor expansion of the sigmoid that underlies the closed form $\tilde{\theta}^\star = (2/\beta)\Sigma^{-1}\mu$ holds quantitatively while this quantity remains below 1.

Figures 4–6 report three panels per setting: the spurious-norm trajectory $\|\tilde{\theta}_s(t)\|$ against the closed-form predictions, the margin diagnostic $\max_i |\beta\, \theta_t^\top \Delta\phi_i|$ with horizontal references at 0.5 and 1, and a bar comparison of the final spurious norm against the Linear, DPO, and (where applicable) curvature-corrected closed forms. In NO_BT (Fig. 4), the margin stays below 1 throughout training and the SGD trajectory rises toward the Linear/DPO equilibrium, with the empirical norm lying within sampling variability of the closed-form predictions at 15,000 iterations. In BT_CAUSAL (Fig. 5), the margin grows past the reference threshold during training, exiting the local regime; the trajectory overshoots and then drifts downward, and the final SGD norm lies above the closed-form predictions, indicating that the local regime no longer captures the late-time training dynamics. In BT_FULL with curvature correction (Fig. 6), the margin sits well above the boundary of the local regime; the curvature-corrected prediction ($\zeta$-corr) recovers the empirical norm where the uncorrected Linear bar undershoots. Table 2 reports key hyperparameters for the log-linear mechanism experiments.

**Deployment error.** Spurious learning induces a deployment vulnerability that persists under distribution shift. Figure 7 shows in-distribution ($P$) and out-of-distribution ($Q$) accuracy as a function of training set size $n$, in a setup where the spurious feature flips sign between $P$ and $Q$. In-distribution accuracy grows with $n$ while out-of-distribution accuracy plateaus far below it, illustrating the irreducibility of deployment failure under spurious shift. Figure 8 reports the same phenomenon for $\beta = 1$, validating Theorem C.8: as $n$ grows, the empirical suboptimality converges to the shift floor $\tilde{J}^{(Q)}(\theta_Q^\star) - \tilde{J}^{(Q)}(\theta_{\mathrm{train}})$ while the parameter-space estimation error $\|\hat{\theta} - \theta_{\mathrm{train}}\|_{H_P}$ decays at the expected $1/\sqrt{n}$ rate. The empirical suboptimality remains below the theoretical upper bound at every $n$, consistent with the theoretical guarantee. Table 3 reports key hyperparameters for the deployment-error experiments.

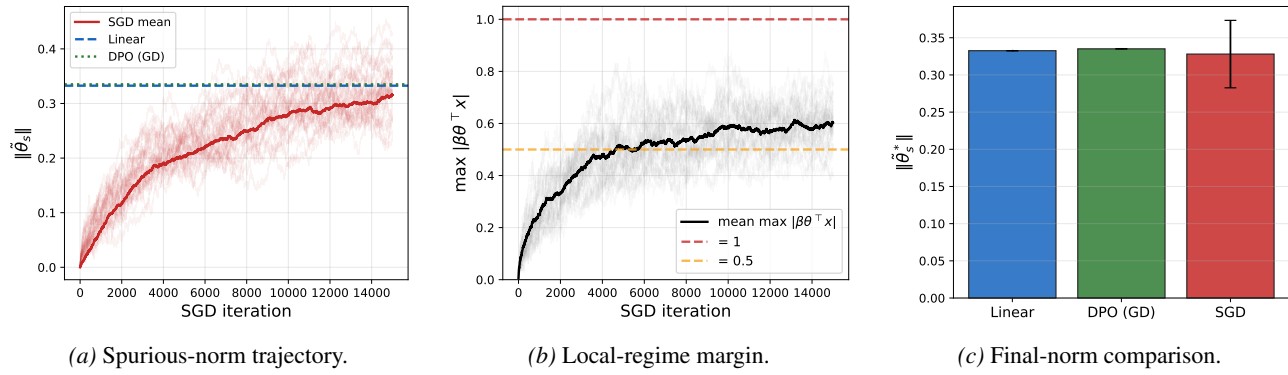

*(a)* Spurious-norm trajectory.    *(b)* Local-regime margin.    *(c)* Final-norm comparison.

*Figure 4.* SGD ablation, NO_BT. The margin stays below 1 throughout training and the trajectory approaches the Linear/DPO prediction.

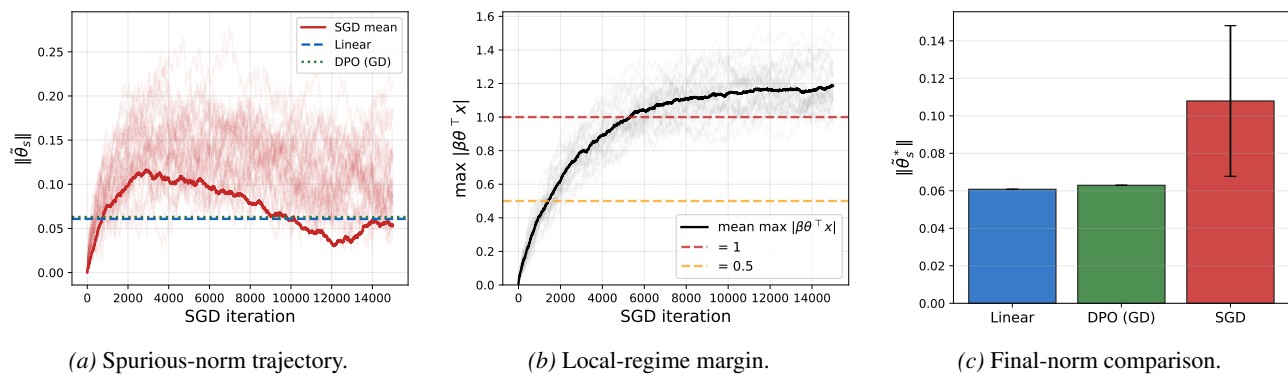

*(a)* Spurious-norm trajectory.    *(b)* Local-regime margin.    *(c)* Final-norm comparison.

*Figure 5.* SGD ablation, BT_CAUSAL. The margin grows past 1, the trajectory overshoots, and the final SGD norm lies above the closed-form predictions, reflecting departure from the local regime.

*Table 3.* Key hyperparameters for the deployment-error experiments. Remaining settings follow the defaults in our released code.

| Hyperparameter | Value |
| --- | --- |
| *Data-generating process* | |
| Causal feature dim $d_c$ | 5 |
| Spurious feature dim $d_s$ | 5 |
| Spurious correlation $\rho$ | 0.75 |
| Label model | $y \sim \mathrm{Bern}(\sigma(\theta_c^{\star\top} x_c))$ |
| $P$ vs $Q$ | sign flip on spurious block |
| *Training and evaluation* | |
| Training sizes $n$ | $\{200, 400, \ldots, 50{,}000\}$ |
| Test pairs per distribution | 20,000 |
| Ridge regularization $\lambda$ | $10^{-4}$ |
| Parameter ball radius $B$ | 1 |
| Confidence $\delta$ | 0.05 |
| Seeds | 20 |

### F.1.5. TIE TRAINING

Tie training suppresses spurious reliance in a predictable, population-level way. Figure 9 plots the theoretical reduction factor $r_{\mathrm{th}}(\alpha)$ for different spurious variance ratios $\sigma^2/\lambda_0$, showing monotone suppression as tie fraction increases. This prediction holds independently of sample size and isolates the irreducible effect of tie training on spurious learning. We use this curve to interpret empirical reductions in $\|\hat{\theta}_s\|$ across $\alpha$. Table 4 reports key hyperparameters for the tie-training experiments.

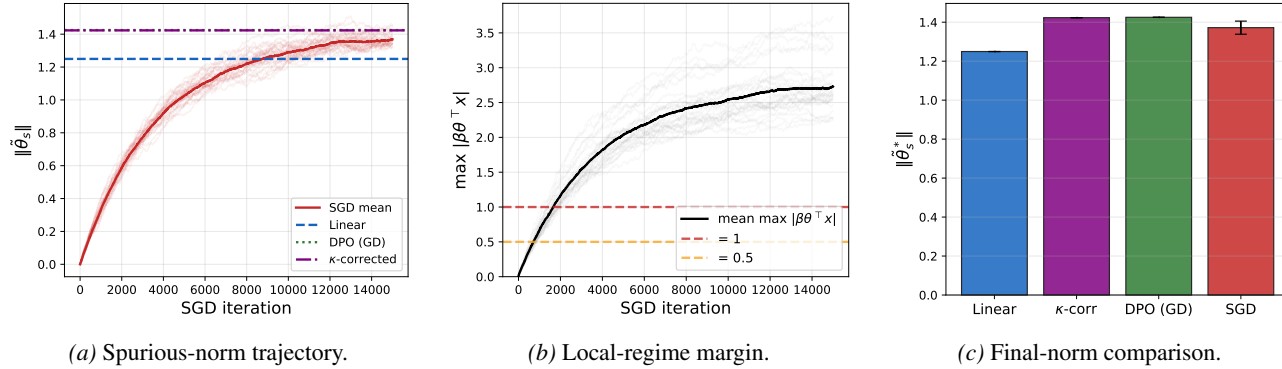

*(a)* Spurious-norm trajectory.      *(b)* Local-regime margin.      *(c)* Final-norm comparison.

*Figure 6.* SGD ablation, $\mathrm{BT\_FULL}$. The margin sits well above the boundary of the local regime; the curvature-corrected prediction ($\zeta$-corr) recovers the empirical norm where the uncorrected Linear bar undershoots.

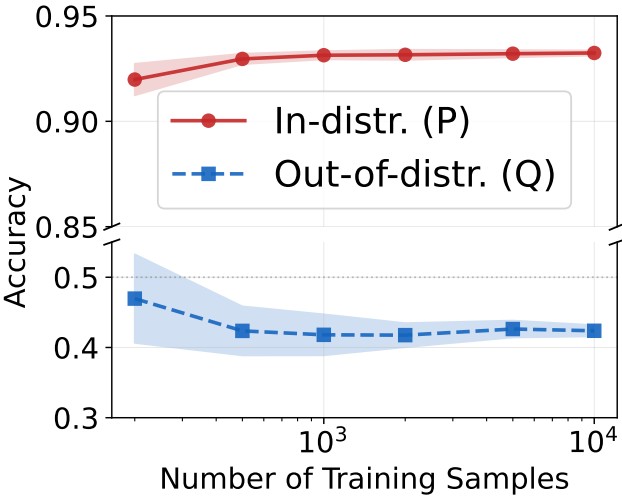

*Figure 7.* In-distribution ($P$) vs out-of-distribution ($Q$) accuracy under spurious shift, as a function of training set size $n$. Both test sets have the same size; under $Q$ the spurious feature flips sign. In-distribution accuracy grows mildly with $n$, while out-of-distribution accuracy plateaus far below it (just below the random-guess line), demonstrating that spurious reliance learned under $P$ cannot be fixed by additional training data from $P$.

*Table 4.* Key hyperparameters for the tie-training experiment. Remaining settings follow the defaults in our released code.

| Hyperparameter | Value |
|---|---|
| *Data-generating process* | |
| Causal feature dim $d_c$ | 5 |
| Spurious feature dim $d_s$ | 5 |
| Training samples $N$ | 8,000 |
| Spurious variance $\lambda_0$ | 1 |
| Spurious variance ratio $\sigma^2/\lambda_0$ | 5 |
| KL strength $\beta$ | 0.3 |
| *Mixing* | |
| Strict fraction $\alpha$ | $\{0.5, 0.6, 0.7, 0.8, 0.9, 1.0\}$ |
| Tie construction | soft (spurious-only, random labels) |
| $\ell_2$ regularization | 0 |
| Seeds | 30 |

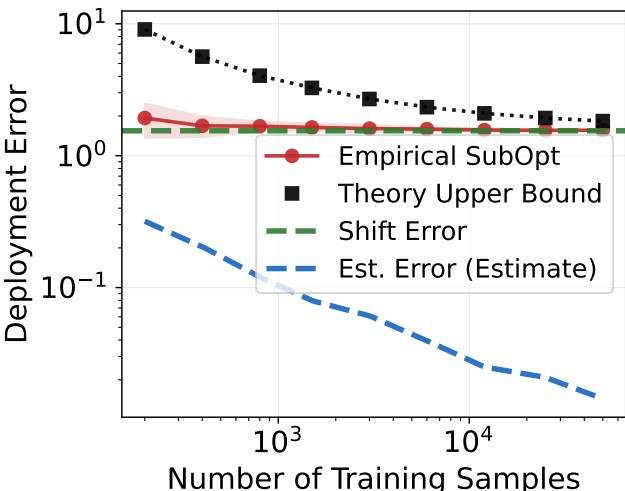

*Figure 8.* Empirical deployment suboptimality and its decomposition under distribution shift, for $\beta = 1$. The figure shows four quantities as a function of the number of training samples $n$: (i) empirical deployment suboptimality $\mathrm{SubOpt}_Q(\hat{\theta})$; (ii) shift error $\tilde{J}^{(Q)}(\theta_Q^\star) - \tilde{J}^{(Q)}(\theta_{\text{train}})$; (iii) measured parameter estimation error $\|\hat{\theta} - \theta_{\text{train}}\|_{H_P}$; (iv) theoretical upper bound from Theorem C.8. As $n$ grows, the estimation error decays at the expected $1/\sqrt{n}$ rate, demonstrating that deployment error is irreducible with additional training data from $P$. The empirical suboptimality remains below the theoretical upper bound at every $n$, consistent with the theoretical guarantee.

### F.1.6. LOG-LINEAR EXPERIMENT: CAUSAL DECONTAMINATION AND NEAR TIE ROBUSTNESS

**Intuition for causal decontamination.** Tie training does not necessarily shrink the causal parameters. Instead, it reduces the contamination of the causal block induced by causal–spurious coupling. This can be seen in the same block setting used in the log-linear experiments. Let

$$\Sigma = \begin{pmatrix} \Sigma_{cc} & \Sigma_{cs} \\ \Sigma_{sc} & \lambda_0 I \end{pmatrix}, \qquad \mu = \begin{pmatrix} \mu_c \\ \mu_s \end{pmatrix},$$

denote the second moment and mean of the feature differences. Adding isotropic ties in the spurious block gives mixed mean $\alpha\mu$ and effective spurious covariance $\alpha\lambda_0 I + (1-\alpha)\sigma^2 I$, since ties have zero mean. In the linearized equilibrium $\tilde{\theta}^\star = \frac{2}{\beta}\Sigma^{-1}\mu$, the causal block can then be written via the Schur complement as

$$\theta_{c,\text{mix}} = \frac{2}{\beta}\left(\Sigma_{cc} - \omega(\alpha)\,\Sigma_{cs}\Sigma_{sc}\right)^{-1}\left(\mu_c - \omega(\alpha)\,\Sigma_{cs}\mu_s\right), \qquad \omega(\alpha) = \frac{\alpha}{\alpha\lambda_0 + (1-\alpha)\sigma^2}.$$

Thus, in this isotropic setting, the influence of the spurious block on the causal solution enters through the single scalar $\omega(\alpha)$, which decreases from $1/\lambda_0$ toward 0 as the tie contribution $1 - \alpha$ increases. Smaller $\omega(\alpha)$ weakens the causal–spurious coupling, and in the limit $\omega(\alpha) \to 0$ the causal block coincides with the pure-causal solution obtained from the causal features alone,

$$\theta_c^{\text{pure}} = \frac{2}{\beta}\Sigma_{cc}^{-1}\mu_c.$$

This motivates the metric used below: we compare the causal block to the pure-causal solution rather than measuring the norm of the causal parameters, since the latter may move in either direction and does not capture decontamination.

**Experimental setup.** We sample feature differences directly, $\Delta\phi \sim \mathcal{N}(\mu, \Sigma)$. This lets us set the local regime by selecting $\|\mu\|$ small. We use $d_c = d_s = 5$, $\beta = 0.3$, isotropic blocks $\Sigma_{cc} = I$, $\Sigma_{ss} = \lambda_0 I$ with $\lambda_0 = 1$, and a diagonal causal–spurious coupling $\Sigma_{cs} = \rho I$ with $\rho = 0.5$. The mean $\mu$ has small nonzero entries in both blocks, so the spurious features leak into the strict-preference solution. Strict data are drawn from $\mathcal{N}(\mu, \Sigma)$. Tie data have $\Delta\phi_c = 0$ and $\Delta\phi_s \sim \mathcal{N}(0, \sigma^2 I)$ with $\sigma^2 = 5$. For each mixing level $\alpha$ we solve DPO on the pooled strict-plus-tie data to obtain $\theta_{c,\text{mix}}$, on the strict data alone for $\theta_{c,\text{strict}}$, and on the strict data with the spurious coordinates zeroed for $\theta_c^{\text{pure}}$. We report the relative distance

$$\frac{\|\theta_{c,\text{mix}} - \theta_c^{\text{pure}}\|_2}{\|\theta_{c,\text{strict}} - \theta_c^{\text{pure}}\|_2},$$

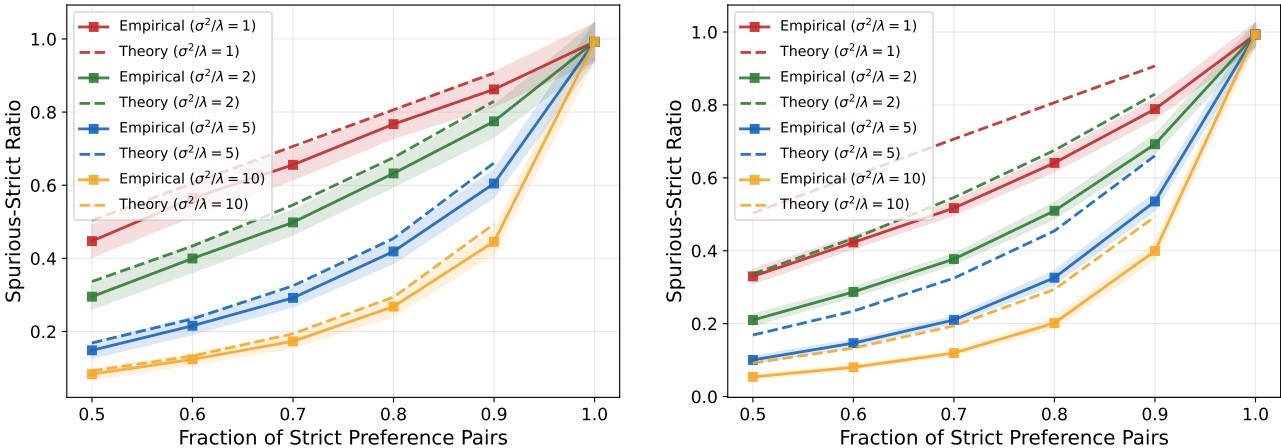

*Figure 9.* Theoretical prediction for spurious reliance under tie training. The curve shows the reduction factor $r_{\text{th}}(\alpha) = \frac{\alpha\lambda_0}{\alpha\lambda_0 + (1-\alpha)\sigma^2}$ as a function of the strict-preference fraction $\alpha$, for different spurious variance ratios $\sigma^2/\lambda_0$. Increasing the proportion of tie examples $(1-\alpha)$ monotonically suppresses reliance on spurious features, with stronger suppression when ties inject higher spurious variance. This bound holds independently of sample size and captures the irreducible effect of tie augmentation on spurious learning. KL-strength *Left:* $\beta = 0.3$ and *Right:* $\beta = 0.7$.

averaged over random seeds.

**Results.** Figure 10 shows that increasing the tie contribution $1-\alpha$ moves the causal block substantially closer to the pure-causal solution: the relative distance falls from 1 in the strict-only case ($\alpha = 1$) to roughly 0.2, closely tracking the isotropic factor $\alpha\lambda_0/(\alpha\lambda_0 + (1-\alpha)\sigma^2)$ predicted by the linearized solution (the small residual gap at large tie contribution is consistent with finite-sample variation in the distance estimate). In contrast, the causal *norm* ratio $\|\theta_{c,\text{mix}}\|/\|\theta_{c,\text{strict}}\|$ changes much less and does not track the decontamination curve. This supports the interpretation that tie training decontaminates the causal component by weakening causal–spurious coupling, rather than by shrinking causal weights directly.

**Near ties (robustness under imperfect ties).** We also test an imperfect-tie setting in which tie pairs differ slightly in causal features. Near ties have zero mean but nonzero variance in both blocks, with causal variance $\tau^2 I$ and spurious variance $\sigma^2 I$, where $\tau^2 \ll \sigma^2$:

$$\Delta\phi_c^{\text{near}} \sim \mathcal{N}(0, \tau^2 I), \qquad \Delta\phi_s^{\text{near}} \sim \mathcal{N}(0, \sigma^2 I),$$

with the two blocks sampled independently. Thus near ties still add far more curvature in spurious directions than in causal directions, but they no longer leave the causal block exactly unchanged. The mixed moments become

$$\Sigma_{cc}^{\text{mix}} = \alpha\Sigma_{cc} + (1-\alpha)\tau^2 I, \quad \Sigma_{cs}^{\text{mix}} = \alpha\Sigma_{cs}, \quad \Sigma_{ss}^{\text{mix}} = \alpha\lambda_0 I + (1-\alpha)\sigma^2 I, \quad \mu^{\text{mix}} = \alpha\mu,$$

and the linearized causal block can be written with two parameters,

$$\theta_{c,\text{mix}} = \frac{2}{\beta}\left(\Sigma_{cc} + \gamma(\alpha)I - \omega(\alpha)\,\Sigma_{cs}\Sigma_{sc}\right)^{-1}\left(\mu_c - \omega(\alpha)\,\Sigma_{cs}\mu_s\right), \quad \gamma(\alpha) = \frac{(1-\alpha)\tau^2}{\alpha}, \quad \omega(\alpha) = \frac{\alpha}{\alpha\lambda_0 + (1-\alpha)\sigma^2}.$$

Here $\omega(\alpha)$ controls the reduction of causal–spurious coupling, exactly as for exact ties, while $\gamma(\alpha)$ is the cost of imperfect ties: a small extra causal-curvature term arising from causal leakage. When $\tau^2 \ll \sigma^2$ the dominant effect is still spurious suppression and decontamination; as the leakage $\tau^2$ grows, causal preservation weakens. Near ties therefore serve as a robustness test of the exact-tie mechanism: they should suppress spurious reliance while still moving the causal block toward the pure-causal solution, provided the causal leakage is small relative to the spurious variation. Accordingly we report two quantities together, the spurious norm ratio $\|\theta_{s,\text{mix}}\|_2/\|\theta_{s,\text{strict}}\|_2$ and the causal decontamination ratio $\|\theta_{c,\text{mix}} - \theta_c^{\text{pure}}\|_2/\|\theta_{c,\text{strict}} - \theta_c^{\text{pure}}\|_2$.

**Results.** Figure 11 shows that the same mechanism persists under near ties. Although near ties introduce small causal leakage, increasing the tie contribution suppresses spurious reliance: the spurious norm ratio decreases sharply and closely

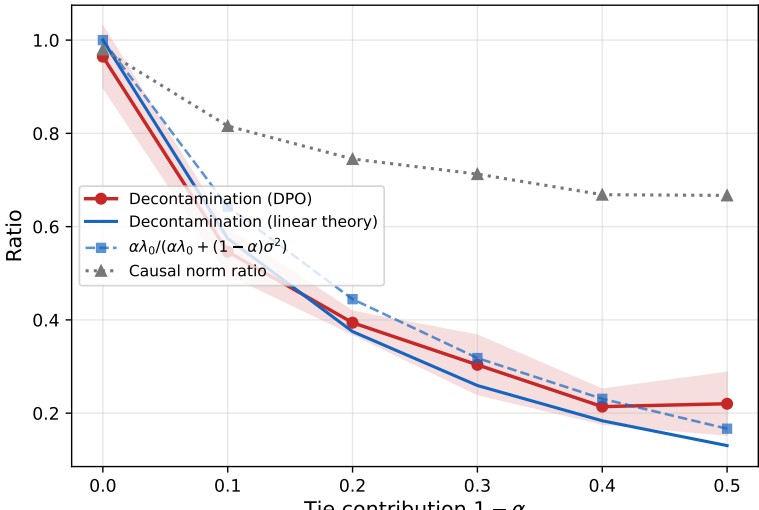

*Figure 10.* Causal decontamination ($d_c = d_s = 5$). As the tie contribution $1 - \alpha$ increases, the causal block moves toward the pure-causal solution (relative distance, red), closely following the linearized prediction (blue) and the isotropic factor $\alpha\lambda_0/(\alpha\lambda_0 + (1-\alpha)\sigma^2)$. The causal norm ratio (grey) changes much less and does not track this contraction, confirming that tie training decontaminates rather than merely shrinks the causal block.

follows the linearized prediction. At the same time, the causal decontamination ratio also decreases, showing that the causal block moves closer to the pure-causal solution. In contrast, the causal norm ratio changes much less and does not track either suppression curve. Thus, near ties do not merely shrink all parameters; they primarily reduce spurious reliance while still improving causal decontamination when the causal leakage is small relative to the spurious variation.

### F.1.7. TAKEAWAY (LINEAR)

The linear experiments provide confirmation of Theorems 4.1, 5.3, and 6.2. Strict-only training learns persistent spurious parameters and induces a vulnerability to irreducible deployment shift error under $Q$. Tie training acts as a selective regularizer on $\phi_s$ by injecting spurious variance. This reduces $\|\hat{\theta}_s\|$ and lowers shift-induced deployment error without relying on additional samples from $P$. As a result, tie training can remove the irreducible vulnerability as predicted by our mathematical analysis.

### F.2. Neural Networks (Nonlinear Regime)

**Motivation.** Real-world preference models are nonlinear and do not expose an explicit causal–spurious feature decomposition. As a result, the developed linear theory fails to apply exactly in this regime. Thus, our goal in this appendix is mechanism validation rather than exact prediction. In particular, we test whether the qualitative spurious-learning mechanisms identified in the linear analysis persist when representations are nonlinear and hidden. This allows us to assess whether the theory captures dominant dynamics rather than model-specific artifacts.

### F.2.1. DATASET CONSTRUCTION

**Latent variables.** We generate data from a latent quality variable $q \sim \mathcal{N}(0, 1)$ that determines true preference ordering. Causal features are constructed as nonlinear functions of $q$ with additive noise,

$$\phi_c = f_c(q, \varepsilon),$$

where $f_c$ can include transformations such as $q$, $q^2$, or $\sin q$ (we use $q$ in our experiments). These features contain information about $q$ but are not linearly related to it. As a result, recovering quality requires nonlinear processing.

Spurious features are generated as correlated but non-causal functions of $q$,

$$\phi_s = \rho\, f_s(q) + \sqrt{1 - \rho^2}\, \xi,$$

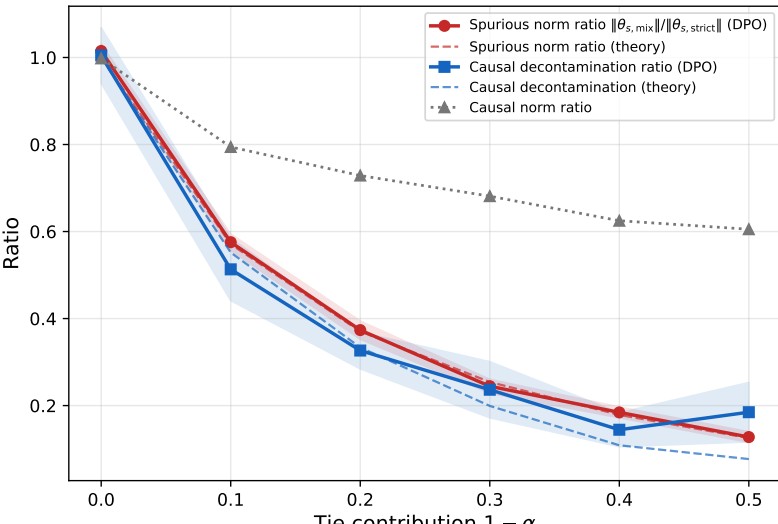

*Figure 11.* Near-tie robustness ($\tau^2 = 0.1 \ll \sigma^2 = 5$). Near ties introduce small causal leakage but much larger spurious variation. As the tie contribution $1 - \alpha$ increases, spurious reliance is suppressed and the causal block moves closer to the pure-causal solution. Both DPO curves closely follow the corresponding linearized predictions. The causal norm ratio changes much less, showing that the effect is not simple parameter shrinkage.

where $f_s(q)$ is correlated with $q$ and $\xi$ is independent noise. The parameter $\rho$ controls the strength of spurious correlation during training. These features are predictive in-distribution but have no causal relationship to preference labels. This construction mirrors spurious cues in real preference datasets.

**Nonlinear mixing.** The observed input to the model is a nonlinear mixture of causal and spurious features,

$$\phi = g([\phi_c; \phi_s]),$$

where $g(\cdot)$ is a fixed, nonlinear function unknown to the model (we use as a random MLP in the experiments): a generic, architecture-agnostic entangling of causal and spurious features that prevents the model from exploiting a hand-designed structure. The mechanisms we study do not depend on this specific choice. This mixing prevents direct access to $\phi_c$ or $\phi_s$. As a result, the model must learn representations internally. This setting tests whether spurious reliance emerges even when the decomposition is hidden.

**Tie construction.** We construct tie examples by enforcing $|q_1 - q_2| \leq \tau$ for a small threshold $\tau$. Preference labels for ties are assigned randomly, so causal signal is intentionally weak. We manipulate spurious features independently of $q$, either by assigning opposing extremes or by randomizing them. This creates pairs with minimal causal difference and strong spurious contrast. These ties provide targeted gradient signal against spurious reliance.

F.2.2. MODEL AND OBJECTIVE.

**Model.** We train a multilayer perceptron (MLP) reward model $r_\theta(\phi)$ that maps nonlinear inputs to scalar scores. The model has no architectural bias toward separating causal and spurious components. All structure must be learned from data. This setting reflects realistic nonlinear preference models. It therefore provides a stringent test of the theory.

**Training loss.** Training uses a pairwise logistic objective,

$$\log \sigma\big(\beta\big(r(x_w) - r(x_l)\big)\big),$$

which matches the standard preference optimization loss (we use $\beta = 1.0$ in the experiments). We vary the fraction of strict versus tie comparisons using the parameter $\alpha$. When $\alpha = 1$, training uses only strict comparisons. As $\alpha$ decreases, a larger fraction of tie data is introduced.

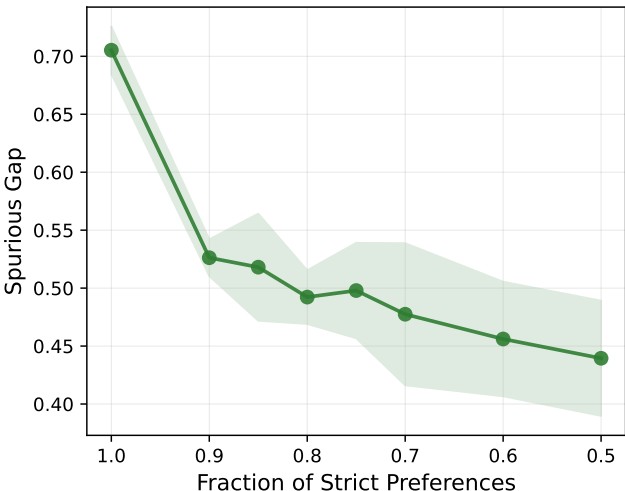

*Figure 12.* Spurious gap (accuracy difference between aligned and misaligned spurious conditions) as a function of the fraction of strict preferences $\alpha$. Note that as $\alpha$ decreases, the number of ties increases. Thus, tie training reduces spurious reliance despite hidden representations.

### F.2.3. PROXY METRICS

**Spurious gap.** We measure the spurious gap as the difference between accuracy on pairs where spurious features align with true quality and accuracy on pairs where they conflict. Let $\mathrm{ACC_{aligned}}$ and $\mathrm{ACC_{misaligned}}$ denote these accuracies. Their difference quantifies reliance on spurious cues. A large gap indicates strong spurious dependence.

**Adversarial accuracy.** We evaluate adversarial accuracy under a distribution where spurious correlations are reversed. Performance in this setting isolates failure due to spurious reliance. Because the deployment objective is nonlinear, we use accuracy as a proxy for utility. Persistent degradation under this shift indicates misgeneralization.

**Counterfactual margin.** We measure counterfactual sensitivity by flipping spurious features while holding causal features fixed. The counterfactual margin is defined as

$$\mathbb{E}[|r(c, s) - r(c, -s)|].$$

Large values indicate that the learned reward depends strongly on spurious features. Tie training is expected to reduce this margin.

### F.2.4. RESULTS

**Spurious learning.** Models trained without ties exhibit clear spurious learning. They show a large spurious gap, indicating substantially different performance when spurious cues align or conflict with quality. Figure 12 reports this gap across training conditions. These models also perform poorly under adversarial evaluation, showing that spurious reliance translates into deployment failures.

**Tie training.** Introducing tie training reduces sensitivity to spurious features. Figure 13 shows that the counterfactual margin decreases sharply as the fraction of tie data increases, indicating reduced dependence on spurious cues. Figure 14 illustrates that tie training also improves adversarial accuracy under adversarial reversed correlations. These trends qualitatively match the predictions of the linear analysis.

**Takeaway (Nonlinear).** In the nonlinear regime, the exact linear theory no longer applies. Nonetheless, the same qualitative spurious-learning mechanisms persist. Tie training reduces spurious reliance and improves robustness under distribution shift. These results suggest that the linear analysis captures dominant dynamics of preference learning. The theory therefore provides useful guidance beyond the linear setting.

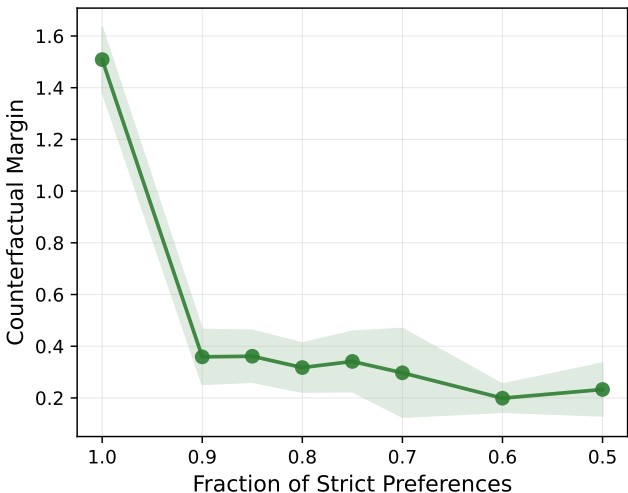

*Figure 13.* **Tie training reduces spurious reliance.** Counterfactual margin $\mathbb{E}[|r(\phi) - r(\phi_{\mathrm{cf}})|]$ as a function of the fraction of strict preferences $\alpha$. As $\alpha$ decreases (more tie comparisons), the counterfactual margin drops sharply, indicating reduced sensitivity of the learned model to spurious features.

**Hyperparameters.** Table 5 reports key hyperparameters for the nonlinear experiments.

### F.3. Large Language Models (Synthetic Hotel Benchmark)

We study a large-scale synthetic language-model setting, where preferences are expressed in natural language and spurious attributes resemble real-world surface cues.

**Dataset: synthetic hotel preferences.** We construct a synthetic hotel comparison dataset designed to study spurious correlation learning and the effect of tie training in large language models. Each example consists of a user context, two hotel options $(A, B)$, and a binary preference label indicating which hotel is preferred. The dataset explicitly separates causal utility, which determines true quality, from spurious features, which are surface attributes correlated with utility during training. This separation allows controlled experiments in which correlations can be manipulated without changing the underlying task. As a result, robustness under distribution shift can be evaluated in isolation.

Each hotel is assigned a latent true utility $u(h)$ computed from task-relevant attributes, including `price`, `distance_to_destination`, `star_rating`, and context-dependent `amenities`. Preference labels are generated by a teacher that depends only on this true utility, so higher $u$ always corresponds to higher quality. The true utility is never directly observed by the model and must be inferred from preference supervision. This ensures that causal signal is present only implicitly. Consequently, any reliance on non-causal attributes reflects spurious learning.

We designate the following hotel attributes as spurious features:

- `street_number` (100–9999)

- `floor_number` (1–20)

- `building_age` (1–50; lower is better)

- `renovation_year` (2000–2024)

- `hotel_chain_tier` ∈ {Budget, Standard, Premium}

- `lobby_size_sqft` (500–5000)

- `employee_count` (10–200)

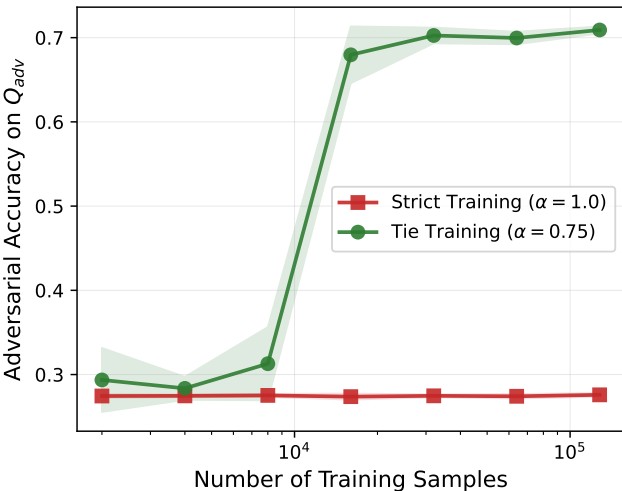

*Figure 14.* **Strict-only training plateaus under distribution shift; tie training improves robustness.** Adversarial accuracy on $Q_{\mathrm{adv}}$, where spurious correlations flip, as a function of the number of training samples. Strict-only training ($\alpha = 1.0$) exhibits a persistent accuracy plateau despite increasing data. In contrast, tie training ($\alpha = 0.75$) improves adversarial accuracy, breaking the plateau.

These attributes do not affect true utility but are correlated with utility during training. Their values are explicitly manipulated to create different correlation regimes at deployment. This design ensures that spurious cues are strong, structured, and controllable. As a result, failures under shift can be directly attributed to spurious reliance.

Spurious features are assigned as deterministic functions of a normalized utility level $u_{\mathrm{norm}} \in [0, 1]$. In the *normal* correlation mode, higher-utility hotels receive systematically better spurious attributes, such as newer buildings, premium chains, larger lobbies, and more employees. In *suppression* mode, spurious attributes are decorrelated from utility. In *adversarial* mode, the mapping is inverted using $1 - u_{\mathrm{norm}}$, so high-utility hotels receive worse spurious attributes. This construction induces sharp distribution shifts without altering the causal preference structure.

For standard (non-tie) training examples, we sample two hotels $(A, B)$, compute their true utilities $(u_A, u_B)$, assign spurious features according to the chosen correlation mode, and label the pair by the utility ordering. This procedure induces strong correlations between spurious attributes, preference labels, and true quality under the training distribution. These correlations are systematic rather than noisy. As a result, standard preference optimization objectives are incentivized to rely on spurious cues. An example of a hotel preference sample is:

```
You are helping someone choose the right hotel for their stay. ...

--- Option A ---
Hilton Plaza is prominently located at 4126 Second Ave. ...

--- Option B ---
Hampton Inn Central is prominently located at 6560 Park Blvd. ...

--- Task ---
Which of these two options is the better choice for the user?
```

The full dataset and generation code are available at https://github.com/cmoyacal/tie-training.

**Tie construction in the LLM setting.**  A tie is defined as a hotel pair $(A, B)$ such that the utility difference satisfies $|u_A - u_B| < \tau$ for a small threshold $\tau$. In these pairs, the causal signal is intentionally weak by construction. Preference labels are assigned randomly, with $y \sim \mathrm{Bernoulli}(1/2)$. This ensures that labels are independent of both utility and spurious features. Ties therefore isolate non-causal learning signals.

**Informative ties.**  We make tie examples informative by explicitly decorrelating spurious features from utility. Our default informative strategy assigns one hotel maximal spurious features and the other minimal spurious features. The assignment is

*Table 5.* Key hyperparameters for the nonlinear synthetic experiments (spurious gap vs. $\alpha$, counterfactual margin vs. $\alpha$, and adversarial accuracy vs. sample size $N$). Remaining settings (initializations, loss reduction, dataloader shuffling) follow the defaults in our released code.

| Hyperparameter | Value |
|---|---|
| *Data generating process* | |
| Causal latent dim $d_c$ | 10 |
| Spurious vector dim $d_s$ | 10 |
| Shortcut scalar dim | 1 |
| Spurious-vector correlation $\rho$ | 0.9 |
| Shortcut–$q$ correlation $\alpha_{\text{spur}}$ | 6.0 |
| Noise scales $(\sigma_c, \sigma_s, \sigma_t)$ | (1.0, 1.0, 0.15) |
| Bradley–Terry teacher $\beta_{\text{teacher}}$ | 1.0 |
| Mixer $g$ | Frozen MLP, Linear–Tanh–Linear–Tanh |
| Mixer hidden width | 64 |
| *Scorer model & training* | |
| Scorer architecture | 3-layer MLP, ReLU |
| Hidden width | 128 |
| Loss | Pairwise BCE-with-logits |
| Model temperature $\beta_{\text{model}}$ | 1.0 |
| Optimizer | AdamW |
| Learning rate | $2 \times 10^{-3}$ |
| Weight decay | $1 \times 10^{-4}$ |
| Epochs | 8 |
| Batch size | 1024 |
| *Tie construction & data* | |
| Tie construction | Spurious flip: $(c, s, t)$ vs. $(c, -s, -t)$, label $\sim \text{Bern}(0.5)$ |
| Training pairs (gap / CF-margin sweeps) | 50,000 |
| Training pairs ($N$-sweep) | {2,000, 4,000, 8,000, 16,000, 32,000, 64,000, 128,000} |
| Eval pairs (spurious gap, adv. acc.) | 30,000 |
| Eval pairs (CF margin) | 20,000 |
| Mixing ratio $\alpha$ ($\alpha$-sweeps) | {1.0, 0.9, 0.85, 0.8, 0.75, 0.7, 0.6, 0.5} |
| Mixing ratio $\alpha$ ($N$-sweep) | 1.0 (strict baseline), 0.75 (tie training) |
| Test distributions | $P$ ($\rho_{\text{sign}} = +1$), $Q_{\text{adv}}$ ($\rho_{\text{sign}} = -1$) |
| Seeds | 5 |

random between $A$ and $B$, so spurious direction is uninformative. This produces pairs with near-zero causal margin but maximal spurious contrast. As a result, gradients from these examples penalize spurious reliance.

*Informative ties* satisfy three properties simultaneously. First, the causal signal is weak because $|u_A - u_B|$ is small. Second, the spurious contrast is large because spurious features are maximally separated. Third, labels are independent of spurious attributes by construction. Together, these properties ensure that tie gradients push against spurious features while preserving causal learning.

We also evaluate an alternative tie construction strategy. In a standard-monotonic strategy, we assign spurious features monotonically from utility. These *non-informative ties* provide weak regularization signal.

**Model and training.** We fine-tune a fixed base language model `Llama-3.2-1B-Instruct` using Direct Preference Optimization (DPO) (Rafailov et al., 2023) and Low-Rank Adaptation (LoRA) (Hu et al., 2022) for one epoch. All experiments use the same architecture and optimization settings. We compare strict training, which uses only standard preference pairs, with tie training, which augments the dataset with ties.

*Table 6.* Tie Training (TT) DPO with informative ties improves robustness to spurious distribution shift. While standard DPO performs well in-distribution ($P$), its accuracy degrades under suppressed and adversarial spurious correlations ($Q$). Informative tie training preserves high accuracy under both shifts without sacrificing in-distribution performance. The mixture variants (TT-Mixture) sample informative ties with probability $p$ and otherwise emit non-informative ties, interpolating between the two regimes. We report $p = 0.10$ and $p = 0.25$. Non-informative ties inject spurious contrast in the same direction as the causal signal. The modest robustness they do provide stems from the random tie label and near-tie causal perturbation, which act as weak regularizers. Results are reported for overall accuracy and per-option accuracy (Hotel A/B), illustrating that robustness gains are systematic rather than label-specific. In all experiments, we use $1 - \alpha = 0.3$ augmented ties.

| | Accuracy (overall) | Accuracy (Hotel A) | Accuracy (Hotel B) |
|---|---|---|---|
| DPO-Strict-In-Distr.($P$) | 92.25% | 89.54% | 94.88% |
| DPO-Strict-Suppressed($Q$) | 74.00% | 74.48% | 77.81% |
| DPO-Strict-Adversarial($Q$) | 64.20% | 55.72% | 72.76% |
| DPO-TT-Informative-In-Distr.($P$) | 92.40% | 88.02% | 96.65% |
| DPO-TT-Informative-Suppressed($Q$) | 82.85% | 83.17% | 82.50% |
| DPO-TT-Informative-Adversarial($Q$) | **86.70%** | 80.60% | **92.86%** |
| DPO-TT-Mixture ($p$=0.25)-In-Distr.($P$) | 89.40% | 92.76% | 86.34% |
| DPO-TT-Mixture ($p$=0.25)-Suppressed($Q$) | 83.40% | 90.81% | 76.30% |
| DPO-TT-Mixture ($p$=0.25)-Adversarial($Q$) | 83.20% | **85.84%** | 80.47% |
| DPO-TT-Mixture ($p$=0.10)-In-Distr.($P$) | 90.75% | 91.92% | 89.68% |
| DPO-TT-Mixture ($p$=0.10)-Suppressed($Q$) | 84.10% | 86.72% | 81.59% |
| DPO-TT-Mixture ($p$=0.10)-Adversarial($Q$) | 81.70% | 79.84% | 83.62% |
| DPO-TT-Non-Informative-In-Distr.($P$) | 91.20% | 92.13% | 90.35% |
| DPO-TT-Non-Informative-Suppressed($Q$) | 81.85% | 83.04% | 80.71% |
| DPO-TT-Non-Informative-Adversarial($Q$) | 76.90% | 72.47% | 81.49% |

**Evaluation metrics.** We measure *in-distribution accuracy* on standard test pairs drawn from the training distribution $P$. This evaluates whether models trained with ties retain performance on the original task. High in-distribution accuracy indicates that causal learning is preserved. We report overall accuracy and per-option accuracy. This allows us to detect asymmetric degradation.

We evaluate robustness under deployment distributions $Q$ where spurious correlations are suppressed or adversarially reversed. These settings isolate failures caused by spurious reliance. Accuracy is measured using the same preference labels derived from true utility. Performance degradation under $Q$ reflects reliance on spurious features. Robust models should maintain accuracy across shifts.

**Results.** Table 6 shows that Tie Training (TT) DPO with informative ties improves robustness to spurious distribution shift. While standard DPO performs well in-distribution ($P$), its accuracy degrades under suppressed and adversarial spurious correlations ($Q$). Informative tie training preserves high accuracy under both shifts without sacrificing in-distribution performance. The mixture variants (TT-Mixture) sample informative ties with probability $p$ and otherwise emit non-informative ties, interpolating between the two regimes. We report $p = 0.10$ and $p = 0.25$. Non-informative ties inject spurious contrast in the same direction as the causal signal. The modest robustness they do provide stems from the random tie label and near-tie causal perturbation, which act as weak regularizers.

**Ablation of $\alpha$.** We fine-tune Llama-3.2-1B-Instruct with DPO (LoRA, $r = 16$, $\beta$=0.1) on 5,000 strict preference pairs and sweep the tie augmentation ratio $1 - \alpha = n_{\text{ties}}/(n_{\text{strict}}+n_{\text{ties}})$, the fraction of tie pairs in the augmented training set. Ties are near-tie pairs in which the spurious feature is decorrelated from utility; $\alpha$=1.0 is the strict-only baseline (no ties) and lower $\alpha$ injects more ties. All models are evaluated on a fixed adversarial test set ($Q_{\text{adv}}$, 2,000 pairs, spurious correlation strength 0.99) held constant across seeds. We report mean $\pm$ standard deviation over 5 seeds.

**Results.** Adversarial accuracy rises monotonically as ties are added, from 64.1% at $\alpha$=1.0 to 86.4% at $\alpha$=0.5, a 22.3-point gain over the strict-only baseline (Figure 15). The largest improvement comes from the first increment of ties ($\alpha$=1.0→0.9, +11.4 points), with diminishing returns thereafter. Variance also shrinks as $\alpha$ decreases (std 0.029→0.009), indicating that

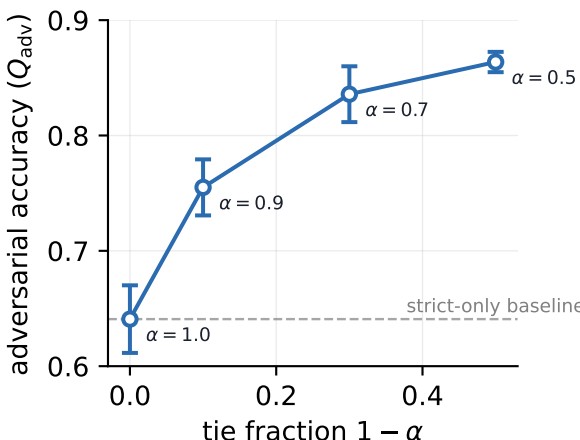

*Figure 15.* Tie augmentation improves adversarial robustness. Adversarial accuracy ($Q_{\mathrm{adv}}$, anti-correlated spurious feature) of Llama-3.2-1B-Instruct fine-tuned with DPO, as a function of the tie fraction $1 - \alpha$, where $\alpha = n_{\mathrm{strict}}/(n_{\mathrm{strict}} + n_{\mathrm{ties}})$ is the share of strict pairs in the training set. The leftmost point ($1 - \alpha = 0$, i.e. $\alpha = 1.0$) is the strict-only baseline (dashed line). Adding decorrelated ties raises accuracy from $64.1\%$ to $86.4\%$, with the gain concentrated in the first increment and variance shrinking as more ties are added. Markers show the mean over 5 seeds; error bars show $\pm 1$ standard deviation.

*Table 7.* Key hyperparameters for the LLM Hotels experiments. Remaining settings (LoRA dropout, target modules, optimizer, sequence lengths, LR schedule) follow the defaults in our released code.

| Hyperparameter | Value |
|---|---|
| *DPO training* | |
| Base model | Llama-3.2-1B-Instruct |
| Fine-tuning | LoRA (4-bit), $r = 16$, $\alpha_{\mathrm{LoRA}} = 32$ |
| DPO $\beta$ | $0.1$ |
| Learning rate | $1 \times 10^{-5}$ |
| Epochs | $1$ |
| Effective batch size | $8$ |
| *Data* | |
| Training pairs (strict) | $5{,}000$ |
| Test pairs per distribution | $2{,}000$ |
| Test distributions | $P, Q_{\mathrm{sup}}, Q_{\mathrm{adv}}$ |
| Spurious correlation strength | $0.99$ |
| Tie construction | near-tie |
| Mixing ratio $\alpha$ | $1.0$ (baseline), $0.7$ (tie training) |
| Seeds | $5$ |

tie augmentation yields not only higher but also more consistent robustness across seeds.

**Takeaway (LLMs).** Spurious correlation learning persists in large language models trained with preference optimization. Increasing data alone does not eliminate spurious reliance or improve robustness under distribution shift. Informative tie training provides targeted robustness gains by directly penalizing spurious features. These effects mirror those observed in linear models and neural networks. Together, the results suggest that the linearized theory captures core dynamics of preference learning under spurious correlations.

**Hyperparameters.** Table 7 reports key hyperparameters for the LLM hotel experiments.

# G. RLHF-Style Reward Learning with Greedy Decoding

## G.1. Reward Learning Setup

We study linear reward learning under RLHF and show that it is a special case of Direct Preference Optimization (DPO). Specifically, RLHF reward learning corresponds to DPO with scaling parameter $\beta = 1$ and reference parameter $\theta_{\mathrm{ref}} = \mathbf{0}$, so that the effective parameter satisfies $\tilde{\theta} = \theta$. Under this specialization, the pairwise reward learning loss reduces to

$$\ell_{\mathrm{RLHF}}(\theta) = -\log \sigma(\theta^\top \Delta \phi),$$

which matches the DPO objective. This equivalence implies that the reward learning dynamics are unchanged relative to DPO in our linear analysis. Thus, the same spurious-learning mechanisms and distribution-shift vulnerabilities apply to standard RLHF reward learning.

## G.2. Deployment with Greedy Policies

Given a learned linear reward $r_\theta(x, y) = \theta^\top \phi(x, y)$, deployment uses greedy decoding to select outputs. This induces the greedy policy

$$\pi(x) = \arg\max_y \theta^\top \phi(x, y),$$

which is optimal under $r_\theta$ but can be suboptimal under the true deployment utility. To measure this gap under a shifted deployment distribution $Q$, we evaluate the true deployment value functional $V^\star$ rather than the learned reward. Let $\pi^\star$ denote the optimal policy for the deployment problem under $Q$ and $V^\star$, and define the deployment suboptimality as

$$\mathrm{SubOpt}_Q(\pi) := V^\star(\pi^\star) - V^\star(\pi).$$

This definition turns reward mislearning into a policy-level metric that we can track under distribution shift.

## G.3. Spurious Learning under RLHF Reward Learining

We now show that spurious correlation learning persists under RLHF reward learning and directly impacts greedy deployment.

**Setup.** We evaluate the RLHF reward learning dynamics using a synthetic environment where features $\phi(x) \in \mathbb{R}^d$ are decomposed into causal features $\phi_c \in \mathbb{R}^3$ and a spurious feature $\phi_s \in \mathbb{R}^1$. The full feature vector is given by the concatenation $\phi(x) = [\phi_c(x), \phi_s(x)]$.

We construct a set of five items $\mathcal{A} = \{a_1, \ldots, a_5\}$ with a fixed spurious correlation coefficient $\sigma = 1.0$. The feature representations are defined as follows:

$$\Phi = \begin{bmatrix} 1 & 1 & 0 & \sigma \\ 1 & 0 & 0 & 0 \\ 0 & 0 & 0 & 0 \\ 0 & 1 & 0 & \sigma \\ 0 & 0 & 1 & 0 \end{bmatrix}$$

Ground truth preference data is generated via a Bradley-Terry model $P(i \succ j) = \sigma(\theta^{*\top}(\phi(x_i) - \phi(x_j)))$. The ground truth parameter is set to $\theta^\star = [-1.0, 0.1, 0.05, 0.0]^\top$.

*Sampling Strategy.* We generate a dataset of $N$ pairwise comparisons with a fixed distribution of outcome types: strict preferences (75%). The samples are drawn evenly from three specific comparison pairs to create the experimental structure: $a_1$ vs $a_2$ (difference: $[0, 1, 0, \sigma]$), $a_2$ vs $a_3$ (difference: $[1, 0, 0, 0]$), and $a_5$ vs $a_3$ (difference: $[0, 0, 1, 0]$). Ties (25%). The remaining samples are labeled as ties.

**Results.** Figure 16 shows that training with $\ell_{\mathrm{RLHF}}$ yields a reward parameter $\hat{\theta}$ whose spurious component is nonzero, i.e., $\hat{\theta}_s \neq 0$. Similarly, Figure 17 illustrates that increasing the number of samples from $P$ reduces estimation error around this optimum but does not remove the spurious component.

## G.4. Effect of Tie Training

We next evaluate how spurious learning induces deployment suboptimality and how tie training mitigates this effect.

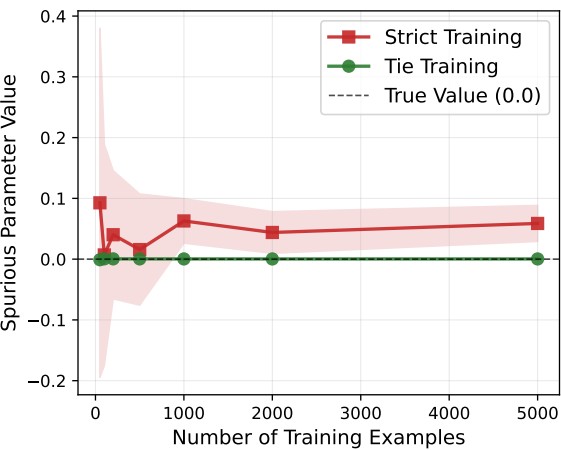

*Figure 16.* Under standard RLHF learning training, the learned policy exhibits nonzero reliance on spurious features ($\theta_{\mathrm{s}} \neq 0$), and this reliance does not vanish with additional data drawn from the training distribution $P$. Tie training explicitly counteracts this effect, driving spurious reliance toward zero.

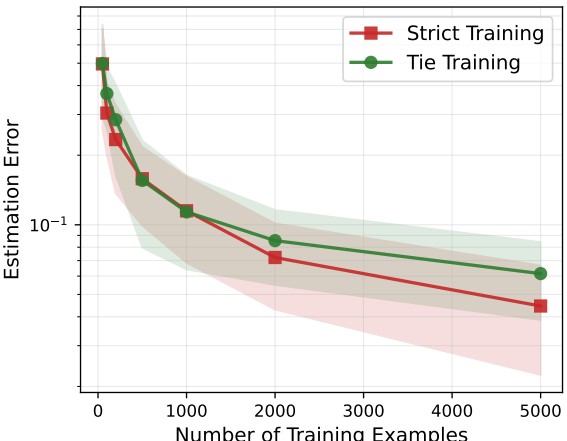

*Figure 17.* As the number of training samples increases, estimation error, defined as the weighted norm $\|\hat{\theta} - \theta^{\star}\|_{\Sigma}$ decreases at comparable rates for strict MLE training and tie training, showing that tie training reduces spurious reliance without sacrificing estimation accuracy.

**Results.** Without tie training, Figure 18 shows that the greedy policy $\hat{\pi}(x) = \arg\max_y \hat{\theta}^{\top}\phi(x,y)$ achieves low error on $P$ but incurs nonzero $\mathrm{SubOpt}_Q(\hat{\pi})$ when the deployment distribution $Q$ suppresses or reverses spurious correlations. As the number of training samples increases, the shift-induced component of the error persists. This persistence is strongest in adversarial reversals. Tie training reduces $\mathrm{SubOpt}_Q$ by shrinking $\hat{\theta}_{\mathrm{s}}$ while preserving the causal component, so the greedy policy becomes less sensitive to spurious shifts.

### G.5. Discussion and Conclusion

Our results show that greedy decoding does not affect what reward learning fits, but it could amplifies the behavioral impact of spurious reward errors. Under distribution shift, small spurious weights can flip greedy decisions and induce nonzero deployment suboptimality. Without tie training, this error persists in the infinite-data limit. Additional samples reduce estimation error but do not remove spurious reliance. In contrast, tie training shrinks spurious weights and reduces the resulting deployment error.

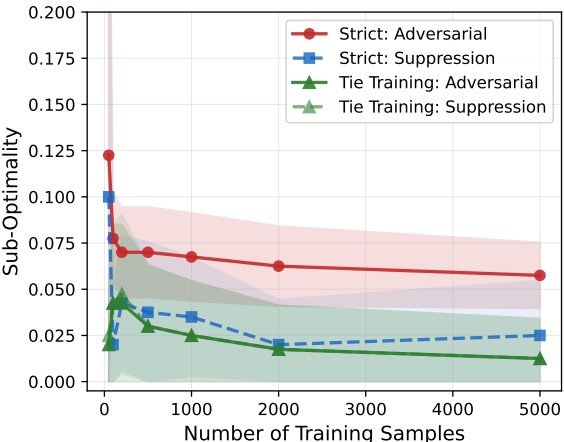

*Figure 18.* Greedy decoding of a log-linear RLHF policy does not introduce additional error mechanisms, but exposes spurious reward learning under shift: performance, measured as $\mathrm{SubOpt}_Q(\pi) := V^\star(\pi^\star) - V^\star(\pi)$ degrades in both adversarial and suppression settings, and this error does not vanish with more data from $P$. Tie training reduces this shift-induced error.

## H. Limitations and Future Work

**Local regime and linearization.** Our theoretical analysis relies on a local regime where the learned policy $\pi_\theta$ remains close to the reference policy $\pi_{\mathrm{ref}}$, enabling linearization of the KL-regularized objective.

**Tie construction and approximate equality.** The analysis assumes access to informative ties, preference pairs $(x, y_A, y_B)$, where responses have near-equal utility but differing spurious features. In practice, exact utility equality is difficult to verify, since true utility is latent and must be estimated from noisy human feedback or imperfect reward models.

However, the mechanism requires only that causal utility differences are small relative to spurious feature variability, not exact equality. Formally, what matters is that $|\phi_c(y_A) - \phi_c(y_B)|$ is small compared to $|\phi_s(y_A) - \phi_s(y_B)|$ in expectation over the tie distribution. Our experiments (Appendix F) demonstrate that ties constructed by selecting pairs with similar scores effectively reduce spurious learning, even when utilities are not exactly equal.

Nevertheless, formal guarantees for imperfect tie construction remain an open problem. How much utility mismatch can be tolerated before tie training becomes ineffective? How should one trade off the number of ties versus their quality? Can adaptive tie construction algorithms identify informative ties online during training? We will study these questions in our future work.

**Analytical assumptions and feature decomposition.** We assume features decompose as $\phi(y) = [\phi_c(y)^\top, \phi_s(y)^\top]^\top$ into causal and spurious components. This decomposition makes the spurious learning mechanism explicit and enables clean theoretical statements, but it is an idealization.

Importantly, this assumption is used only for analysis, not for the method. The decomposition serves as an analytical tool to understand why tie training works, not as a prerequisite for its application. Extending the theory to settings without a clear causal–spurious decomposition remains open.

**Scale and scope of validation.** Our experiments validate the theoretical mechanisms we derive: we demonstrate that predictions from the population-level theory match finite-sample behavior, that tie training reduces spurious parameters as predicted, and that these reductions translate to improved deployment robustness. However, these experiments are designed to test theoretical predictions in controlled settings, not to optimize end-to-end performance of production alignment systems.

Large-scale validation in production alignment pipelines remains important future work. This includes: applying tie training to frontier language models with billions of parameters, developing practical methods for tie construction from human feedback at scale, evaluating robustness improvements on diverse downstream tasks and distribution shifts, and comparing tie training to other robustness interventions such as distributionally robust optimization or causal regularization. Such validation would determine whether the gains observed in controlled experiments translate to meaningful improvements in deployed systems.

