# OpenReview forum: "Spurious Correlation Learning in Preference Optimization: Mechanisms, Consequences, and Mitigation via Tie Training"
_ICML.cc/2026/Conference — ICML 2026 regular_

### Official Review · Reviewer_ozVC · 2026-03-10

**Soundness:** 3
**Presentation:** 4
**Significance:** 3
**Originality:** 3
**Overall Recommendation:** 5
**Confidence:** 3

**Summary:**

This paper formalizes spurious correlation learning in DPO-style preference optimization and proposes a data-augmentation-based mitigation. The authors start by formalizing the problem of spurious learning and the concept of distributional shift. They then prove that spurious learning will happen under mild conditions (though assuming a log-linear learner) and lead to irreducible error. They theoretically predict the size of that error. Finally, they propose tie-training as a mitigation and analyze it both theoretically and empirically. They introduce synthetic scenarios of varying realism for evaluation: linear (matching theoretical assumptions), neural network, and large language model. They show promise of the proposed tie training technique, although limited by the practical difficulty of generating tie samples in realistic settings.

**Compliance With Llm Reviewing Policy:**

Affirmed.

**Final Justification:**

The contribution is theoretically strong and, after clarifications in the rebuttal, also well-supported empirically. I was torn between a 4 and a 5 (given that the clarifications are difficult to review without seeing the fully revised manuscript), but decided on a 5 because the authors have addressed my points in the rebuttal and the released code greatly reduces any clarity concerns.

**Key Questions For Authors:**

Most of the weaknesses above can be interpreted as questions. Of those, W1 and W2 have by far the highest impact on my score.

- **Q1:** Can the authors provide additional details on the empirical evaluation as requested in W1? The gold standard would be that the paper has sufficient detail for an independent reproduction of the empirical results *and* ideally the source code of the experiments are provided.
- **Q2:** How does tie training relate to RRM (W2) and iterated learning techniques (W3)?
- **Q3:** Are "spurious" features causal? If they are not, should the experimental setup be modified? (W4)

**Limitations:**

The biggest limitation is the practical difficulty of generating tie samples. I believe this limitation is reasonable for primarily theoretical work and is adequately discussed, however.

**Strengths And Weaknesses:**

### Strengths

- The paper is very well-written and polished.
- The formalization is good.
- The synthetic experiments are clever and test the theory well.
    - I particularly like the idea of generating latent features first and then mixing them nonlinearly.

---

### Weaknesses

The following weaknesses are ordered by severity:

- **W1:** The empirical setup lacks clarity. This is a big factor holding back soundness. A non-exhaustive list of clarity concerns follows:
    1. How many seeds do you use in the experiments? What exactly do shaded areas indicate in the plots?
    2. Which transformations exactly make up $f_c$? How were they chosen?
    3. What exactly is $g$ and how was it chosen (both variants; the letter $g$ is used for two distinct concepts in Appendix F.2.1)?.
    4. What are your hyperparameters (e.g., learning rate, tie threshold)? How did you choose them? What are your network architectures?
    5. How exactly do you generate tied pairs? How do you enforce the utility difference does not exceed the threshold? Where do the candidate pairs come from?
    6. What are the "second-order corrections" you describe in Figure 1?
    7. Where does the base hotel data come from? How does a textual hotel description look like?
    8. In F.1.2 you claim the model class is log-linear, but then define $r_\theta$ to be linear. Am I misunderstanding something?
- **W2:** The authors do not discuss RRM [Liu et al., 2025], which seems very closely related to the spurious correlation problem and the proposed tie training approach.
- **W3:** The authors do not discuss how spurious correlation relates to iterative variants of preference optimization [Xiong et al., 2024], where the fine-tuned policy is used to generate new pairwise comparisons, which are then used to further fine-tune the policy. I think such an iterative setup would at least partially mitigate spurious correlation learning and should be discussed.
- **W4:** The definition of spurious features is not quite clear to me. Are these features that happen to correlate with outcomes but are not causal of them, or are they features that *do* actually cause the outcomes but that, for some reason, we do not want to learn? Intuitively it should be the former, but in practice I think it is often the latter (e.g., length actually causes an annotator to prefer a response).
    - The empirical evaluation strengthens this confusion: In F.1, if I understand it correctly, you describe that the "spurious" features *are* used by the ground truth preference model; so they do influence the decisions. At the same time the spurious features you sample are completely independent of the causal features. I would have expected that the ground-truth is only based on the causal features and spurious features are correlated with them (but do not directly influence the choice).
    - This is closely related to Assumption 3.1 (which I believe does not hold in practice for examples like length or style bias). This may also be related to your distinction between spurious bias and spurious correlation.
    - If your data is generated according to BT ground truth, why would it ever have mean spurious bias?
- **W5:** The key results as I understand them (models pick up on any features that are correlated with the outputs on the training distribution, even if the correlation is not what we want to capture; additional training data that provides "evidence" against this correlation prevents this) are not very surprising to me. I think it is still very valuable to formalize and theoretically examine this, however, and the empirical study further strengthens the work. This makes the difference between significance 3 or 4 for me, but otherwise does not greatly influence my score.
- **W6:** Misc questions and minor concerns
    1. Your models are trained on the feature difference. The standard setup is to take the difference in predictions, not train on the feature difference. Does this make a practical difference for your evaluation?
    2. I was surprised that DPO on hotel data (sec 7.3) performs better under the Adversarial condition compared to the suppressed one. Do you have a hypothesis as to why?
    3. Could you test to which degree Assumption 3.2 holds in your experiments?
- **W7:** "Why not alternatives" (I do not expect that you ablate every possible design choice; but I would be interested in your reasoning).
    1. Why do you assign tie labels randomly as opposed to assigning equal target probability to both options?
    2. Could you use random neural networks as your random nonlinear functions instead of the custom combinations of nonlinear operations? Is there a reason to prefer the nonlinear operations you have right now?
    3. Instead of sampling two items for the tied pairs and then ensuring their utility is close, could you modify only the spurious features of the same item (ensuring utility difference 0)?
    4. Why do the tie strategies differ across the empirical setups (e.g., why adversarial for neural networks vs random_extreme for the LLM setup? Could you not apply the same strategies to all?)
- **Minor** points that do not influence my score (no response necessary, but I recommend clarifying in the paper).
    1. Appendix G is not referenced in the paper.
    2. When initially reading the "Bradley-Terry preference model" paragraph it was not immediately clear to me that you describe idealized assumptions here (not describing a learning model). This could be clarified.
    3. The way DPO is framed in the paper it appears like DPO also directly trains a reward model; it should be made explicit in which way $\theta$ can be viewed as the parameters of a reward model vs the parameters of a policy.
    4. Figure 2 on the right should explicitly list the $\alpha$ that was used (0.75?)
    5. Your evaluation measures the DPO objective on deployment data, which is only indirectly meaningful as in practice you would not run DPO on deployment (you'd deploy the trained policy instead). This should be briefly discussed.


[Liu et al., 2025] https://openreview.net/forum?id=88AS5MQnmC
[Xiong et al., 2024] https://proceedings.mlr.press/v235/xiong24a.html

---

> ### Author Rebuttal · Authors · 2026-03-31
>
> ## Reviewer ozVC
> Thank you for your detailed review.
>
> ### W1 & Q1. Experimental details
> **Response.** We agree and will improve the appendix. Code is available at [[LINK]](https://github.com/anonymous-ml-submission-1304/tie-training). The revision will add hyperparameters and tie-generation algorithms. Brief clarifications: 10 seeds with $\pm 1$ std shading; $g$ is a frozen random neural network in our code (we will ablate alternatives); $g$ is overloaded (mixing func. vs. spurious feature map) and will be given distinct notation; hotel data is in the repository.
>
> *Tie generation.* Log-linear ties have zero causal component and Gaussian spurious differences with Bernoulli(0.5) labels. Nonlinear and LLM ties are selected by accepting candidate pairs with near-matched causal utility.
>
> *Second-order correction* is a mean-field curvature adjustment that accounts for BT annotator (used only in log-linear experiments, not required by the theory), which pushes data moments toward the tails where linearization is less accurate but directionally correct. Full details in the revision.
>
> *Log-linear policy vs. BT annotator.* The policy is log-linear throughout. The BT annotator is a data generation choice in some experiments, not required by the theory, which depends only on $(\mu, \Sigma)$. Experiments assume uniform reference (making the DPO implicit reward close to a linear score). The full DPO loss with reference model is used for the LLM experiment.
>
> ### W2 & Q2. Connection to RRM (Liu et al., 2025).
> **Response.** Thank you for pointing out RRM, which addresses spurious correlations in reward learning via causal framework and feature-agnostic data augmentation. We will add a discussion.
>
> Both works share the insight that tie-like supervision reduces spurious reliance. RRM provides a general causal framework with a practical, feature-agnostic construction: clear strengths. Our work differs in two aspects: we provide mechanism level, quantitative structure characterizing when spurious learning is inevitable and how tie training suppresses it. We operate using symmetric hard-labeled pairs in the unmodified DPO loss, whereas RRM targets reward model training with soft labels for neutrals. We view RRM’s augmentation as introducing preference-neutral signal at the data level. Our framework can offer complementary lens to explain when such signals are most effective, an interesting direction for future work.
>
> ### W3 & Q2. Relation to iterative preference optimization
> **Response.** Thank you for Xiong et al. (2024), which addresses distribution mismatch by collecting on-policy data across rounds. We will add a discussion.
>
> Our analysis is complementary. Theorem 4.1 shows that for any given data distribution, if $\mu_s \neq 0$ or $\Sigma_{cs} \neq 0$, the learned solution contains spurious components. This applies to each iteration of an iterative pipeline. Iterative approaches can modify spurious correlations across rounds, depending on the data collected. The two approaches are composable: iterative methods update the distribution across rounds, while tie training targets the covariance structure within each round.
>
> ### W4 & Q4. Role of spurious features
> **Response.** "Spurious" denotes features the designer does not want the policy to rely on at deployment, a design choice, not an ontological claim. Features like length may genuinely influence annotators; we call them spurious because their correlation with preferences may shift at deployment.
>
> The reviewer correctly identifies a source of confusion: the BT model is not required by the theory, which depends only on feature statistics $(\mu, \Sigma)$. BT was introduced to simulate annotator behavior in experiments. We will make this separation explicit. $\mu_s \neq 0$ is a property of the dataset, the feature-difference statistics the theory operates on. Assumption 3.1 is an idealization for the theory; even when spurious features partially cause annotator preferences, tie training suppresses reliance on them at deployment.
>
> ### W5. Novelty
> **Response.** We agree that the intuition is familiar. The contribution is the structure revealed by the formalization: two distinct mechanisms, an irreducible deployment error, and none of these follow from intuition alone. The structure is also constructive: tie training follows from the decomposition, targeting the mechanisms simultaneously.
>
> ### W6 & W7. Clarifications
> **Response.** Log-linear policy implies that DPO implicit reward reduces to $\beta \theta^\top \Delta\phi$, making feature differences equivalent to the standard loss. We use random hard labels because they require no change to the DPO loss. Remaining questions (adversarial vs. suppressed gap, Assumption 3.2, mixing ablation, tie unification) will be addressed in the revision.
>
> **Closing.** We hope these clarifications address your concerns. Due to space constraints, we welcome follow-up questions on any deferred points and would appreciate an updated assessment.

---

> > ### Author Rebuttal · Reviewer_ozVC · 2026-04-03
> >
> > Thank you for your thorough response, which addresses my concerns. I will raise my score to recommend acceptance.

---

> > > ### Author Response · Authors · 2026-04-06
> > >
> > > Thank you for the thorough feedback and for taking the time to review our rebuttal. We are glad the clarifications addressed your concerns. We will incorporate your suggestions, including improving the experimental details, clarifying assumptions and notation, and strengthening the discussion and positioning, in the final version.

---

### Official Review · Reviewer_5Z3G · 2026-03-11

**Soundness:** 3
**Presentation:** 3
**Significance:** 3
**Originality:** 3
**Overall Recommendation:** 5
**Confidence:** 3

**Summary:**

This paper addresses the issue of spurious correlations affecting models in preference optimization methods by providing a unified theoretical framework. Under a local linearization assumption of log-linear policies, the authors identify systematic deployment errors arising from mean spurious bias and causal-spurious correlation leakage. Based on this analysis, the paper introduces Tie Training, a method that suppresses learning of spurious features, and empirically validate the theoretical predictions.

**Compliance With Llm Reviewing Policy:**

Affirmed.

**Final Justification:**

The author's rebuttal addressed my main concerns, so I improved my score.

**Key Questions For Authors:**

1. Please clarify the definitions of “proxy objective” and “intended goal” introduced in the introduction—are these referring to the reward model’s learned signal versus the true human intent?
2. Assumption 3.2 requires operation near initialization, bounded feature magnitudes, or a sufficiently small KL-regularization coefficient β. Are these conditions reasonably satisfied in typical preference alignment setups?
3. Theorem 4.1 appears conceptually related to prior work on out-of-distribution (OOD) generalization. Could the authors compare their theoretical findings with existing OOD literature [1,2]?
4. In Section 7.2, the definition of the proxy metric is ambiguous. Please provide its formal mathematical expression and a complete description of its construction and purpose.
5. In Figure 2, the adversarial accuracy of Tie Training plateaus at the beginning as the number of samples increases. Can the current theory account for this phenomenon?

**References:**

[1] Understanding the failure modes of out-of-distribution generalization. ICLR, 2021.

[2] Feature contamination: Neural networks learn uncorrelated features and fail to generalize. ICML, 2024.

**Limitations:**

yes

**Strengths And Weaknesses:**

**Strengths:**

1. The paper is well-written, with clear motivation and precise, consistent mathematical notation.
2. The theoretical derivation is rigorous and logically coherent. Experiments span from linear models to large language models (LLMs), validating the theoretical predictions across multiple settings.
3. Spurious correlations in preference alignment constitute a critical and timely challenge in LLM alignment. The paper makes an original contribution by providing a preliminary theoretical framework for this issue and designing a corresponding mitigation strategy, offering valuable guidance for developing more robust alignment algorithms.

**Weaknesses:**

1. The theoretical analysis relies on a local regime assumption. In practical preference alignment training—where the parameter β or gaps between preference pairs may be large—this assumption may not hold. Moreover, all derivations are confined to linear settings, limiting applicability to nonlinear models.
2. Tie Training requires pairs of preferences with similar utilities, which may be difficult to construct reliably in real-world training scenarios, potentially hindering practical deployment.
3. The LLM experiments remain somewhat toy-like and lack validation on larger-scale models or more diverse tasks, weakening the empirical support for broader applicability.

---

> ### Author Rebuttal · Authors · 2026-03-31
>
> ## Reviewer 5Z3G
> Thank you for your review and for recognizing the theoretical framework and its validation across settings.
>
> **Reminder of contribution.** We provide a mechanism-level analysis of spurious correlations in preference optimization. In the local log-linear DPO regime, mean spurious bias $\mu_s$ and causal–spurious leakage $\Sigma_{cs}$ induce spurious reliance at equilibrium (Theorem 4.1), yielding irreducible deployment error (Theorem 5.3). These mechanisms directly prescribe tie training as the intervention (Theorem 6.2).
>
> ### W1. Scope of the local regime
> **Concern.** The theory relies on local regime.
>
> **Response.** We agree that the closed-form results are derived in the local regime, and we will state this boundary more explicitly. Our claim is mechanism-level and constructive: the local analysis identifies what drives spurious learning, why it creates an irreducible term, and why tie training is the intervention. The neural and LLM results then test this pattern beyond the assumptions. See [Reviewer apky → W1] for more details.
>
> ### W2. Tie construction
> **Concern.** Constructing ties may be difficult in practice.
>
> **Response.** We agree that constructing ties can be challenging in practice, and we will state this limitation more explicitly in the revision. Importantly, our method does not require exact equality or perfect knowledge of spurious features. Ties can be built by direct perturbation when a spurious factor is known, by content-preserving rewrites over candidate axes, or by more agnostic constructions that approximate isotropic ties; the latter are still covered by our theory (Corollary 6.3). See [Reviewer apky → W3] for more details.
>
> ### W3. Controlled settings vs. broader evaluation
> **Concern.** LLM experiments use a controlled setting.
>
> **Response.** We agree that the benchmark is controlled and that broader evaluation on more realistic preference tasks is important future work. The controlled construction is deliberate:  it separates causal utility from spurious attributes and varies spurious statistics across training and deployment while holding the causal preference fixed. This level of control is generally not available in realistic settings and is exactly what is needed to test our theory. See [Reviewer apky → W4] for more details.
>
> ### Q1. Proxy objective vs. intended goal
> **Answer.** "Proxy objective" refers to the effective training signal the model exploits, including non-causal correlates that help reduce training loss. "Intended goal" refers to the preference-relevant utility that should remain invariant when only spurious features change. We will revise to make this distinction explicit and connect it to Assumption 3.1.
>
> ### Q2. Assumption 3.2
> **Answer.** We do not claim that Assumption 3.2 holds globally throughout practical preference optimization. It defines a local analytic regime, e.g., near initialization or bounded features, in which the population objective becomes tractable and the spurious-learning mechanism can be identified.
>
> - *New experiment.* We now probe this boundary directly in the log-linear DPO experiment with nonlinear GD and SGD. We report the regime diagnostic $|\beta \theta^\top \Delta \phi|$, which shows where the local approximation holds. Code available at [[LINK]](https://github.com/anonymous-ml-submission-1304/tie-training).
>
> - *Beyond the regime.* We also test nonlinear model and LLM, where the assumption need not hold, and still observe the same qualitative pattern, suggesting the structural insight persists beyond the approximation.
>
> ### Q3. Connection to OOD literature
> **Answer.** The connection is genuine. Nagarajan studies  spurious-feature-induced failure under distribution shift in classification, and like our work, they use linear analysis to obtain tractable characterizations. We instead study pairwise preference optimization, derive an irreducible shift term, and propose tie training as a mitigation. Zhang identifies a complementary failure mode (feature contamination from SGD) outside our scope. We will cite both.
>
> ### Q4. Proxy metric definition
> **Answer.** We agree the proxy metrics should be defined formally at first use. We define *spurious gap* as $ACC_{\text{aligned}} - ACC_{\text{unaligned}}$,  where $ACC_{\text{aligned}}$ and $ACC_{\text{unaligned}}$ denote pairwise preference accuracy when spurious features align or conflict with true utility. *Adversarial accuracy* as accuracy under reversed spurious correlation. We will move these definitions into the main text and clarify their purpose.
>
> ### Q5. Theory vs. observed plateau
> **Answer.** Qualitatively, per Theorems 5.3 & 6.2, deployment error decomposes into estimation and shift components. At small $n$, estimation error dominates and tie training benefit is not yet visible; as $n$ grows, estimation error decays and the advantage of tie training emerges.
>
> **Closing.** We hope these clarifications address your concerns and would appreciate an updated assessment.

---

> > ### Author Rebuttal · Reviewer_5Z3G · 2026-04-03
> >
> > Thanks for the authors' reply. I have no further questions and will increase my score.

---

> > > ### Author Response · Authors · 2026-04-06
> > >
> > > Thank you for the thoughtful feedback and for taking the time to review our rebuttal. We are glad the clarifications addressed your concerns. We will incorporate your suggestions, including clarifying the scope of the theory, strengthening the empirical presentation, and improving the definitions and discussion, in the final version.

---

### Official Review · Reviewer_apky · 2026-03-11

**Soundness:** 3
**Presentation:** 3
**Significance:** 2
**Originality:** 3
**Overall Recommendation:** 5
**Confidence:** 3

**Summary:**

This paper studies the problem of spurious correlations (such as verbosity and format) in LLM preference optimization, which causes misgeneralization under distribution shift. The authors propose tie training. The core idea is to add training pairs with equal utility but different spurious features, and assign random preferences. This method aims to reduce the model’s reliance on non-causal features.

**Compliance With Llm Reviewing Policy:**

Affirmed.

**Final Justification:**

My questions are all solved, so I'm willing to raise my score.

**Key Questions For Authors:**

All key questions are included in the Weaknesses section.

**Limitations:**

The proposed method relies on manual tie construction and strong local assumptions, which limit its effectiveness in real LLM alignment scenarios.

**Strengths And Weaknesses:**

## Strengths
1. The paper is well-written and logically clear.
2. The decomposition of deployment suboptimality into estimation and shift terms provides a solid theoretical framework for reward hacking.
3. Mitigating spurious correlations in LLM alignment is an important and practical topic.

## Weaknesses
1. The theoretical analysis strongly depends on the local regime assumption $|\beta_{\tilde{\theta}}^\top \Delta\phi| \ll 1$, which is often violated in real DPO training. The theory cannot well explain the late-stage training dynamics.
2. The method needs manual construction of informative ties, which limits its practical application. Scalable and automatic construction is still unsolved.
3. The LLM experiments are only conducted on a synthetic dataset. The authors do not verify the effectiveness on real human preference data.

---

> ### Author Rebuttal · Authors · 2026-03-31
>
> ## Reviewer apky
>
> Thank you for your review. We are encouraged that you found the theoretical framework and the framing of tie training well-motivated.
>
> **Reminder of contribution.** We provide a mechanism-level analysis of spurious correlations in preference optimization. In the local log-linear DPO regime, mean spurious bias $\mu_s$ and causal–spurious leakage $\Sigma_{cs}$ induce nonzero spurious parameters at equilibrium (Theorem 4.1), yielding irreducible deployment error (Theorem 5.3). These mechanisms directly prescribe tie training as the intervention (Theorem 6.2).
>
> ### W1. Scope of the local regime
>
> **Concern.** Theory relies on local regime.
>
> **Response.** We agree the closed-form results hold in the local regime and will make this boundary more explicit. Our claim is constructive and mechanism-level: the theory identifies the mechanisms that drive spurious learning, shows how they induce an irreducible term, and directly motivates tie training as a targeted intervention.
>
> - *Key point.* The local analysis is used to derive what drives spurious learning and how tie training changes it; the mechanism is then tested beyond the assumptions.
>
> - *Implication.* Even when late-stage training leaves the regime, the theory remains useful because it prescribes the intervention, while neural and LLM experiments verify the same qualitative pattern: spurious reliance emerges, scaling alone does not remove error, and tie training reduces it.
>
> - *Evidence.* Theorem 4.1, Theorem 5.3, Theorem 6.2, Corollary 6.3, and the nonlinear and LLM experiments. Additionally, we now report the diagnostic $|\beta \tilde \theta^\top \Delta \phi|$ in our log-linear experiments, confirming where the local approximation holds and where it breaks down; code is available at [[LINK]](https://github.com/anonymous-ml-submission-1304/tie-training).
>
> ###  W2. Tie construction
>
> **Concern.** Constructing informative ties may require manual design.
>
> **Response.** We agree that fully automatic tie construction is not solved, and we will state this limitation more explicitly. Our method, however, is not restricted to hand-crafted ties: the mechanism we analyze naturally suggests three practical regimes:
>
> 1. *Known spurious attribute.* When the spurious factor is identified (as in our LLM experiments), ties can be constructed by direct perturbation.
> 2. *Candidate axes.* When the spurious direction is unknown but a set of plausible candidates can be listed (e.g., verbosity, style, formatting), ties can be constructed by applying content-preserving perturbations across these axes to generate random ties.
> 3. *Fully agnostic.* Cross-prompt constructions provide a way to form tie-like pairs that perturb multiple directions simultaneously.
>
> Both (ii) and (iii) approximate isotropic ties, for which Corollary 6.3 guarantees spurious suppression, though less efficiently than (i).
>
> - *Key point.* Our theory further reveals that suppression effectiveness depends on the alignment between ties and the induced spurious geometry through $\Sigma_{ss}$; direction matters, not just volume.
>
> - *Implication.* Under a finite tie budget, the main open problem is therefore not only whether ties can be constructed beyond manual design, but how to allocate them most effectively. Scalable tie allocation is an important direction for future work.
>
> ### W3. Controlled setting vs. natural data
>
> **Concern.** Controlled experiments raise questions about generalization.
>
> **Response.**  We agree that the benchmark is controlled and will make this limitation explicit. The controlled construction is deliberate: it lets us separate causal and spurious attributes, vary the spurious statistics across training and deployment, and keep the underlying preference structure fixed. This level of control is generally not possible in real preference data, where causal and spurious factors are entangled, and is necessary to directly test the paper's central claims. Note that the spurious correlations we study are reported in deployed alignment systems, so the mechanisms we isolate are practically relevant even though the benchmark is synthetic.
>
> - *Key point.* The benchmark is designed to isolate the paper’s claims: spurious reliance emerges under correlation, persists under shift, and is reduced by tie training.
>
> - *Implication.* This makes the experiment well-suited for mechanism validation, while broader evaluation on natural preference data remains an important next step. Importantly, the method itself does not require explicit identification of spurious factors; the controlled construction is used only to make the mechanism empirically testable.
>
> - *Evidence.* The LLM experiment holds the preference structure fixed while varying the spurious correlation, and Corollary 6.3 supports the unknown-spurious setting by showing that isotropic ties suppress spurious reliance.
>
> **Closing.** We hope these clarifications address your concerns and would appreciate an updated assessment.

---

> > ### Author Rebuttal · Reviewer_apky · 2026-04-03
> >
> > Thanks for the replies. My questions are all solved, so I'm willing to raise my score.

---

> > > ### Author Response · Authors · 2026-04-06
> > >
> > > Thank you for the thoughtful feedback and for taking the time to review our rebuttal. We are glad the clarifications addressed your concerns. We will incorporate your suggestions, including clarifying the scope of the theory, strengthening the empirical presentation, and expanding the discussion of tie construction and tradeoffs, in the final version.

---

### Official Review · Reviewer_Zszm · 2026-03-12

**Soundness:** 3
**Presentation:** 4
**Significance:** 2
**Originality:** 3
**Overall Recommendation:** 4
**Confidence:** 3

**Summary:**

This paper studies spurious correlation learning in preference optimization, focusing on a DPO-style setting. Its main claim is that even when true preferences depend only on causal features, preference optimization can still learn non-causal spurious features at the population level. The paper attributes this to two mechanisms: mean spurious bias and causal–spurious correlation leakage. It then argues that this induces an irreducible deployment vulnerability under spurious shift, in the sense that additional data from the same training distribution reduces estimation error but does not eliminate the shift-induced component. Finally, the paper proposes tie training, where equal-utility pairs with randomized labels are used to selectively reduce spurious reliance, and supports this proposal with both theoretical analysis and experiments in linear, neural, and LLM-based settings.

**Compliance With Llm Reviewing Policy:**

Affirmed.

**Final Justification:**

My concerns W1, W2, and W3 have not been fully resolved, so my final recommendation is weak accept.

**Key Questions For Authors:**

Please address the weaknesses above. I also have the following specific questions:

* **Q1:** How sensitive is tie training to approximate ties? In particular, how much utility mismatch can be tolerated before the method becomes ineffective?

* **Q2:** How should practitioners construct informative ties in realistic pipelines where causal and spurious attributes are not explicitly known? The paper offers some discussion in lines 2184–2188, but it is still not clear what concrete procedure a practitioner should follow in such settings.

* **Q3:** How should one choose $\alpha$ in practice?

**Limitations:**

yes

**Strengths And Weaknesses:**

### Strengths

* Clear and coherent theory-to-method story. The paper has a strong internal structure. It explains why spurious learning arises, why it matters under deployment shift, and then proposes a mitigation that directly targets the identified mechanism. The paper is also well written.

* The irreducibility claim is conceptually important. The decomposition between estimation error and shift-induced error is one of the most compelling parts of the paper. It makes the important point that scaling data from the same training distribution does not necessarily resolve the problem.

* Tie training is simple and well-motivated. Using equal-utility pairs with randomized labels to inject curvature along spurious directions is a clever idea.

---

### Weaknesses

* **W1:** The main theory relies on a local linearized regime. The clean formulas and the main mechanism theorem depend on the regime in which the DPO objective can be linearized. This is a reasonable starting point, but it limits how strongly I can interpret the theory for realistic large-model preference optimization. The finite-sample deployment result also relies on additional assumptions such as boundedness and a geometry-transfer condition.

* **W2:** The empirical setting is still relatively controlled. While I liked the LLM experiment, it is based on a controlled synthetic hotel preference benchmark that explicitly separates causal utility from spurious features and manipulates those correlations across train and deployment conditions. This is useful as a mechanism test, but I would be more convinced by evidence on more natural preference datasets or settings in which the spurious attribute is not explicitly engineered.

* **W3:** The role of the mixing weight $\alpha$ is not fully characterized, both theoretically and empirically. This matters because both extremes seem undesirable: when $\alpha \to 1$, the method approaches strict training and may retain spurious reliance, while when $\alpha \to 0$, the mixed-equilibrium formula suggests that learning becomes weak as tie data dominates. Thus, the interesting regime should lie at an intermediate $\alpha$, but Theorem 6.2 (iii) does not analyze how the full deployment suboptimality bound varies with $\alpha$, nor does it characterize the tradeoff between reducing spurious shift and preserving useful preference signal. This gap is also reflected empirically: the LLM section keeps the tie proportion fixed. It would strengthen the paper to include LLM results across several $\alpha$ values and show how both in-distribution and shifted accuracies change with $\alpha$. Even a simple special-case theoretical analysis, together with an $\alpha$ sweep in the LLM setting, would make the practical implications much clearer.

---

> ### Author Rebuttal · Authors · 2026-03-31
>
> ## Reviewer Zszm
> Thank you for your review and for recognizing the clear theory-to-method narrative of our paper.
>
> **Reminder of contribution.** We provide a mechanism-level analysis of spurious correlations in preference optimization. In the local log-linear DPO regime, mean spurious bias $\mu_s$ and causal–spurious leakage $\Sigma_{cs}$ induce nonzero spurious parameters at equilibrium (Theorem 4.1), yielding irreducible deployment error (Theorem 5.3). These mechanisms directly prescribe tie training as the intervention (Theorem 6.2).
>
> ### W1 - Scope of the local regime.
> **Concern.** Theory relies on local regime.
>
> **Response.** We agree the theoretical results are derived under a local regime and standard regularity conditions, and we will state this boundary more explicitly. Our claim is mechanism-level and constructive: the local analysis identifies what drives spurious learning, why it creates an irreducible term (under regularity conditions), and why tie training is the intervention. The neural and LLM results then test this pattern beyond the assumptions. See [Reviewer apky → W1] for more details.
>
> ### W2. Controlled setting vs. natural data
> **Concern.** Controlled experiments raise questions about generalization.
>
> **Response.** We agree that the benchmark is controlled and that evaluation on natural preference data is important future work. The controlled construction is deliberate:  it separates causal from spurious attributes and varies spurious statistics across training and deployment while holding the causal preference fixed. This level of control is generally not available in natural preference data and is exactly what is needed to test the theory. See [Reviewer apky → W4] for more details.
>
> ### W3. Role of $\alpha$
> **Concern.** Role of $\alpha$ is not fully characterized.
>
> **Response.** We agree that the $\alpha$ tradeoff deserves fuller treatment and will address this in revision with both theory and experiments.
>
> *Key point.* $\alpha$ controls the strict–tie mixture, so decreasing $\alpha$ increases tie mass.
>
> *Implication.* Corollary 6.3 establishes monotone spurious shrinkage as tie mass increases. We will add a complementary result showing that both causal and spurious components vary with $\alpha$, with the spurious component more sensitive, formalizing the tradeoff the reviewer identifies. Preliminary log-linear results already support this claim and are available in our tie-training code repository. Code available at [[LINK]](https://github.com/anonymous-ml-submission-1304/tie-training) .
>
> *New experiment.* Pilot LLM $\alpha$-sweep (augmentation, 5k strict pairs, 2k adversarial test, single seed): $\alpha$=1.0: 68.9%, $\alpha$=0.9: 74.5%, $\alpha$=0.7: 76.3%, $\alpha$=0.5: 82.4% adversarial accuracy. Monotone improvement confirms the theory. Full-scale results, reporting in-distribution and shifted accuracy, with multiple seeds in the revision.
>
> ### Q1. Tolerance to approximate ties
> **Answer.** Approximate ties are naturally supported. Exact utility equality is not required: near-ties preserve a weak causal margin while injecting variance along spurious directions.
>
> *Theory.* Our framework extends to near-ties by replacing ideal tie statistics with approximate ones, yielding explicit tolerance conditions on causal mismatch. We will provide full theoretical characterization in the revision. Results from the log-linear approximate ties training sweep are available in our code repository.
>
> *Experiment.* Our LLM and nonlinear experiments already operate in this regime (ties use approximate utility).
>
> ### Q2. Tie construction
> **Answer.** LLM experiment demonstrates one procedure: select response pairs with similar reward scores but differing surface attributes. We discuss in [Reviewer apky → W3] practical regimes for tie construction that our theory also supports, including when spurious features are unknown. Fully automatic tie construction remains open and will be stated explicitly in the revision.
>
> ### Q3. Selecting $\alpha$
> **Answer.** As discussed in W3, $\alpha$ is an analytic quantity in the mixture model. In practice, it corresponds to tie budget.
>
> A practical guideline: use a moderate tie budget, sufficient to induce spurious-direction regularization while preserving preference signal. We used 0.3 in the LLM experiment, deliberately not optimized, to demonstrate that tie training works.
>
> Our theory shows that performance depends not only on how many ties are added ($\alpha$), but which ones. Suppression scales with alignment to $\Sigma_{ss}$, so under a fixed tie budget, selecting informative pairs can yield substantially greater suppression gains than uniformly increasing $\alpha$. This reframes the practical problem from tuning $\alpha$ to allocating ties to high-impact directions. Developing scalable methods for this surgical allocation is an important open problem.
>
> **Closing.** We hope these clarifications address your concerns and would appreciate an updated assessment.

---

> > ### Author Rebuttal · Reviewer_Zszm · 2026-04-04
> >
> > Thank you for the helpful rebuttal. I appreciate the authors’ clarifications and their responses to my comments. While concerns W1, W2, and W3 have not been fully resolved, I remain comfortable with my current positive score and will not increase it further.

---

> > > ### Author Response · Authors · 2026-04-06
> > >
> > > Thank you for your careful review and for engaging with our rebuttal. We appreciate your feedback and will incorporate your suggestions to further clarify the presentation and strengthen the final version.

---

### Decision · Program_Chairs · 2026-04-30

**Decision:**

Accept (regular)

**Comment:**

This paper study the problem of preference learning methods like DPO causing language models to rely on spurious correlations through mean spurious bias and causal spurious correlation leakage. The authors argue that adding more training data from the same distribution fails to remedy the problem and as a result models can fall into sycophancy and length bias. The authors propose tie training, a data augmentation strategy that uses equal utility preference pairs, to reduce spurious learning. Empirical results are provided on both simple models and large models. Reviewers found the problem to be compelling, and appreciated the creative solution provided by the authors. In addition, the reviewers found the experimental evidence to be compelling. While there was some shared concern around the local linearity assumption, these concerns were largely assuaged during discussion where the authors have promised a more clearly stated discussion surrounding the theoretical results.